



# Influence of cloudy/clear-sky partitions, aerosols and geometry on the recent variability of surface solar irradiance's components in northern France

Gabriel Chesnoiu[1], Nicolas Ferlay[1], Isabelle Chiapello[1], Frédérique Auriol[1], Diane Catalfamo[1], Mathieu Compiègne[2], Thierry Elias[2], and Isabelle Jankowiak[1]

[1]Université de Lille, CNRS, UMR 8518 - LOA, Lille F-59000, France
[2]HYGEOS, Euratechnologies, Avenue de Bretagne, 59000 Lille, France

**Correspondence:** Nicolas Ferlay (nicolas.ferlay@univ-lille.fr)

**Abstract.** The surface solar irradiance (SSI) is a fundamental parameter whose components (direct and diffuse) and variabilities are highly influenced by changes in atmospheric content and scene's parameters. The respective importance of cloudy sky conditions and atmospheric aerosols on SSI evolutions is region dependent and only partially quantified. Here we provide a comprehensive analysis of SSI variabilities recorded in northern France, a region with extensive variabilities of sky conditions and aerosol loads. Through the application of automatic filtering methods on 1 min resolution SSI ground-based measurements over Lille, sky conditions are classified as clear-sky, 11 %, clear-sun-with-cloud, 22 %, and cloudy-sun situations, 67 %, over 2010 to 2022, for which we analyze the statistics and variabilities of the global horizontal (GHI), direct (BHI) and diffuse (DHI) solar irradiances. Coincident photometric measurements of aerosol properties and radiative transfer simulations provide the mean to conduct a multivariate analysis of the SSI observed trends and year-to-year evolutions, and to estimate aerosol and cloud forcings under clear-sun conditions. The analysis of the record value of all sky GHI in spring 2020 attributes 89 % of the changes to the exceptional sunlight conditions (57 % of clear-sun situations). It highlights also for that season the importance of solar zenithal angle's changes whose positive effect on clear-sun conditions surpasses those due to aerosols. Our results show all-sky GHI and BHI positive trends of around +4 and +4.4 W/m²/year respectively, both in spring and summer, that are explained at more than 60 % by an increase of clear-sun occurrences of +1 % per year. Additional significant BHI's increase under clear-sun conditions are mainly explained in spring by the negative trend in aerosol optical depth (-0.011 per year), partly by angular effects in summer. Moreover, we find that clear-sun with cloud situations are frequently marked by irradiance enhancement due to clouds, with on monthly-average 13 % more GHI and 10 % additional diffuse proportion than in clear-sky situations. Under such conditions, clouds add on average 25 W/m² of diffuse irradiance that set the amount of solar irradiance at the remarkable level of pristine (aerosol and cloud free) conditions, or even higher by more than +10 W/m² in summer and for low aerosol loads. Overall, our results highlight the dominant and complex influence of cloudy conditions on SSI, which precedes or combines with that of aerosols and geometrical effects, and leads to remarkable global level of SSI in clear-sun with-cloud situations.



## 1 Introduction

The amount of solar energy that reaches the Earth's surface plays a critical role in the Earth's energy balance and governs a
wide range of key physical processes, including evaporation and associated hydrological components, snow and glacier melt,
plant photosynthesis and related terrestrial carbon uptake, as well as the diurnal and seasonal course of surface temperatures.
Furthermore, the amount of solar irradiance incident at the surface (SSI) has significant implications for solar energy production
technologies and agricultural productivity.

Studies conducted in recent decades, both through observations and modeling, suggest that surface solar radiation is not
necessarily constant on a decadal time scale. Instead, it exhibits significant decadal variations with a worldwide decreasing
(dimming) trend until the $1990^s$, followed by an increasing (brightening) trend from then onwards (Wild et al., 2005; Wild,
2009; Liepert, 2002; Norris and Wild, 2007).

Determining the cause of the observed trends in SSI has proven to be a challenging task. However, an increasing number
of studies indicate that the aerosol-radiation interaction is likely responsible for an increase in all-sky radiation since 1985
(Philipona et al., 2009; Manara et al., 2016; Ruckstuhl et al., 2008; Wild et al., 2021). It should be noted that the timing and
intensity of the minimum cannot be accurately simulated using current aerosol emission inventories, as reported by Liepert and
Tegen (2002); Romanou et al. (2007); Ruckstuhl and Norris (2009); Turnock et al. (2015). This is due to the high temporal and
spatial variability of clouds and aerosols which is particularly difficult to represent in models.

Regional studies that record changes in surface radiation while taking into account atmospheric parameters that can impact
such changes are crucial, as they provide more insight on the relationship between the variations of these parameters and the
changes in surface radiation. A few studies were recently conducted in the Benelux region, especially over the Netherlands
(Boers et al., 2017, 2019; Heerwaarden et al., 2021), and over larger European areas (Pfeifroth et al., 2018; Schwarz et al., 2020)
characterized by substantial proportions of clouds (Hahn and Warren, 2007) as well as a high population density which results
in elevated levels of atmospheric particulate pollution. Most of these studies have so far primarily focused on the analysis
of the impact of clouds and aerosols on the evolution of the global surface solar radiation trends, and did not consider the
variability of the direct and diffuse components. However, depending on their optical properties, aerosols and clouds influence
incident radiation by altering both the direct (increase) and diffuse (decrease) components. Although the impact on photovoltaic
production depends mainly on the total amount of solar irradiation reaching the surface, the performances of technologies are
also sensitive to the partition between diffuse and direct components (Kirn et al., 2015; Lindsay et al., 2020).

The study of incident direct and diffuse radiations is particularly important in the context of climate change as significant
changes in the composition of the atmosphere are expected by the end of the century, with varying magnitudes depending on
the corresponding SSP (Shared Socioeconomic Pathways) and RCP (Representative Concentration Pathway) scenario (Moss
et al., 2010; Hauglustaine et al., 2014; Drugé et al., 2021), leading to potentially very different solar environments. Moreover,
it is now established that mitigating the impact of greenhouse gas emissions on climate change requires the development
of alternative methods of energy production. Currently, photovoltaic technologies represent one of the most important and
promising technologies, in addition to energy produced by wind turbines. Several studies have investigated the future resilience





of renewable installations (Tobin et al., 2018), especially photovoltaic ones (Gutiérrez et al., 2020; Hou et al., 2021; Jerez et al., 2015). These studies show that the future evolution of the solar environment and of the surface temperature, will have a significant impact on the future development of these technologies and their production.

In the current study, we analyze the variability of the global surface solar irradiance and its direct and diffuse components in northern France over the years 2010-2022 as measured at the ATOLL (ATmospheric Observations in LiLle) platform, as well as the variability of coincident sub-hourly measurements of aerosol optical properties. Our analysis relies, in particular, on the development of a classification of the sky conditions based on automated processing of irradiance measurements by cloud filters and identification of aerosol's class. Beside obtaining climatologies at our site, our objective is to disentangle and

quantify the influence of the different parameters that contribute to year-to-year irradiance variabilities observed in Lille over the period 2010-2022. These classifications allow us to distinguish the influence of the occurrence of cloudy and clear-sky situations from the influence of the atmospheric content (aerosols and gases) and other parameters, such as the solar zenith angle, on the variability of the SSI and its direct and diffuse components. The effect of atmospheric parameter's variability is quantified with radiative transfer simulations in clear-sky conditions.

Section 2 provides a brief overview of the data and methods used to identify cloudy and clear sky moments. A classification of the aerosol content based on the optical properties measured coincidentally as part of the AERONET network is also introduced. The aim of this classification is two-fold. On the one hand, it enables a more precise characterization of the radiative impact of aerosols observed in Lille through the definition of new models of aerosol optical properties which are used to perform simulations using the radiative transfer code SOLARTDECO. On the other hand, it facilitates the study of the variability of

the aerosol content in Lille. In particular, used in tandem with the classification of the sky conditions and SOLARTDECO simulations it enables the quantification of the contribution of the different atmospheric and geometric parameters to the variability of the surface solar irradiance over the period 2010-2022, as described in Section 3. An analysis of the direct radiative effect of atmospheric particles, aerosols and clouds, over 2010-2022 is also presented in Section 4 for various sky conditions. Finally, Section 5 provides the concluding remarks and outlines the possible directions for future works.

## 2   Data and methods

In this section, the measurement datasets, algorithms and methods used in this study are described. Information on the data availability is given separately at the end of the article.

### 2.1   Ground-based measurements from the ATOLL platform

#### 2.1.1   Description of the site

This study relies on the coincident aerosol optical properties and surface solar irradiance measurements routinely performed at the ATOLL platform located in the North of France (50.61°N, 3.14°E, 70 m a.s.l), 6 km southeast of Lille downtown area, on the rooftop of a building of the University of Lille in Villeneuve d'Ascq. As the metropolitan area of Lille is characterized




by a high population density and more than 1.1 million inhabitants, the ATOLL site can be considered as suburban. As shows Figure 1, with the Belgian border at a distance of roughly 15km, the site is located very close to the Benelux region, one of the most densely urbanized and industrialized area of northwestern Europe. It is thus influenced by many anthropogenic emission sources, both regional and trans-boundary, with strong contributions from the residential and industrial sectors, as well as transport and agriculture, which contribute to relatively frequent particulate pollution episodes (Potier et al., 2019; Favez et al., 2021; Velazquez-Garcia et al., 2023). It should be noted that the site is also subject to a significant maritime influence due to its proximity with the English Channel and the North Sea, located less than 80 km away. According to the Köppen-Geiger climate classification (Beck et al., 2018) the climate is described as Cfb, meaning it is mostly temperate (C) with a warm summer (b) and an absence of dry season (f). The typical precipitation's amount is of about 600 to 800 mm/yr which is related to a relatively high frequency of clouds throughout the year with average cloud fraction values ranging between 60 to 80% depending on the time of year (Warren et al., 2007).

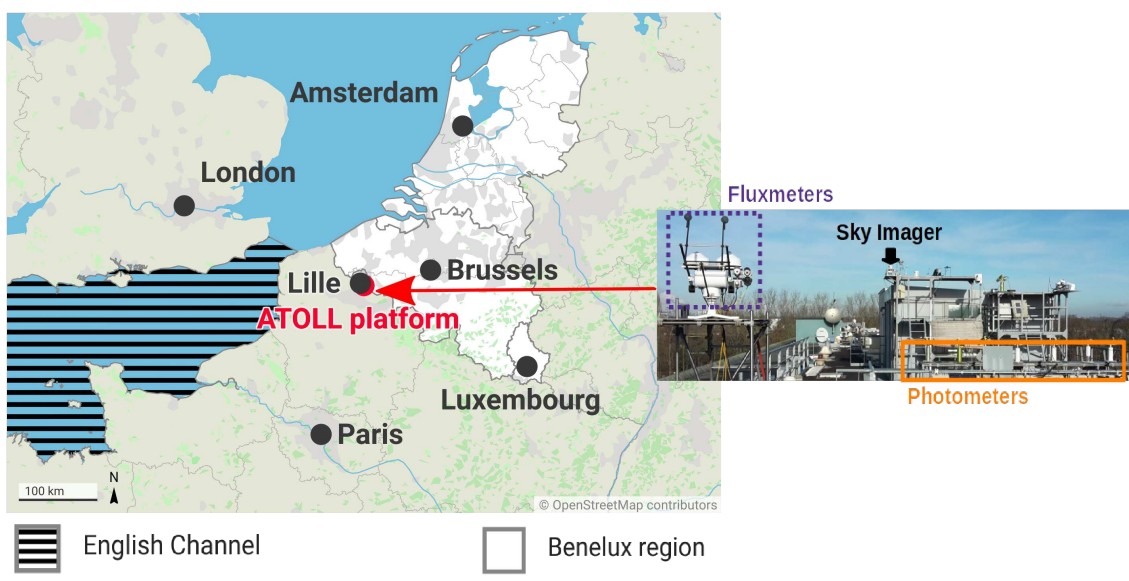

**Figure 1.** Map representing the location of the ATOLL platform (red marker). The white area represents the Benelux region and the black hatches the English Channel. Adjacent to the map is a picture of the platform including the main instruments used in this study, i.e. the set of pyronameter and pyrheliometer on a Sun-tracking device (purple box), the AERONET sunphotometers (orange box) and the sky imager (black arrow).

### 2.1.2 Irradiance measurements

A Kipp & Zonen CHP1 pyrheliometer, mounted on a sun-tracking device, measures the Direct (or beam) Normal Irradiance (DNI) incident at the surface in the direction of the Sun with a field of view of $5\pm0.2°$. The Beam Horizontal Irradiance (BHI) is then derived from these measurements using the cosine of the solar zenith angle ($\mu_0 = \cos(\text{SZA})$) as $\text{BHI} = \text{DNI} \times \mu_0$. In



addition to the pyrheliometer, a Kipp & Zonen CMP22 pyranometer is also placed on the sun-tracker. The pyranometer is paired with a shading sphere that blocks the incoming direct irradiance, allowing for coincident measurements of the Diffuse

Horizontal Irradiance (DHI). The Global Horizontal Irradiance (GHI) is then computed as the sum of the BHI and DHI. Such a procedure has the advantage of reducing cosine errors at low sun angles compared to measurements obtained with a single pyranometer (Michalsky et al., 1999). Both instruments measure radiation in the broadband range between 210 and 3600 nm covering most of the solar spectrum (200 - 4000 $\mu$m). Therefore, we have access to measurements of the incoming surface solar irradiance and its direct and diffuse components on an horizontal plane at a 1-minute time resolution since February 2009.

To guarantee the smooth operation of the system, regular maintenance is performed, including daily cleanings. Moreover, both instruments were calibrated in 2012, 2017, and 2022, and their sensitivities have remained relatively stable throughout the period. Under such operational conditions, the uncertainties in the measurements are expected to be of the order of 2 to 3% for the GHI and DHI, and 1.5% for the BHI, as typically reported in the literature (Derimian et al., 2008, 2012; Vuilleumier et al., 2014; Wild et al., 2021). It is worth noting that there are some missing values in the measurement series for winter months

of several years, particularly in January and February. This gap occurs because the instruments were regularly sent either in M'Bour (Senegal), for calibration of local instruments, or in Delft (Netherlands), for a recalibration by the manufacturer.

### 2.1.3 Aerosol measurements

A CIMEL sun/sky photometer from the PHOTONS (PHOtométrie pour le Traitement Opérationnel de Normalisation Satellitaire) network, which is the French part of AERONET (Aerosol Robotic Network) (Holben et al., 2001) is used to characterize

and monitor aerosol optical and columnar microphysical properties in Lille. The photometer provides direct measurements of the Aerosol Optical Depth (AOD) at six different wavelengths (340, 380, 440, 500, 870 and 1020 nm) along with their corresponding Angström Exponents (AE). In addition, Precipitable Water Vapor content (PWV in cm) is derived from measurements at 940 nm. AERONET almucantar inversions (Dubovik and King, 2000; Dubovik et al., 2002) of aerosol size distribution, complex refractive index (m = n + ik), and single scattering albedo (SSA) were also used in this study. However, as it requires a

relatively important air mass and a perfectly cloudless sky, the amount of available inversions is small (less than 500 data points). Their climatological weight is thus relatively limited. Therefore, they have been mainly used to compute site-specific aerosol optical properties through Mie calculations, as described in Section 2.3.2. In this study, mostly level 2.0 (cloud-screened and quality controlled) data has been used, except for the measurements from September to December 2022, for which only level 1.5 (cloud-screened only) data were available at the time of writing. The uncertainty in measured AOD is estimated to be

of approximately 0.02 for the UV channels (340 and 380 nm) and 0.01 otherwise (Giles et al., 2019). The reported uncertainty in the PWV measurements is around 10-15 % (Pérez-Ramírez et al., 2014; Smirnov et al., 2004), with a dry-bias of around 5-6 % (Pérez-Ramírez et al., 2014). The retrieval of complex refractive index and volume size distribution is a challenging task, and the associated uncertainties are highly variable depending on the observing conditions. For example, the uncertainty in the volume size distribution ranges from 15 to 100 %, depending on the size of the particles, while for the imaginary part of the

refractive index (k), it varies between 30 and 50 % for strongly and weakly absorbing aerosols, respectively (Dubovik et al., 2000; Dubovik and King, 2000; Nakajima et al., 1983, 1996).





### 2.1.4 Coincidence of aerosol and irradiance measurements

For this study, we chose to analyze the period from 2010 to 2022, as it represents the longest period of continuous coincident aerosol and irradiance measurements available. The number of coincident data points is mainly determined by the frequency of
AERONET direct observations. Prior to 2016, the standard time resolution of the photometric measurements was 15 minutes, which was later increased to 3 minutes due to a change in the instrument's model. This temporal resolution can be considered a maximum, as it is often reduced by the presence of clouds in the direction of the Sun. The complete dataset of synchronous aerosol and irradiance measurements comprises approximately 82 000 coincident observations in clear-sun conditions.

### 2.1.5 Other measurements

In addition to aerosol optical properties and irradiance measurements, other coincidental observations from the ATOLL platform have been used. Notably, sky images captured by a sky-imager (CMS Schreder VIS-J1006) since 2009, with a time resolution of 3 minutes, have been visually examined to assess the cloud cover state over short periods of time. This manual evaluation of the sky images helped validate more systematic cloud-screening methods based on irradiance measurements, as described in Section 2.2. Meteorological observations of relative humidity (RH), wind speed, and wind direction have also
been used to perform simulations of the surface solar irradiance in clear-sky conditions, and to complete the climatological study of the aerosol content and irradiance measurements, as described in Sections 2.3.1 and 3.1.

### 2.2 Classification of the sky conditions over Lille

To analyze the variability of surface solar irradiance and understand its underlying causes, it is necessary to first assess the sky conditions. Indeed, the presence of clouds, particularly in the direction of the Sun, leads to situations with very contrasting
levels and characteristics of surface solar irradiance. The aim is thus to distinguish these different situations in order to adapt the analysis of the sensitivity of the surface solar irradiance. Clear-sun (CSUN) situations, when the Sun is visible with (clear-sun with clouds, CSWC) or without (clear-sky, CSKY) residual clouds in the sky, are characterized by high levels of surface solar irradiance with a predominant contribution of the direct component. They are more sensitive to the aerosol content than cloudy-sun (CLOS) situations, i.e. when the Sun is masked by clouds, for which the sensitivity of the SSI to the aerosol content
can be considered negligible aside from a few exceptions characterized by thin cirrus or isolated clouds. As only AERONET level 1.5 data or more is used in this study, some knowledge is already available on the state of the cloud cover as these datasets are automatically cloud cleared through the study of the variability of triplets of photometric measurements. However, there are some drawbacks. On the one hand, this method exclusively detects the presence of clouds in the direction of the Sun which means that it only distinguishes clear-sun moments from cloudy-sun moments. Thus, in clear-sun with clouds conditions, the
effect of residual clouds on the diffuse and global irradiances cannot be distinguished from that of aerosols. On the other hand, as the time resolution of the photometric measurements is lower than that available for irradiance measurements, it is less relevant for the intended climatological study. Hence two cloud-screening methods based on the available 1-minute irradiance measurements were selected for this study based on the performance analysis of Gueymard et al. (2019). The method of Batlles



et al. (2000) is used to distinguish clear-sun moments from cloudy-sun situations, while the algorithm of García et al. (2014)
isolates clear skies (no clouds, i.e. CSKY) from cloudy situations (CSWC and CLOS). This allows for the classification of
the sky conditions in three distinct categories (cloudy-sun, clear-sun with clouds, and clear-sky) with contrasting levels of
irradiance as described in Section 3.1.

This section describes the two methods used in this study and presents an evaluation of their performances for the months of
January and May 2018 by comparison with manual assessments of the cloud cover based on visual inspection of the coincident
sky images from ATOLL at a 3-minute resolution.

### 2.2.1 Description of the cloud-screening methods

The method of Batlles et al. (2000) was initially intended for the detection of clear-sky conditions from hourly surface solar
irradiance data. However, the study of Gueymard et al. (2019) showed that on a 1-minute basis it performs better for the
identification of clear-sun moments with results among the best performing methods. This is a fairly straightforward method
based on two criteria : (i) the clearness index ($K = \frac{GHI}{\mu_0 I_0}$) which represents the transmittance of the atmosphere relative to the
incoming extraterrestrial solar irradiance ($I_0$), should be larger than a threshold value, $K_{lim}$; while (ii) the fraction of diffuse
radiation ($D_f = \frac{DHI}{GHI}$) should be smaller than a maximum value $D_{f,lim}$. Note that both thresholds vary with the sun elevation
angle $h = 90 - SZA$ as defined in Equations 1 and 2:

$$K_{lim} = -0.3262 - 0.0032h + 0.6843log_{10}(h) \tag{1}$$

$$D_{f,lim} = 1.0827 - 0.3893log_{10}(h) \tag{2}$$

The method of García et al. (2014) is based on the algorithm originally presented in Long and Ackerman (2000). This
method relies on a series of four tests based on GHI and DHI measurements of high frequency (3 minutes or less). The first two
tests remove obvious cloudy moments characterized by extreme values of the normalized GHI (test 1) and measured diffuse
irradiance (test 2) through the definition of threshold values of both quantities. The third and fourth tests are more elaborate
allowing the more subtle detection of cloud covers through the analysis of the temporal variability of the global irradiance and
the normalized diffuse irradiance ratio, respectively. The normalization of the GHI and DHI in the first and fourth tests is based
on a power law of the cosine of the solar zenith angle as in Equation 3:

$$F_N = \frac{F}{\mu_0^{b_F}} \tag{3}$$

where F correspond to the measured global irradiance (GHI) or diffuse irradiance ratio ($D_f$) and the coefficient «$b_F$» represents
the variations of the associated clear-sky quantities with the cosine of the solar zenith angle $\mu_0$. The subscript «N» represents
the normalized quantities.

The four tests are applied in an iterative process to provide each time a new set of clear-sky moments, which are then used
to calculate coefficients for the first and fourth tests for days with enough clear-sky observations by fitting the GHI and $D_f$



1-minute measurements with respect to $\mu_0$, as in Equations 4 and 5 :

$$GHI = a_{GHI,\,day}\mu_0^{b_{GHI,\,day}} \tag{4}$$

$$D_f = a_{D_f,\,day}\mu_0^{b_{D_f,\,day}} \tag{5}$$

where GHI and $D_f$ are the observed global irradiance and diffuse ratio, the parameters «a» represent the associated clear-sky quantities for a solar zenith angle of 0°. The coefficients «b» correspond to the parameters used for the normalization of the GHI and $D_f$ as in Equation 3. A new set of parameters is then derived as the mean of the daily values and the iteration process continues until convergence is achieved within a tolerance of 5%. This method is thus quite versatile and can theoretically be applied to any observational site equipped with measurements of both global and diffuse irradiances.

In their work, García et al. (2014) showed that for the particular conditions of the Izaña Observatory, a high-elevation arid site located in the Canary Islands, the parameters «a» and «b» from Equations 4 and 5 could be expressed as functions of the daily AOD at 500 nm. This possibly reduces the method's versatility as it requires collocated aerosol optical properties. However, in this study, thanks to the availability of coincident photometric measurements, this method was selected as it gives more satisfactory results than the original algorithm as shown in Section 2.2.2.

In the present study, as described in (Elias et al., 2024), the variability of the parameters "$a_{GHI}$", "$b_{DHI}$" and "$b_{GHI}$" with the AOD at 500 nm was adapted in order to better represent the conditions observed in Lille as they are quite different from those observed at the Izaña Observatory.

Finally, for the sake of simplification, in what follows, the method of García et al. (2014) will be labelled simply as «clear-sky» (CSKY) and the method of Batlles et al. (2000) as «clear-sun» (CSUN). Furthermore, situations qualified as CSUN but not CSKY will be referred to as CSWC (clear-sun with clouds) as they should correspond to moments when the sky is (partially) cloudy without clouds in the Sun aureole.

### 2.2.2 Performances

To validate the performance of the cloud-screening methods, a comparison was made with ground observations of the sky conditions based on manual inspection of sky images at a 3-minute time resolution for the months of January and May 2018. These images were individually analyzed to determine the presence or absence of clouds in the sky and in the vicinity of the Sun. In the latter case, the assessment of clouds in the direction of the Sun was purposefully strict in order to account for the rather large field of view of the pyrheliometer (5°), which is larger than the Sun's viewing angle (0.5°). The potential motion of clouds in or out of the pyrheliometer's field of view was also considered to account for any time delay between the irradiance measurements and the sky images. Since the output of both manual and irradiance-based estimates is binary, i.e. clear or cloudy, there are only four possible outcomes. This allows the definition of a confusion matrix as follows :

  – True Positive (TP), the sky (Sun) is clear in both manual and filter-based estimates,

  – True Negative (TN), the sky (Sun) correctly identified as cloudy,





– False Positive (FP), the sky (Sun) is identified as clear by the cloud-screening method while it is manually assessed as
       cloudy,

       – False Negative (FN), the sky (Sun) is wrongly assessed as cloudy by the algorithm.

The results are presented in Table 1 for both clear-sun and clear-sky conditions simultaneously. For the clear-sky detection,
the revised method of García et al. (2014) gives satisfying results with a very low FP score (2.7%) compared to the TP score
(12.5%) which leads to an overall good precision of 82%. The latter score highlights that 82% of the time the sky is correctly
identified as clear by the algorithm. The risk is also quite low as the identification of the sky condition only fails 5% of the
time. For the identification of clear-sun conditions the results are also satisfactory although the method of Batlles et al. (2000)
is less successful with a relatively high FP score (10.9%) compared to the TP score (21.8%). The precision is thus lower (67%)
and the risk higher (12%) which might lead to a slight overestimation of the proportion of clear-sun situations.

It should be noted that both methods perform quite well under almost any aerosol load. Both algorithms properly transcribe
the global probability density function (PDF) of $AOD_{440}$ over 2010 - 2022 as from the unfiltered AERONET observations.
Comparison was made with the initial algorithm of Long and Ackerman (2000). The overall precision of this method is higher
(90%) in link with a much lower FP score (1.0%). However, the corresponding AOD probability density function suggests that
this algorithm is less representative of the variability of the aerosol content in Lille, as it tends to misidentify clear skies for
AOD values greater than 0.3.

     Finally, irradiance measurements considered in this study were limited to times between sunrise + 30 minutes and sunset -
30 minutes in order to reduce the bias observed in Long and Ackerman (2000) for large solar zenith angles as such situations
tend to be systematically classified as cloudy by the algorithm. This limitation was chosen to eliminate as many measurements
in winter as in summer, which would not have been the case for a criterion based on the solar zenith angle.

| | | TP | TN | FN | FP | Precision | Risk |
|---|---|---|---|---|---|---|---|
| | Definition | Filter → Clear Obs → Clear | Filter → Cloudy Obs → Cloudy | Filter → Cloudy Obs → Clear | Filter → Clear Obs → Cloudy | $\frac{TP}{TP+FP}$ | $\frac{FP+FN}{FP+FN+TP+TN}$ |
| Clear-sky (Garcia) | Nb of cases | 1549 (12.5%) | 10186 (82.1%) | 328 (2.6%) | 340 (2.7%) | 82% | 5% |
| Clear-sun (Batlles) | (% of cases) | 2702 (21.8%) | 8247 (66.5%) | 98 (0.8%) | 1356 (10.9%) | 67% | 12% |

**Table 1.** Confusion matrix for the estimation of the sky condition between irradiance based filters and manual ground observations from sky images for January and May 2018 in Lille.



## 2.3 Radiative transfer simulations

This section describes the radiative transfer code and the related aerosol optical properties models used in this study. A validation, over the period 2010-2022, of SOLARTDECO radiative transfer simulations in clear-sky conditions is presented in Section 2.3.3 to ensure their good transcription of the direct radiative effect of aerosols in Lille.

### 2.3.1 Description of the SOLARTDECO code

The radiative transfer code SOLARTDECO is a «solar» version of ARTDECO (Atmospheric Radiative Transfer Database for Earth Climate Observation) (Dubuisson et al., 2016), a toolbox comprising data of the properties of atmospheric components as well as several radiative transfer models intended for the simulation of atmospheric radiances and radiative fluxes. The scientific numerical core is written in Fortran90 while the configuration files and libraries are defined through python and ASCII formats. It is thus flexible and highly portable hence its selection for this study. Initially, ARTDECO is designed for radiative transfer simulations over the whole ultraviolet (UV) to thermal infrared (IR) range. It works for specific channels (monochromatic mode) as well as for more general spectra through the use of k-distributions (Lacis and Oinas, 1991).

SOLARTDECO is dedicated to the study of the incoming surface solar radiation in clear-sky conditions. The spectral resolution of the simulations can be defined using two different k-distributions initially implemented in ARTDECO. The first one was defined by Kato et al. (1999) and has a relatively low resolution with only 32 spectral bands spanning the solar range (0.24 - 4.0 $\mu$m). It is well suited for the computation of integrated irradiances however its depiction of the spectral variability is somewhat limited. Therefore another k-distribution, with more than 200 spectral bands, is also available for refined spectral simulations (Dubuisson et al., 2004, 2005). These k-distributions are coupled with the gas concentration vertical profiles from Anderson et al. (1986) for a mid-latitude summer to represent the gaseous absorption of the atmosphere. The profiles of $O_3$ and $H_2O$ are scaled for each simulation with respect to the total ozone (Dobson units) and precipitable water vapor (cm) contents available in the AERONET datasets. Note that SOLARTDECO also accounts for the absorption of $CO_2$ and $O_2$ with homogeneous concentrations of respectively 407 and 209 500 ppmv. In what follows and unless stated otherwise, the k-distribution from Kato et al. (1999) is used.

The remaining inputs include the solar and viewing zenith and azimuth angles as well as $AOD_{440}$ and $AOD_{550}$, the $AE_{440-870}$ and the relative humidity at the ground level. This allows the determination of the geometry of the simulations and corresponding aerosol optical properties. For the present study, new models of aerosol optical properties were defined, for all k-distribution bands, using the available AERONET inversions of the aerosol volume size distribution and complex refractive index in Lille over the period 2010 - 2020. For each radiative transfer simulation a particular model is chosen and its properties adjusted relative to the inputs of $AOD_{440}$, $AOD_{550}$, $AE_{440-870}$ and RH as described in the following Section 2.3.2.

SOLARTDECO also accounts for reflections at the surface which is defined as lambertian with a spectrally homogeneous albedo arbitrarily set at 0.15 in the present study. The aforementioned properties of the atmosphere (gaseous absorption, aerosol extinction and surface reflectivity) then allow for computations of the incoming and outgoing spectral irradiances through the use of the radiative transfer model DISORT (Tsay et al., 2000). Finally, although the present study focuses on the analysis of the





horizontal irradiances, SOLARTDECO can also compute radiance fields. Hence, it is possible to simulate the solar irradiances over titled surfaces for any orientation. This is of particular interest for solar related technologies which are usually placed on
sun-tracking devices or present specific tilts and orientations to maximize their output.

### 2.3.2 Definition of the models of aerosol optical properties

We define new models of aerosol optical properties based on 6 classes of $AOD_{440}$ and $AE_{440-870}$ measured in Lille. This classification, inspired by the work of Toledano et al. (2007), represents a rough categorization of the aerosols properties in Lille. The aim of this classification is twofold. On the one hand, it enables the definition of the new models of aerosol
optical properties suitable for typical aerosol conditions encountered in Northern France. On the other hand, it also serves a climatological purpose as it facilitates the characterization of the nature and variability of the aerosol content and type present in the Lille area over the period 2010 - 2022. The 6 classes are defined as follows :

- Continental - mostly fine (anthropogenic) particles ($AE_{440-870} \geq 1$) with low to medium aerosol loads ($AOD_{440} < 0.21$),

- Continental polluted - same as continental with stronger AOD values ($0.21 \leq AOD_{440} < 0.85$),

- Maritime - mostly coarse particles ($AE_{440-870} < 1$) with low to medium aerosol loads ($AOD_{440} < 0.19$),

- Desert dust - mainly coarse particles with higher $AOD_{440}$ values than maritime aerosols ($AOD_{440} > 0.12$),

- Strong events - events characterized by very high AOD values ($AOD_{440} \geq 0.85$) and dominated by fine particles,

- Mixed - situations with important contributions of both fine and coarse particles.

These site-specific thresholds are summarized in Table 2a alongside the respective proportions of each aerosol class measured
in Lille over the period 2010 - 2022. Figure 2 which represents the scatter plot of $AOD_{440}$ against $AE_{440-870}$ for all AERONET level 2.0 measurements acquired in Lille from 2010 to 2022 illustrates this classification. An independent arbitrary threshold of 0.1 in $AOD_{440}$ was also defined to distinguish «clean» and «polluted» situations in terms of aerosol optical depth (Table 2b).

Lille appears to be a fairly polluted site as only 27% of $AOD_{440}$ show values lower than 0.1. The main contribution is from the Continental class (38%), followed by Mixed cases (22%) and Continental polluted events (20%). Hence, aerosols observed
in Lille are mostly related to anthropogenic activities as fine particles prevail at least 58% of the time. However, a non negligible contribution of coarse particles is observed as the Maritime and Desert dust classes represent 14% and 5% of the observations, respectively. Note that the proportion of desert dust influence obtained from our aerosol classification in clear-sun conditions in Lille is consistent with that previously provided by Putaud et al. (2010), based on particulate matter measurements in all-sky conditions, as these authors estimate a mean proportion of desert dust particles in the range of 5 to 12% in the North of Europe.
For the definition of the new models of aerosol properties, these 6 classes were used to split AERONET almucantar inversions into several datasets. Because AERONET inversions are limited in terms of air mass and AOD (Dubovik et al., 2002), level 2.0 data was used for classes which present AOD values greater than 0.4 (i.e. Continental polluted, Desert dust, Strong events and Mixed) while level 1.5 inversions were used otherwise, provided that apart from the AOD threshold, all the other criteria were





| Class | $AOD_{440}$ | $AE_{440-870}$ | Proportion (%) |
|---|---|---|---|
| Continental | $[0, 0.10[$ | $[1.00, 1.34[$ | 38 |
| | $[0, 0.21[$ | $[1.34, 2.10[$ | |
| Continental polluted | $[0.21, 0.85[$ | $[1.34, 2.30[$ | 20 |
| Maritime | $[0, 0.19[$ | $[-0.04, 0.17[$ | 14 |
| | $[0, 0.12[$ | $[0.17, 0.82[$ | |
| | $[0, 0.1[$ | $[0.82, 1.00[$ | |
| Desert dust | $[0.12, 2.4[$ | $[0.17, 0.82[$ | 5 |
| Mixed | $[0.10, 0.85[$ | $[0.82, 1.34[$ | 22 |
| Strong event | $[0.85, 2.90[$ | $[0.82, 2.30[$ | 0.3 |

(a) Thresholds of aerosol classes

| | $AOD_{440}$ | Proportion (%) |
|---|---|---|
| Clean | $[0, 0.1]$ | 27 |
| Polluted | $]0.1, 2.90]$ | 73 |
| Nb of points | 95 923 | |

(b) Clean and polluted thresholds

**Table 2.** (a) Selected $AOD_{440}$ and $AE_{440-870}$ thresholds for each aerosol class defined in Lille. (b) Additional thresholds used to distinguish photometric observations in Lille between relatively clean and polluted conditions based on $AOD_{440}$ measurements. The last column of each table represents the respective proportion of each aerosol class measured in Lille over the period 2010-2022.

met. The resulting dataset is first divided according to the six aerosol classes defined above, with 10 bins of surface relative
humidity for each class ranging from 0 to 100%, hence 60 different datasets of AERONET inversions of complex refractive
index and size distributions were defined in this study. As the size distribution provided by AERONET is bi-modal, we chose
to separate each data set into two distinct modes, one associated with fine particles (fine mode), the other with larger particles
(coarse mode). Using a Mie code we then defined a look-up table (LUT) of 120 datasets of mean aerosol optical properties (60
for each mode) which allows us to compute specific aerosol optical properties for each simulation through the following steps
:

– First, the appropriate pair of modes is selected using the measured values of $AOD_{440}$, $AE_{440-870}$ and relative humidity.



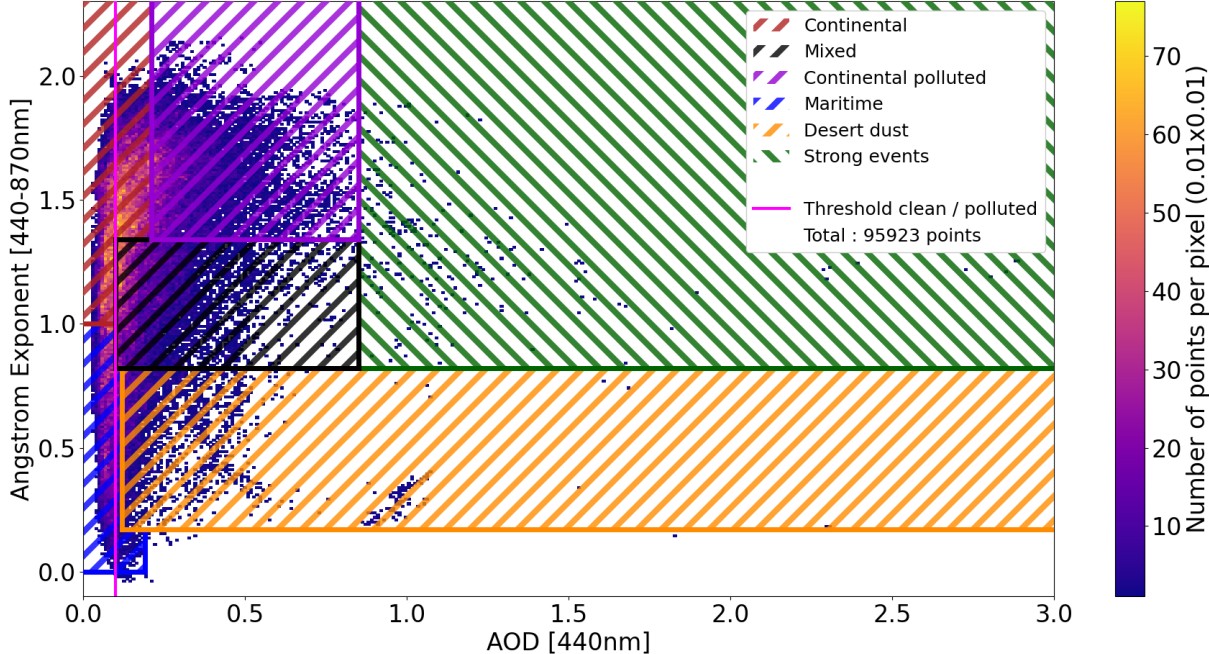

**Figure 2.** Scatter plot of $AOD_{440}$ against $AE_{440-870}$ for all AERONET level 2.0 measurements in Lille from 2010 to 2022. Colored boxes represent the class thresholds from Table 2a. The overall proportions of each class, and of «clean» ($AOD_{440} \leq 0.1$) and «polluted» ($AOD_{440} > 0.1$) measurements are reported in Table 2b.

– Secondly, a system of two equations with two unknowns based on the measured and simulated AOD and AE values is solved to compute the fine ($ff$) and coarse ($cf$) mode fractions (Eqs. 6 and 7) :

$$AOD_{440,tot} = ff \times AOD_{440,fine} + cf \times AOD_{440,coarse} \tag{6}$$

$$AE_{tot} = ff \times AE_{fine} + cf \times AE_{coarse} \tag{7}$$

with $AOD_{440,fine}$, $AOD_{440,coarse}$, $AE_{fine}$ and $AE_{coarse}$ the respective AOD values at 440nm and AE between 440 and 870 nm of the fine and coarse modes. To determine these quantities, we assume that the observed aerosol layer corresponds to a mixture of two distinct observations, one characterized exclusively of fine mode aerosols and the other one only of coarse mode aerosols. Following this hypothesis, the measured aerosol optical depth at 550 nm ($AOD_{550}$) corresponds to the AOD at 550 nm of each mode. It is thus possible to determine the AOD of each mode (either fine or coarse) at 440 nm from the measurement at 550 nm, using the extinction coefficients ($C_{ext,\lambda,mode}$) simulated by the Mie code at the wavelength $\lambda$ (here, 440 and 550 nm), as in the following equation :

$$\begin{aligned} AOD_{440,mode} &= AOD_{550} \times \left(\frac{440}{550}\right)^{-AE_{[440,550],mode}} \\ &= AOD_{550} \times \frac{C_{ext,440,mode}}{C_{ext,550,mode}} \end{aligned} \tag{8}$$




Note that the Angstrom coefficients between 440 and 870 nm are computed for each mode directly from the associated extinction coefficients, using Equation 9 :

$$\text{AE}_{mode} = -\frac{log(\frac{C_{ext,440,mode}}{C_{ext,870,mode}})}{log(\frac{440}{870})} \tag{9}$$

– The computation of the fractions *ff* and *cf* allows us to estimate new mixing ratios, $R_{fine}$ and $R_{coarse}$, using the number concentrations $NC_{fine}$ and $NC_{coarse}$ derived from AERONET inversions, through Equations 10 and 11 :

$$R_{fine} = \frac{ff \times NC_{fine}}{(ff \times NC_{fine} + cf \times NC_{coarse})} \tag{10}$$

$$R_{coarse} = \frac{cf \times NC_{coarse}}{(ff \times NC_{fine} + cf \times NC_{coarse})} \tag{11}$$

– Finally, these mixing ratios enable the computation of the new model of aerosol optical properties specific to the simulation by combination of the properties of each mode at each wavelengths while avoiding further Mie calculations, which can be quite time consuming, using Equations 12 to 15:

$$Cext_{\lambda,new} = Cext_{\lambda,fine} \times R_{fine} + Cext_{\lambda,coarse} \times R_{coarse} \tag{12}$$

$$Csca_{\lambda,new} = Csca_{\lambda,fine} \times R_{fine} + Csca_{\lambda,coarse} \times R_{coarse} \tag{13}$$

$$SSA_{\lambda,new} = \frac{Csca_{\lambda,new}}{Cext_{\lambda,new}} \tag{14}$$

$$P_{\lambda,new} = \frac{P_{\lambda,fine} \times Csca_{\lambda,fine} \times R_{fine}}{Csca_{\lambda,new}}$$
$$+ \frac{P_{\lambda,coarse} \times Csca_{\lambda,coarse} \times R_{coarse}}{Csca_{\lambda,new}} \tag{15}$$

where $Csca_{\lambda,mode} = Cext_{\lambda,mode} \times SSA_{\lambda,mode}$ is the scattering coefficient, $SSA_{\lambda,mode}$ the single scattering albedo and *P* represents the components of the phase function ($P_{11}$, $P_{21}$, $P_{34}$ and $P_{44}$). The subscript «new» stands for the new optical properties computed from the combination of the fine and coarse modes.

### 2.3.3   Validation of SOLARTDECO irradiance simulations over 2010 - 2022

Figures 3a-c represent the scatter plots between simulations and measurements of (a) GHI, (b) BHI and (c) DHI over the period
2010-2022 in Lille for all AERONET observations performed between sunrise plus 30 minutes and sunset minus 30 minutes in clear-sky situations over the period 2010-2022 (44 239 comparisons). The identification of clear-sky situations is issued from the algorithm presented in Section 2.2. Simulated and measured mean flux values and associated standard deviations are plotted on the axes. Comparison statistics such as the mean bias (MBD), mean absolute bias (MAD), root mean square error (RMSD) are also shown on each figure in a beige box.

Overall, the performances of SOLARTDECO simulations are satisfactory for all irradiance components with RMSD values lower than 10% for the DHI (8.06%) and 5% for the BHI (3.18%) and GHI (2.66%), which are comparable to the results





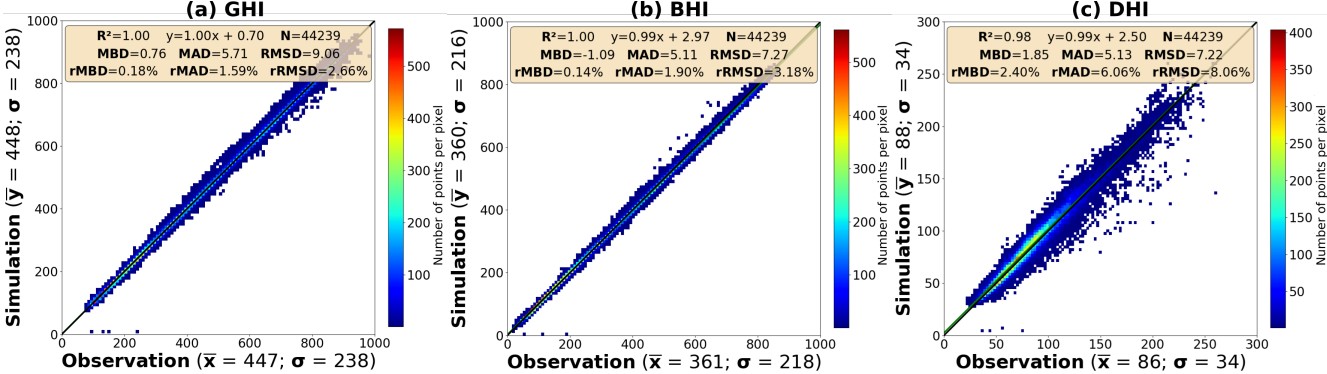

**Figure 3.** Scatter plot of SOLARTDECO simulations, based on level 2.0 AERONET inputs, against corresponding measurements of (a) GHI, (b) BHI and (c) DHI performed in Lille over the period 2010-2022. The black line represents the 1:1 line. Mean irradiance values and associated standard deviations are shown for both simulations and observations on their respective axes. Absolute and relative values of MBD (Mean Bias Difference), MAD (Mean Absolute difference) and RMSD (Root Mean Square Difference) are also displayed in the beige box included in the different figures. Only observations that coincide with clear-sky flux measurements (identified by the algorithm presented in Section 2.2) and performed between sunrise plus 30 minutes and sunset minus 30 minutes are considered for these comparisons.

of the best performing clear-sky irradiance models assessed in the latest studies of Sun et al. (2019) for the GHI and Sun et al. (2021) for the BHI and DHI. Note however that Sun et al. (2019, 2021) validations were obtained using reanalysis atmospheric data from MERRA2 (instead of AERONET here) and based on a larger number of worldwide stations of ground-

based irradiance observations. Moreover, the performances of SOLARTDECO are well within the error margins expected for network-operational instruments (Meteorological Organization, 2008) as the mean absolutes biases are lower than 3 W/m$^2$ for all irradiance components and more than 95% of the comparisons have an absolute difference lower than $\pm$20 W/m$^2$.

A focus on the years 2018 and 2019 shows that the performances of SOLARTDECO are also comparable to those of the radiative transfer tool SOLARES under clear-sky conditions (identified using the same filter as in the present study) in Lille

over the same period (Elias et al., 2024), which shows relative mean biases in GHI/BHI/DHI of -0.08/-0.41/+3.34% (compared with 0.18/0.14/2.40% for SOLARTDECO) and relative RMSD values of 1.83/2.29/8.01%.

SOLARTDECO appears to slightly overestimate the DHI. This discrepancy could be linked to the circumsolar contribution, i.e. the diffuse part of the incident solar irradiance located in the vicinity of the Sun direction (0.6° to 8°). Indeed, the circumsolar contribution is measured by the instruments as part of the direct irradiance incident at the surface, while it is generally

accounted for as part of the diffuse irradiance in radiative transfer simulations. This difference between simulations and ground measurements, which represents at most 2 to 3% of the DNI in the majority of cloud-free events (Gueymard, 2001, 2010; Blanc et al., 2014), usually leads to an underestimation of the BHI and overestimation of the DHI in radiative transfer simulations. This is in accordance with the overestimation of the diffuse irradiance by SOLARTDECO. However, no underestimation of the BHI is reflected by the mean biases, which could indicate the existence of other biases with opposite effects.

Most comparisons (94%) of the GHI show relative biases of less than 5% in absolute terms, in line with the relatively low



RMSD values presented in Figures 3a-c. Thus, overall, we can consider that SOLARTDECO simulations are satisfactory and that the aerosol optical property models used for these simulations effectively capture the optical properties of aerosols observed in Lille. Therefore, in the following sections, we have chosen to utilize estimates of optical properties derived from SOLARTDECO, particularly of the Single Scattering Albedo (SSA), asymmetry parameter (g), and fine fraction (ff), to supplement the analysis of solar radiation variability presented in Section 3.

## 3 Analysis of the surface solar irradiance's variability over 2020-2022

In this section, the results of the joint analysis of the simultaneous measurements conducted in Lille over the period 2010-2022 are presented. This analysis relies on both classifications of the sky conditions and aerosol optical properties introduced in Section 2, as well as on the radiative transfer simulations (in clear-sky conditions) of SOLARTDECO. The objective is to obtain a climatology of the solar environment in Lille over the past decade (Section 3.1) and examine the influence of clouds, aerosols, and gases on the variability of surface radiation in northern France (Section 3.2).

Firstly, in Section 3.1.1, the seasonal variability of solar irradiance, aerosol optical properties, and sky conditions is examined. Then we focus in Section 3.1.2 on their inter-annual variability in spring and summer over the period 2010-2022. In section 3.2, we develop a framework that allows to make the distinction between the different variabilities of the scenes, atmospheric components and geometries, that contribute to the variabilities and trends recorded for surface solar irradiance.

### 3.1 Recorded variabilities of sky conditions, aerosols and SSI

#### 3.1.1 Seasonal cycle

Figures 4a-h present the average monthly variations during daytime over the 13-year period 2010-2022 of different measured and derived atmospheric quantities associated to the solar environment in Lille: (Figure 4a) sky conditions identified with the irradiance-based cloud-screening methods described in Section 2.2, (Figures 4b-d) aerosol properties from AERONET measurements, and (Figures 4e-h) irradiances measured for the different sky conditions.

The results of our cloud-screening approach confirm the significant influence of clouds in Lille observed by Warren et al. (2007). Over the 13-year period considered, we find that, on average, the Sun is obscured by clouds (cloudy-sun situations) in nearly 67% of cases, while completely clear skies are observed only 11% of the time, and intermediate conditions of partly cloudy skies with a visible Sun (clear-sun with clouds conditions) represent only an average of 22% of observed situations. Thus, the sun is clear of clouds 33% of the time (clear-sun conditions). Note that cloudy-sun situations also encompass instances characterized by thin clouds or clouds partially covering the Sun (cloud edges), which have a limited impact on incident solar radiation, with Direct Normal Irradiance (DNI) values exceeding 120 W/m$^2$ (sunshine criterion defined by the World Meteorological Organization [WMO (2003)]). On average over the period 2010-2022, these situations, which can be considered as "sunny", represent 9% of observed conditions, and account for about 13% of "cloudy-sun" situations. The maximum (resp. minimum) of clear-sky (resp. cloudy-sun) occurrence happens in March-April. Clear-sky conditions are minimum in winter,



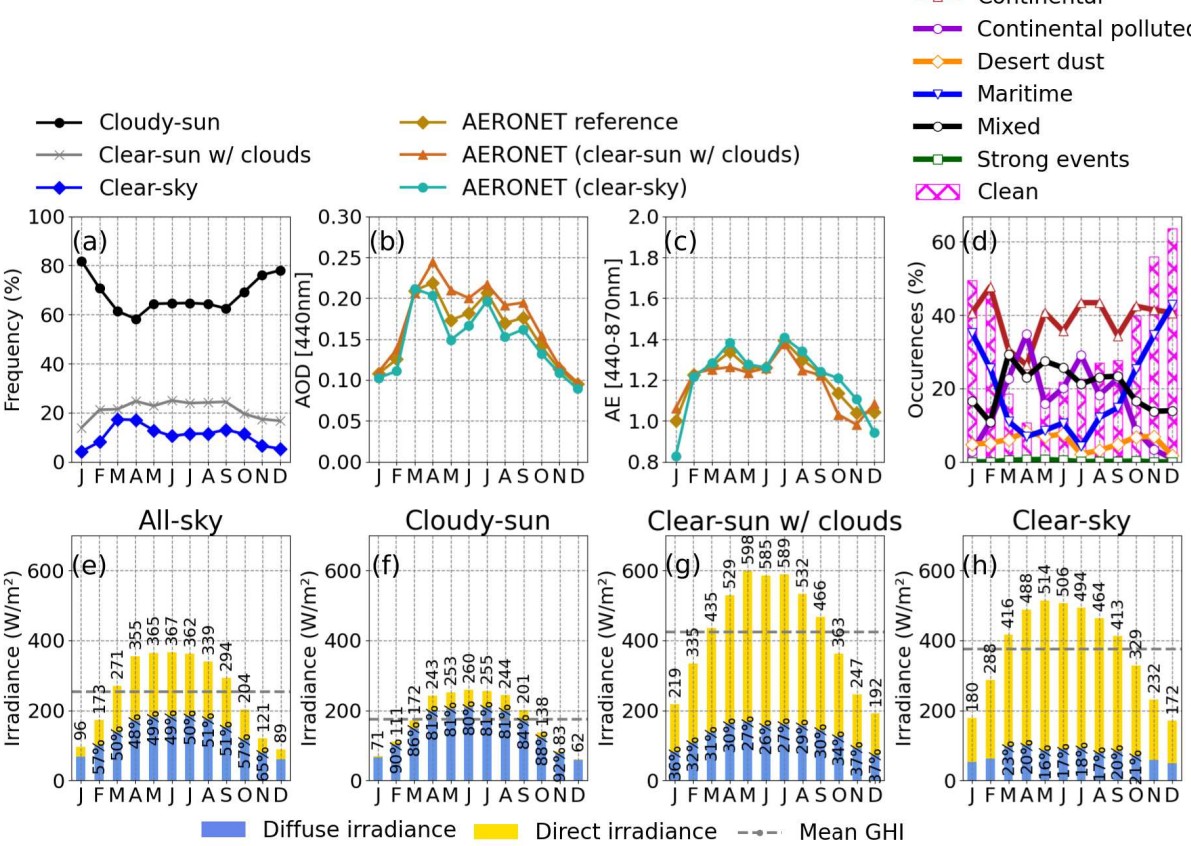

**Figure 4.** Monthly variations in Lille over the period 2010-2022 of several quantities derived from ATOLL measurements. (a) Frequencies of cloudy-sun (black line), clear-sun with clouds (grey line) and clear-sky moments (blue line) in percentages. (b) Monthly mean values of $AOD_{440}$ overall (brown line) and for different sky conditions: clear-sun with clouds (orange line) and clear-sky (blue line). (c) Same as (b) but for the $AE_{440-870}$. (d) Proportions of the different classes presented in Table 2a: continental (red line), continental polluted (purple line), desert dust (orange line), maritime (blue line), mixed (black line) and strong events (green line). The proportion of clean situations ($AOD_{440} \leq 0.1$) for all AERONET coincident AERONET measurements is also represented as pink columns. (e-h) Mean monthly irradiances for different cloud cover states: (e) all-sky, (f) cloudy-sun, (g) clear-sun with clouds and (h) clear-sky. Only values during daytime between [sunrise + 30 minutes; sunset - 30 minutes] are considered. The GHI is represented as columns with the lower blue part corresponding to the DHI and the upper yellow part to the BHI. The grey dashed lines represent the mean GHI over 2010-2022. Note that the percentages reflect the contribution of the DHI to the overall mean yearly GHI which values are reported above each column.





with average occurrences of only around 5%, associated with a maximum of cloudy-sun conditions of around 80%. A relatively stable plateau of minimum (resp. maximum) cloudy-sun (resp. clear-sun) monthly proportions, with values consistently lower (resp. greater) than 65% (11 and 24% for clear-sky and clear-sun with clouds conditions), is also noticeable from March to September, i.e. spring and summer.

Figures 4b-d show that the period from March to September is also significantly influenced by aerosols, with monthly mean $AOD_{440}$ values greater than 0.15 (Figure 4b). The stability of the associated Angström exponents (values greater than 1.2, Figure 4c) suggest a strong influence of fine particles, primarily from anthropogenic emissions, reaching a maximum in April (monthly mean values of $AOD_{440}$ exceeding 0.2), with an minimum proportion of AOD measurements below 0.1 (clean conditions) of only 11% (Figure 4d). It can be noted that spring is characterized by a significant increase in agricultural activity, including fertilizer spreading in open fields, which releases substantial amounts of ammonia into the atmosphere. Moreover, the predominantly northeast wind flow observed in spring in Lille (Figure S2b), may indicate a higher frequency of anticyclonic conditions during this season. Overall, this could explain the overall lower frequency of cloudy moments and the accumulation of anthropogenic particles originating partly from the Benelux region. The accumulation of anthropogenic particles is consistent with the strong influence of the "Continental Polluted" aerosol class, reaching a maximum monthly average of about 35% around April (Figure 4d). Conversely, in winter, when cloudy conditions are more frequent (Figure 4a), the monthly mean values of AOD generally remain below 0.12, reaching a minimum of 0.09 in December (Figure 4b). This winter drop in AOD is associated with a decrease in the monthly mean AE (Figure 4c), which suggests an increased influence of coarse particles. The generally lower AOD and AE values result in a more significant impact of the "Maritime" aerosol class, with monthly proportions exceeding 35% for winter months (Figure 4d). This finding aligns with surface wind direction and speed measurements from the ATOLL platform, which highlight a prevailing influence from the southwest (Figure S2a), particularly from the English Channel region.

Figures 4e-h represent the monthly variability of the incident global surface flux in Lille over the period 2010-2022, as well as the partitioning between direct and diffuse components for (e) all-sky (ASKY), (f) cloudy-sun (CLOS), (g) clear-sun with clouds (CSWC) and (h) clear-sky (CSKY) conditions. Note that only measurements performed between [sunrise + 30 minutes; sunset - 30 minutes] are considered, as the identification of the sky state performs better during this period.

Overall, the climatological yearly mean value of global irradiance (GHI) is minimal in cloudy-sun conditions (192 W/m$^2$), when the Sun is obscured by clouds. In comparison, the measured surface flux in clear-sky conditions is twice as high, with an average value around 435 W/m$^2$. The maximum irradiance is observed for clear-sun with clouds conditions, which, due to the cloud side effects on the diffuse component, show a mean value around 484 W/m$^2$ (11% more than clear-sky). Thus, in conjunction with the frequency of each sky condition (Figure 4a), the average solar radiation incident in Lille under all-sky conditions is approximately 285 W/m$^2$. In addition to the discrepancies in GHI, it is worth mentioning that each sky condition is associated with varying proportions of direct and diffuse irradiance. In particular, the yearly mean diffuse ratio observed in Lille over the period 2010-2022 vary between a minimum of 25% in clear-sky conditions and a maximum of 93% in cloudy-sun situations, with intermediate values of 34 and 72% in clear-sun with clouds and all-sky conditions. This broad range of



values highlights the significant influence on the measured surface global irradiance in Lille of the presence of clouds and their position relative to the Sun and the observer.

Figures 4e-h show that the seasonal variations of the mean measured irradiances for all sky states display almost symmetrical inverted U-shapes over the year, that are linked to changes in solar zenith angle (Figures S1a-d) and thus air mass, which are greater in winter (minimum GHI) than in summer (maximum GHI). Moreover, the variability of the air mass has a great influence on the proportion of diffuse irradiance, which varies, under all-sky conditions, between 69% in summer to more than 85% in winter. Note that in all-sky conditions, the influence of the air mass is enhanced by changes in occurrence of the sky conditions as the frequency of cloudy-sun conditions (Figure 4a) is greater (resp. lower) in winter (resp. summer), leading to overall lower (resp. greater) mean ASKY GHI values. The influence of clouds is particularly important under clear-sun with clouds conditions, as clouds, through 3D effects, enhance the amount of radiation reaching the surface. Most notably, our results highlight that, due to the additional contribution of clouds, the monthly averages of global flux are, throughout the year, consistently higher for clear-sun with clouds conditions (Figure 4g) than for clear-sky situations (Figure 4h), with relative differences varying between 4% in March and 21% in January, and a absolute maximum difference of 95 W/m$^2$ in July. It is worth noting that in CSWC conditions, the gain from clouds results in diffuse flux values around 170 W/m$^2$ in summer, which are relatively comparable to those observed in CLOS conditions, reaching around 200 W/m$^2$ on average for the same season. In relative terms, the contribution of diffuse radiation is however significantly lower in CSWC conditions, with an average proportion of DHI around 30% in summer, compared to 80% in CLOS conditions. The presence of a cloud in the Sun's direction thus significantly reduces incident direct radiation without truly increasing diffuse radiation. The incident global surface radiation in CLOS conditions is thus much lower than in "clear-sun with clouds" situations.

To conclude, our joint analysis of the seasonal variability of the solar environment suggests that the period covering spring and summer is characterized by relatively high amounts of incident radiation and surface energy, linked to smaller solar zenith angles and longer day lengths. Both seasons are thus particularly interesting in terms of solar energy exploitation, as the period from March to September represents nearly 80% of the total energy accumulated over a year (approximately 1 MWh/m$^2$, Figure S1a). Hence, we chose to focus our study of the inter-annual variability of the solar environment in Lille over the period 2010-2022 (Section 3.1.2) on spring and summer seasons. Note that as these two seasons are generally characterized by a lower influence of clouds and relatively high levels of AOD, they are of great interest for the study of the impact of aerosols on the solar irradiance incident at the surface, as the impact of aerosols on the SSI and its direct/diffuse partition should be maximum in spring and summer under both clear-sun and all-sky conditions. A more thorough analysis of the direct radiative effect (DRE) of aerosols is presented in Section 4.

### 3.1.2 Year-to-year variability in spring and summer

Figures 5 and 6 represent the year-to-year evolution from 2010 to 2022 of daytime sky conditions (5a and 6a), aerosol properties obtained from AERONET measurements (5b-d, 6b-d), and surface solar irradiances measured for the different sky conditions (5e-h, 6e-h), in Lille in spring and summer respectively. Figures 5 and 6 show some local extrema of interest for these quantities,





such as the maximum of irradiance and clear-sky occurrence of spring 2020. They show also some interesting trends over the
past decade, part of which are statistically significant.

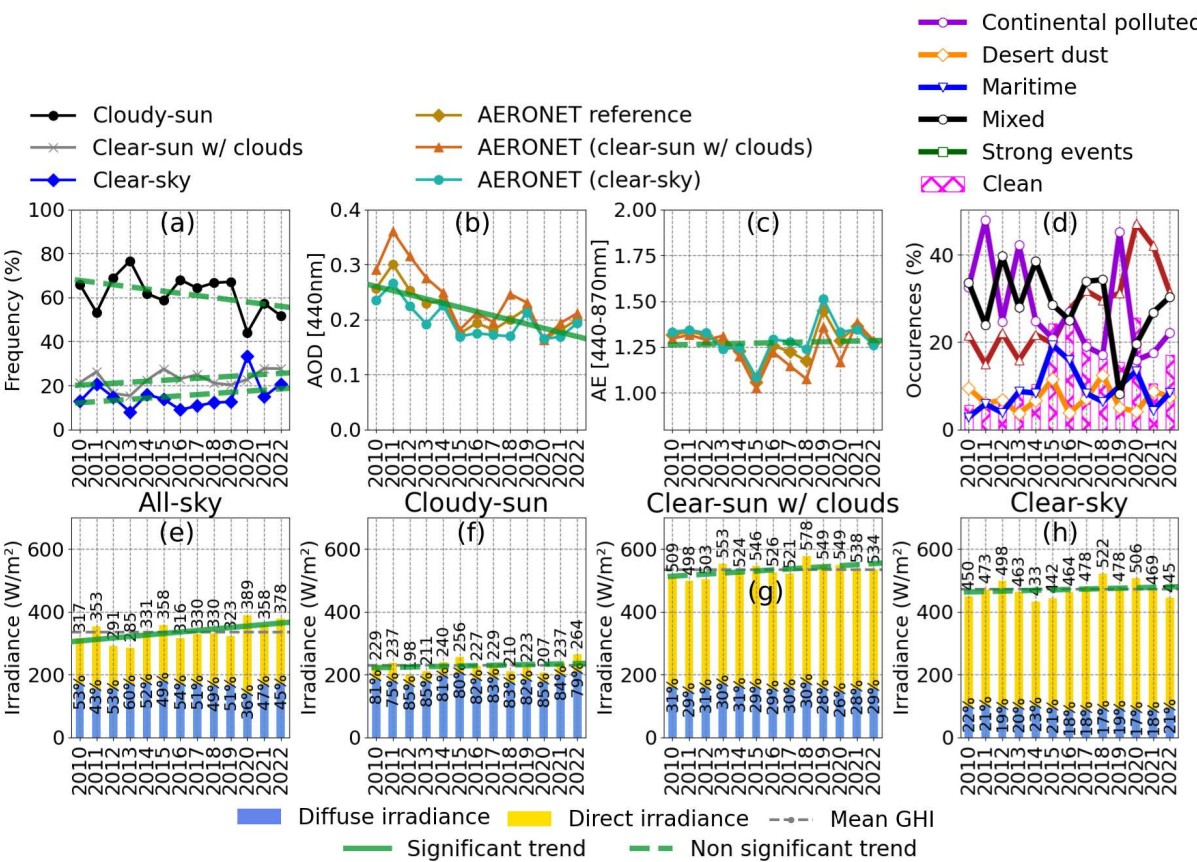

**Figure 5.** Same as Figure 4 but for yearly variations in spring between 2010 and 2022.

The 2010-2022 yearly averaged occurrence of clear-sky conditions represents 15% of the observations in spring, whereas in summer it is substantially lower, corresponding to only 11% of the measurements. Consistently, the Sun is on average obscured by clouds (cloudy-sun) 62% of the time in spring, and 65% of the time in summer. For both seasons, the year-to-year variability reported in Figures 5a and 6a suggest decreasing trends in the frequency of cloudy-sun conditions, more pronounced in summer
($-0.9 \pm 0.4$% per year) than in spring. In summer, our analysis suggests significant increasing trends in both clear-sun with clouds ($+0.5 \pm 0.2$% per year) and clear-sky ($+0.4 \pm 0.3$% per year) frequencies. As indicated in Table 3a, seasonal Mann-Kendall trend tests (Mann, 1945; Kendall, 1990), with a significance level of 5%, validate only the decrease in the frequency of cloudy-sun situations in summer and the opposite increase in clear-sun with clouds conditions. In spring, the trends in sky conditions, although consistent with those reported in summer, are not statistically significant (Table 3a), possibly due to
the higher year-to-year variability recorded for this season. Indeed, the proportion of clear-sky moments in spring oscillates





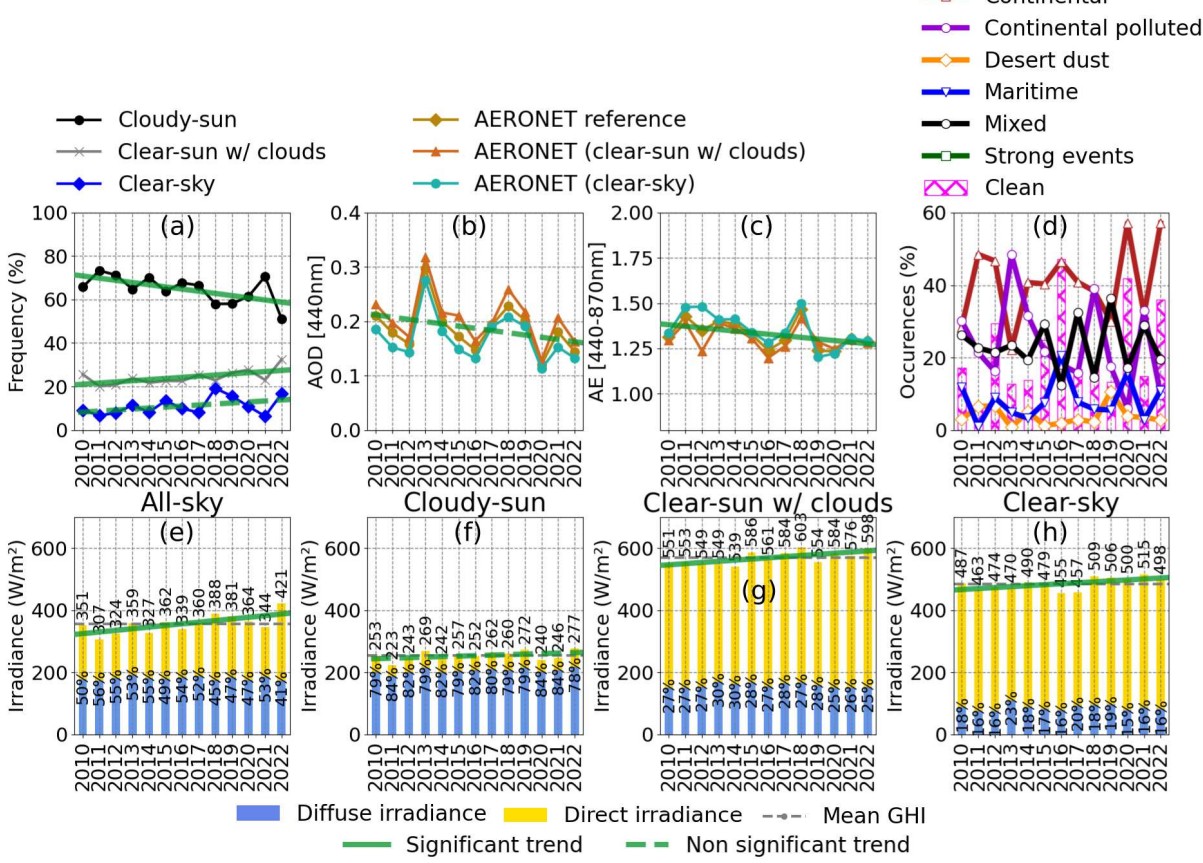

**Figure 6.** Same as Figure 5 but for summer.

between a minimum of around 8% in 2013 (compared to a minimum of 7% in summer 2011) and a maximum in 2020 of 35% (compared to a maximum of 20% in summer 2018), which can be considered as an extremely high value in Lille. Note that correspondingly, spring 2020 presents the lowest proportion of cloudy-sun moments with a mean frequency of 43%, compared to a minimum of 51% in summer 2022. An overall very low cloud fraction was also observed in Cabauw (Netherlands) in spring 2020 (Heerwaarden et al., 2021). This study showed, in particular, that the unique meteorological conditions of spring 2020, especially the low observed cloud fraction, led to record all-sky global irradiance measurements in Cabauw. These conditions were characterized by a relatively low contribution of diffuse radiation, accounting for 38%, as opposed to over 50% on average in spring in Cabauw. Similarly, in Lille, an average global irradiance under all-sky conditions was observed in spring of 2020 (389 W/m²), representing the maximum GHI over the whole period 2010-2022 for the spring season (Figure 5e). In particular, this remarkable value is 55 W/m² (16%) higher than the spring average observed over the period 2010-2022 (334 W/m²). Additionally, the relatively low contribution of the DHI (36%), similar to that measured in Cabauw, is notably lower than the usual seasonal averages observed in spring in Lille, ranging between 43 and 60% between 2010 and 2022. A detailed analysis




| | 2010-2022 mean (%) | |
| | [trend $\pm$ standard-deviation] (% per year) | |
| Proportion of | **Spring** | **Summer** |
|---|---|---|
| **Cloudy-sun** | 62%/[-0.98 $\pm$ 0.60] | 65%/[**-0.92 $\pm$ 0.40**] |
| **Clear-sun w/ clouds** | 23%/[0.42 $\pm$ 0.26] | 24%/[**0.48 $\pm$ 0.20**] |
| **Clear-sky** | 15%/[0.56 $\pm$ 0.49] | 11%/[0.44 $\pm$ 0.28] |

(a) Frequency of sky conditions

| | **Trend $\pm$ standard deviation** | |
| AOD [440nm] (unit per year) | **Spring** | **Summer** |
|---|---|---|
| **Clear-sun** | **(-0.008 $\pm$ 0.002)** | (-0.004 $\pm$ 0.003) |
| **Clear-sun with clouds** | **(-0.011 $\pm$ 0.003)** | (-0.004 $\pm$ 0.003) |
| **Clear-sky** | **(-0.006 $\pm$ 0.002)** | (-0.003 $\pm$ 0.003) |
| Frequency of occurrence (% per year) | | |
| **Continental** <br> **Clear-sun with clouds/Clear-sky** | **(+2.2 $\pm$ 0.6)/(+1.7 $\pm$ 0.5)** | (+1.2 $\pm$ 0.6)/(+0.4 $\pm$ 1.0) |
| **Continental polluted** <br> **Clear-sun with clouds/Clear-sky** | **(-1.8 $\pm$ 0.7)/(-0.9 $\pm$ 0.8)** | (-1.0 $\pm$ 0.6)/(-1.5 $\pm$ 0.9) |

(b) AOD and frequency of occurrence of continental and continental polluted aerosol classes

| | Trend in GHI/BHI/DHI (W/m$^2$/year) $\pm$ standard deviation | |
| | **Spring** | **Summer** |
|---|---|---|
| **All-sky** | **(3.95 $\pm$ 1.90)/(4.36 $\pm$ 2.30)**/(-0.41 $\pm$ 0.77) | **(4.21 $\pm$ 1.85)/(4.66 $\pm$ 1.89)**/(-0.45 $\pm$ 0.46) |
| **Cloudy-sun** | (0.27 $\pm$ 1.31)/(-0.45 $\pm$ 0.65)/(0.72 $\pm$ 0.79) | (0.63 $\pm$ 1.16)/(0.09 $\pm$ 0.61)/(0.54 $\pm$ 0.59) |
| **Clear-sun with clouds** | (3.34 $\pm$ 1.43)/**(3.70 $\pm$ 1.07)**/(-0.36 $\pm$ 0.61) | **(3.73 $\pm$ 1.19)/(3.84 $\pm$ 1.21)**/(-0.11 $\pm$ 0.55) |
| **Clear-sky** | (1.35 $\pm$ 1.99)/(2.27 $\pm$ 2.08)/**(-0.92 $\pm$ 0.29)** | **(3.08 $\pm$ 1.26)/(3.08 $\pm$ 1.33)**/(0.002 $\pm$ 0.76) |

(c) Surface solar irradiances

**Table 3.** Trends per year in frequency of sky conditions (a), aerosols (b), and SSI (c) recorded in Lille in spring and summer over the period 2010-2022. Only parameters with at least one statistically significant trend (Mann-Kendall test) in spring or summer are reported. Note that other parameters such as the solar zenith angle, precipitable water vapor content and Angström exponent were also investigated, however, no significant trends were observed.

of the contribution of the variability of the aerosol content and occurrences of sky conditions to the maximum solar radiation in the spring of 2020 is presented in Section 3.2.3.

Figures 5b-d and 6b-d represent the variations of the yearly-averaged aerosol content and properties in Lille over the period 2010-2022 in spring and summer respectively. For both seasons, strong fluctuations of AOD$_{440}$ are observed (Figures 5b and



6b) when considering all reference AERONET measurements (i.e. clear-sun measurements, black line) as well as only those coincident to clear-sun with clouds (grey line) and clear-sky (blue line) observations.

In spring, our measurements show a significant decreasing trend in $AOD_{440}$ over the period 2010-2022 (-0.008 $\pm$ 0.002 per year) from a yearly mean value of 0.26 in 2010 to around 0.20 in 2022. By comparison, the observed negative trend in summer (-0.004 $\pm$ 0.003 per year) appears to be much lower and statistically non-significant (Table 3b. This decrease in AOD recorded in Lille agrees with the analysis of Ningombam et al. (2019) who identified a generalized decreasing trend in AOD for several European AERONET stations since 1995.

The yearly mean Angström Exponent appears to be relatively stable for both seasons (Figures 5c and 6c) with overall mean values of 1.27 and 1.33 respectively in spring and summer. Interestingly, in spring, the stable AE and decrease in AOD translates in balancing trends in the occurrences of the Continental (+1.9 $\pm$ 0.4 % per year) and Continental polluted classes (-1.2 $\pm$ 0.7 % per year), while in summer no trends in the occurrences of these two aerosol classes is statistically significant (Figures 5d and 6d).

Figures 5e-h and 6e-h depict the annual variations, for the different sky conditions, of the solar irradiance (colored columns) incident at the surface in Lille, and the partition between direct flux (yellow component) and diffuse flux (blue component) over the period 2010-2022 in spring and summer, respectively.

Under all-sky conditions, significant fluctuations of both the global irradiance and its direct component are observed, while the contribution of the diffuse flux remains relatively stable for both seasons (Figures 5e and 6e). In summer, measurements show in particular a difference of approximately 115 W/m$^2$ between the minimum of 2011 (307 W/m$^2$) and the maximum of 2022 (421 W/m$^2$). In spring, a similar gap of around 105 W/m$^2$ is also identified between the recorded minimum of 2013 (285 W/m$^2$) and the maximum of 2020 (389 W/m$^2$) mentioned earlier. These significant variations, which appear to be strongly linked to the year-to-year variability of meteorological conditions, suggest an increasing trend in all-sky solar irradiance incident at the surface in Lille in both spring and summer. Seasonal Mann-Kendall trend tests support this hypothesis, as similar increasing trends are indeed observed for both seasons, with a magnitude of approximately +4 $\pm$ 2 W/m$^2$/year over the period 2010-2022 (Table 3c). As the observed trends in diffuse irradiance are relatively uncertain, the overall increase in global irradiance appears to be primarily linked to a rise of the direct component, which also displays significant trends in both spring (4.4 $\pm$ 2.3 W/m$^2$/year) and summer (4.7 $\pm$ 1.9 W/m$^2$/year).

Our findings are consistent with the conclusions of other studies (Boers et al., 2017; Mateos et al., 2014; Sanchez-Lorenzo et al., 2013), which also observed increasing trends in surface solar irradiance for various European measurement sites, both in the presence and absence of clouds. However, it should be noted that trends of increasing all-sky surface solar irradiance reported in the literature, which often cover longer periods (several decades) or earlier periods, are less pronounced than those highlighted in this study in Lille over the past 13 years. Apart from differences related to geographic position of the measurement sites and periods considered in each study, a possible explanation for these discrepancies is the time range used for our analysis. As mentioned in Section previously, our analysis is based exclusively on daytime measurements, performed between [sunrise + 30 minutes; sunset - 30 minutes]. In contrast, studies from the literature usually include nighttime measurements as they generally consider irradiance measurements over a 24-hour period. Since nighttime measurements are characterized by





an absence of incident solar radiation, the average over 24 hours is lower than when considering only daytime measurements. To put our results into perspective, we applied seasonal Mann-Kendall trend tests to the year-to-year means over the period 2010-2022 with the addition nighttime measurements in our datasets. In this configuration, an increasing trend in all-sky GHI

is statistically significant for all seasons (even in winter), with maximum magnitudes in summer ($2.8 \pm 1.1$ W/m$^2$/year) and spring ($2.3 \pm 1.0$ W/m$^2$/year). Thus, on an annual average, a trend of about 1.5 W/m$^2$/year is observed, which is more in line with trends reported in the literature.

In Lille, significant increasing trends of both global and direct irradiances are also observed under clear-sun with clouds (+$3.8 \pm 1.2$ W/m$^2$/year for both components) and clear-sky (+$3.1 \pm 1.3$ W/m$^2$/year for both components) situations in summer

(Table 3c). In spring, the trends are more uncertain, apart from a significant increase in BHI under clear-sun with clouds conditions of about +$3.7 \pm 1.1$ W/m$^2$/year. Our results also highlight a significant decreasing trend in diffuse irradiance for clear-sky situations of around -$0.9 \pm 0.3$ W/m$^2$/year.

The results of Heerwaarden et al. (2021), regarding spring 2020 in Cabauw, showed a significant influence of sky conditions on the variability of the measured all-sky global irradiance. It is thus very likely that the GHI maximum of spring 2020, as well

as the increasing trends in both spring and summer, observed in Lille under all-sky conditions are strongly associated with the decrease of the frequency of cloudy-sun conditions in favor of clear-sun with clouds and clear-sky situations, which exhibit higher radiation values, especially for the direct component of the flux. Under clear-sun with clouds and clear-sky conditions, the observed trends in irradiances could be partly related to the observed declining trends in AOD$_{440}$, especially the decrease of clear-sky DHI in spring. It could also be related to the evolution of water vapor content, ozone, or even simply the solar

zenith angle under clear-sun conditions, which is influenced by the frequency of occurrence of the sky conditions. However, an analysis of the inter-annual variability of these various atmospheric parameters (not shown) reveals no significant trends over the period 2010-2022, neither in spring nor summer, apart from the negative trend in springtime AOD mentioned previously.

To isolate the contribution of the different parameters (occurrences of sky conditions, AOD, SZA, PWV, etc.) to the observed trends under the different sky conditions, a more detailed analysis was conducted using irradiance measurements from ATOLL

and radiative transfer simulations from SOLARTDECO, based on the decomposition of observed irradiance according to the classifications of the sky conditions and aerosol content introduced earlier. As presented in Section 3.2, this methodology allows for the study of trends over the period 2010-2022 and the year-to-year variability. It is worth noting that it could also be applied at other temporal resolutions, including monthly and intra-daily scales. An analysis of the direct radiative effect (DRE) of aerosols is also presented in Section 4 with a distinction of situations among clear-sun conditions.

## 3.2 Multivariate analysis of the variability of the solar environment

In this section, we undertake a multivariate analysis of the variability of surface solar irradiance over the period 2010-2022. The objective is to disentangle the contributions of the change in sky's conditions (presence or absence of clouds, aerosol class and loading, atmospheric parameters) and geometrical conditions to the observed variabilities. Our analysis focuses on the specific trends and annual extrema observed over the period 2010-2022 in spring and summer, as these seasons are the most significant

in terms of surface solar energy, exhibiting robust trends and substantial yearly variations of SSI over the 13-year period. The



methodology is described in Section 3.2.1. Section 3.2.2 presents the sensitivity study of the SSI to input parameters that is used to perform the multivariate analysis of the year-to-year variability of the SSI, with a focus on the all-sky irradiance record of spring 2020 presented in Section 3.2.3, and the analysis of the observed irradiance temporal trends in spring and summer presented in Section 3.2.4.

### 3.2.1   Methodology

The seasonal mean all-sky solar irradiance can be written as a weighted sum of the observed irradiances per sky condition:

$$F_{ASKY} = \sum_k F_k \times freq_k \tag{16}$$

where $k$ represents the sky's condition (CLOS, CSWC or CSKY), $F_k$ corresponds to the associated measured seasonal mean irradiances (GHI, BHI or DHI), and $freq_k$ represents the yearly mean frequency of the «k-th» sky condition for the corre-
sponding season.

With this approach it is possible to isolate the contributions of the change in frequency of occurrence of sky conditions and of the change in irradiance under each per sky conditions to both inter-annual variability and trends of all-sky irradiances. Indeed, this decomposition allows the analysis of the observed differences between the year "y" and the overall mean over the period 2010-2022:

$$\Delta F_{ASKY,\, y} = \sum_k \Delta F_{k,\, y} \times freq_{k,\, y} + F_{k,\, y} \times \Delta freq_{k,\, y} \tag{17}$$

where $F_{k,\, y}$ and $freq_{k,\, y}$ represent the mean seasonal irradiances and frequencies of the various sky conditions for the year "y", and the terms $\Delta F_{k,\, y}$ and $\Delta freq_{k,\, y}$ correspond, respectively, to the observed differences in irradiance and frequency of sky conditions between the seasonal averages for the year "y" and over the whole period of interest. With this framework, the term $F_{k,\, y} \times \Delta freq_{k,\, y}$ represents the contribution of the change of the sky's frequency of occurrence to the variability
$\Delta F_{ASKY,\, y}$, while the term $\Delta F_{k,\, y} \times freq_{k,\, y}$ corresponds to the contribution of the change of irradiance observed under each sky conditions.

A similar approach can be used to decompose the observed trends under all-sky conditions over the period 2010-2022 following Equation 18:

$$\frac{dF_{ASKY}}{dt} = \sum_k \frac{dF_k}{dt} \times freq_k + F_k \times \frac{dfreq_k}{dt} \tag{18}$$

where F and $freq_k$ are the same as in Equation 17 while $\frac{dF}{dt}$ and $\frac{dfreq}{dt}$ correspond, respectively, to the slopes of the observed seasonal trends in irradiances and frequencies of the various sky conditions. Again, the terms $F_k \times \frac{dfreq_k}{dt}$ and $\frac{dF_k}{dt} \times freq_k$ represent the influence of the trends in the occurrences of the sky conditions and the trends in other atmospheric parameters, respectively.



This analysis of the different contributions to the observed irradiance variabilities can be extended by further decomposing the terms related to the variability of the irradiances observed under each sky conditions, that we note here CSKY, CSWC and CLOS. We describe here a second and third step, for which we decompose the irradiances variability, only in the case of clear sky and CSWC (i.e. CSUN) situations, by accounting for changes in the occurrence of the five main aerosol class and changes in solar irradiances per aerosol class. The decomposition of the BHI under clear-sun conditions (i.e., CSKY and CSWC) is detailed in the following paragraph.

As in the previous paragraph, the seasonal mean solar irradiance for CSKY and CSWC conditions can be written as a weighted sum of the observed mean irradiances per aerosol class:

$$F_{clear} = \sum_i F_i \times freq_i \tag{19}$$

where $F_{clear}$ represents the mean seasonal irradiances measured under clear-sky or clear-sun with clouds conditions, and the terms $freq_i$ and $F_i$ correspond to the mean frequencies and irradiances of the "i-th" aerosol class[1] (i.e., "Continental," "Continental polluted," "Mixed," "Maritime," "Desert dust") for the same conditions (CSKY or CSWC). Then, similarly to the approach used under all-sky conditions, it is possible to isolate the contribution of the change of aerosol class' frequency and the change of irradiance per aerosol class, both for the inter-annual variability (Equation 20) and trends (Equation 21) :

$$\Delta F_{clear,\, y} = \sum_i \Delta F_{i,\, y} \times freq_{i,\, y} + F_{i,\, y} \times \Delta freq_{i,\, y} \tag{20}$$

$$\frac{dF_{clear}}{dt} = \sum_i \frac{dF_i}{dt} \times freq_i + F_i \times \frac{dfreq_i}{dt} \tag{21}$$

where $\Delta F_{i,\, y}$ and $\Delta freq_{i,\, y}$ represent the observed differences in irradiances and frequencies of the different aerosol classes between the seasonal averages for the year "y" and over the period 2010-2022, and the terms $\frac{dF_i}{dt}$ and $\frac{dfreq_i}{dt}$ correspond to the slopes of the observed seasonal linear trends in irradiances and frequencies of the various aerosol classes. In this writing, $\Delta F_{i,\, y}$ and $\frac{dF_{clear}}{dt}$ results from the variabilities of every scene's parameter (AOD, SSA, SZA, PWV, etc.) to which the irradiances are sensitive. In order to isolate and identify the impacts of each parameter, it is necessary to further decompose these terms following Equations 22 and 23 :

$$\Delta F_{i,\, y} = \sum_x \left[\frac{\partial F_{clear}}{\partial x}\right]_{i,\, y} \times \Delta x_{i,\, y} \tag{22}$$

$$\frac{dF_i}{dt} = \sum_x \left[\frac{\partial F}{\partial x}\right]_i \times \frac{dx_i}{dt} \tag{23}$$

where $\left[\frac{\partial F}{\partial x}\right]$ corresponds to the seasonal sensitivity of the irradiances to the various parameters "x" (AOD, SZA, SSA, etc.), and the terms $\Delta x_{i,\, y}$ and $\frac{dx_i}{dt}$ represent the temporal variations (for a given season) of the considered parameter for the "i-th" aerosol class. Note that the comparisons between SOLARTDECO simulations, based on AERONET measurements, and irradiance measurements from ATOLL in clear-sky conditions in Lille showed satisfactory results (Section 2.3.3). This validates both the

---

[1] The "Strong events" class is not considered for the decomposition of the clear-sun irradiances as it represents less than 1% of observations in Lille.





modelling of the aerosol optical properties and the radiative transfer processes. Hence, we use with confidence SOLARTDECO simulations to compute the sensitivities of irradiances to the various atmospheric parameters, as described in the next section 3.2.2.

### 3.2.2    Sensitivity study of clear-sky irradiances to atmospheric parameters

The sensitivity of clear-sky irradiances, «$F$», to each input of SOLARTDECO, «x», is quantified by the computation of partial derivatives, which are obtained by imposing a small perturbation «$\delta x$» to the value of x as in Equation 24:

$$\frac{\partial F}{\partial x} \approx \frac{\delta F}{\delta x} = \frac{F(x+\delta x, \ y_1, \ ... \ , \ y_n) - F(x, \ y_1, \ ... \ , \ y_n)}{\delta x} \tag{24}$$

where $y_1, \ ... \ , y_n$ are the other parameters, beside the variable of interest $x$, needed to compute the clear-sky radiative irradiances
$F$ (i.e. GHI, BHI or DHI). As in Thorsen et al. (2020), the perturbation of the parameter $x$ is taken as an arbitrary increase of 1% of the base value, which should be simultaneously small enough to induce a linear impact on the irradiances and large enough to avoid noise from numerical truncation errors. The analysis of these sensitivities is interesting in itself as it provides understanding about the importance of each parameter, and how accurately they should be defined. It also shows how different the sensitivities of the direct and diffuse irradiances can be, and how they can lead to reduced sensitivities of GHI through
compensation mechanisms. It indicates also the possible importance of aerosol's nature that differently interact with solar radiation over the solar spectrum. As the values of the scene's pertaining parameters have different magnitudes (e.g. SZA versus AOD), we chose to provide here values of the logarithmic sensitivity:

$$\frac{\partial ln(F)}{\partial ln(x)} = \frac{\partial F/F}{\partial x/x} \tag{25}$$

where F represents the irradiances (GHI, BHI or DHI) and «x» the parameter of interest. The logarithmic sensitivities thus
represent the response of the irradiances (in %) to a relative variation in the input parameter (here an increase) of 1 %.

Table 4a presents the logarithmic sensitivities of the clear-sky irradiances to the input parameters of SOLARTDECO (aerosol-related parameters are grouped) in the case of the most frequent aerosol class in Lille, i.e. the continental class. The sensitivities are calculated with, as the reference, a fixed set of properties that correspond to mean values observed in clear-sky conditions over the period 2010-2022. Note that for the aerosol layer height ($H_{aer}$) we used the fixed input value (2 km) from
SOLARTDECO described in Section 2.3.1. The mean value of fine fraction «ff» used in this study comes from AERONET estimates based on the method of O'Neill et al. (2003). Moreover, as the number of available AERONET inversions of SSA is relatively small and thus less representative, the mean single scattering albedo is based on the aerosol optical properties computed by SOLARTDECO (Equation 14) for the clear-sky simulations over 2010-2022. The most important parameter is the solar zenithal angle whose 1 % increase (0.6°) induces a 2.5 % decrease of the GHI. The second most significant parameter is
the single scattering albedo that logically affects only the diffuse component. The third most important parameter is the aerosol optical depth, although the compensation between two opposite in sign effects on BHI and DHI leads to an overall effect on GHI on the order of the effects of PWV variation. It is to be noticed that an increase of the fine fraction, for a constant $AOD_{550}$,




|  | GHI | BHI | DHI |
|---|---|---|---|
| **SZA (°)** | -2.47e+00 | -2.80e+00 | -1.18e+00 |
| AOD$_{550}$ | -4.92e-02 | -1.67e-01 | 4.08e-01 |
| **SSA$_{550}$** | 1.71e-01 | 0.00e+00 | 8.38e-01 |
| **ff** | 1.59e-03 | 2.18e-02 | -7.69e-02 |
| **H$_{aer}$ (km)** | -2.12e-04 | 0.00e+00 | -1.04e-03 |
| **RH (%)** | 2.81e-03 | -3.98e-04 | 1.53e-02 |
| **PWV (cm)** | -5.68e-02 | -6.68e-02 | -1.78e-02 |
| **O$_3$ (DU)** | -2.74e-02 | -2.41e-02 | -4.01e-02 |
| **O$_2$ (ppmv)** | -7.55e-03 | -8.03e-03 | -5.69e-03 |
| **CO$_2$ (ppmv)** | -2.64e-03 | -3.25e-03 | -2.55e-04 |
| $\alpha$ | 1.30e-02 | 0.00e+00 | 6.34e-02 |

(a)

|  | GHI | BHI | DHI |
|---|---|---|---|
| **SZA (°)** | -58.6 | -66.5 | -28.0 |
| AOD$_{550}$ | -3.0 | -10.3 | 25.1 |
| **SSA$_{550}$** | 0.6 | 0.0 | 2.7 |
| **ff** | 0.0 | 0.5 | -1.9 |
| **RH (%)** | 0.1 | -0.0 | 0.5 |
| **PWV (cm)** | -2.6 | -3.0 | -0.8 |
| **O$_3$ (DU)** | -0.3 | -0.2 | -0.4 |

(b)

**Table 4.** Logarithmic sensitivities of global (GHI), direct (BHI) and diffuse (DHI) irradiances to SOLARTDECO input parameters (the parameters related to aerosol properties are grouped) in the case of the continental aerosol class. Results are given in % and equal the relative variation of irradiances for (a) a 1 % variation and (b) a variation of the order of the coefficient of variation, of each input parameter. Computations were carried out for the continental aerosol class using a fixed set of parameters based on average properties observed under clear-sky conditions and over 2010-2022. The parameter ff represents the aerosol fine mode fraction, RH the relative humidity, PWV the precipitable water vapor content, $H_{aer}$ the aerosol layer height (fixed at 2 km), and $\alpha$ the surface albedo.

does increase BHI and decrease DHI, which results from an overall decrease of the aerosol extinction over the solar spectrum, consequence of a higher Angström exponent. Sensitivity's results in the case of larger particles, e.g. maritime, not shown here,
show sensitivities to AOD 5 to 10 % higher in magnitude. It shows also a sensitivity to ff two of three times higher in magnitude. Both differences can be attributed to a modification of the aerosol's optical properties over the whole solar spectrum.
It is nevertheless biased to compare the irradiance's sensitivities to all parameters as in Table 4a, as these parameters have actually different ranges of variation. It is thus more pertaining to account for these different ranges by multiplying each logarithmic sensitivities by the coefficient of variation of each parameter, defined as the ratio between the standard deviation and
the associated mean value of this parameter observed in Lille. Table 4b provides the sensitivities of the irradiances (in %) to the various parameters for changes of the order of the coefficient of variation, still in the case of the aerosol's continental class. Note that the sensitivities to the surface albedo ($\alpha$), oxygen (O$_2$), carbon dioxide (CO$_2$) concentrations, and aerosol layer height ($H_{aer}$) are not represented, as these parameters are constant in our simulations. Accounting for the variation of the input parameters, the clear sky irradiance's sensitivities in Lille are thus, in decreasing order, related to the solar zenith angle, the
aerosol optical thickness, the precipitable water vapor content, the single scattering albedo and the fine fraction, with relatively weak impacts (<1% for all irradiance components) of the relative humidity and ozone content. The ordering of the sensitivities to SSA and PWV are inverted when considering only the diffuse irradiance. These results indicate the primary importance of





the SZA: a temporal change of a clear-sky condition can lead to a significant change of the solar irradiance, through modification of both the air mass and the horizontal projection of radiances, that affect more the direct than the diffuse horizontal
irradiance. Also, a typical increase of aerosol loading $(\overline{\mathrm{AOD}} + \sigma)$ would lead to a 10 % decrease of BHI, a 25 % increase of DHI, and a 3 % decrease of GHI. The global radiative impact of aerosols on the solar irradiance in Lille over 2010-2022, based on SOLARTDECO simulations as well as ATOLL irradiance measurements, is presented in Section 4 for clear-sky conditions, as well as clear-sun with clouds and all-sky conditions, on seasonal and yearly averages, and per aerosol class.

Finally, it is important to note that the sensitivities presented in this section are only for the continental class and over the entire
period 2010-2022. Specific sensitivities, which vary according to the considered time period as well as the aerosol class, have been computed in order to perform the multivariate analysis of inter-annual variability and trends presented in Sections 3.2.3 and 3.2.4.

### 3.2.3   Multivariate analysis of spring 2020

As aforementioned, the approach presented in Section 3.2.1 can be applied to the analysis of the year-to-year variability of the
solar irradiance measured at the surface for all seasons, as well as to the study of the observed seasonal trends in irradiances presented in Section 3.1.2.

As an example, we analyze the record-high mean GHI value of spring 2020 measured at the ATOLL platform. Spring 2020, in connection with the very particular conditions linked to COVID-19 and the associated confinement period, has also been the subject of numerous studies about anthropogenic pollution and air quality in France and around the world (Ordóñez et al.,
2020; Reifenberg et al., 2022; Voigt et al., 2022; Cuesta et al., 2022; Petit et al., 2021; Fu et al., 2020; Velders et al., 2021), as well as about the possible impact of the reduction in anthropogenic emissions on incident solar radiation (Shuvalova et al., 2022; Reifenberg et al., 2022; Heerwaarden et al., 2021). In particular, the study of Heerwaarden et al. (2021) showed similarly high values of all-sky global irradiance in Cabauw (Netherlands), a site about 250 km north-east of Lille, who experienced the same large scale high pressure meteorological situation. The mean GHI for that spring was of 389 W/m$^2$, 55 W/m$^2$ (i.e. 16%)
higher than the average springtime GHI observed in Lille over the 2010-2022 period.

Figures 7a and 7b illustrate the results obtained from our methodology for spring 2020 in Lille. Figure 7a presents the results of the decomposition of all-sky irradiance measurements in terms of sky conditions, using Equation 17, for all irradiance components. Figure 7b presents the additional decomposition of the direct irradiance component under clear-sun with clouds ($\Delta F_{CSWC}$) and clear-sky ($\Delta F_{CSKY}$) conditions in terms of changes in aerosol class' partition and changes in scene's
parameters, as in Equations 20 and 22.

Overall, the decomposition of all-sky irradiances based on the different sky condition categories (cloudy-sun, clear-sun with clouds, and clear-sky), satisfactorily reproduces the measured differences in irradiances between the mean values of spring 2020 and the overall averages over the period 2010-2022 for all components (Figure 7a). Indeed, our approach estimates a variation of +65, +89, and -25 W/m$^2$ for GHI, BHI and DHI, respectively, against the measured +55, +81 and -25 W/m$^2$. As
discussed in Section 3.1.2, spring 2020 was characterized in Lille by an exceptionally low frequency of cloudy-sun conditions (44% compared to an average of 60% in spring over 2010-2022, Figure 5a). In contrast, clear-sky conditions were more than




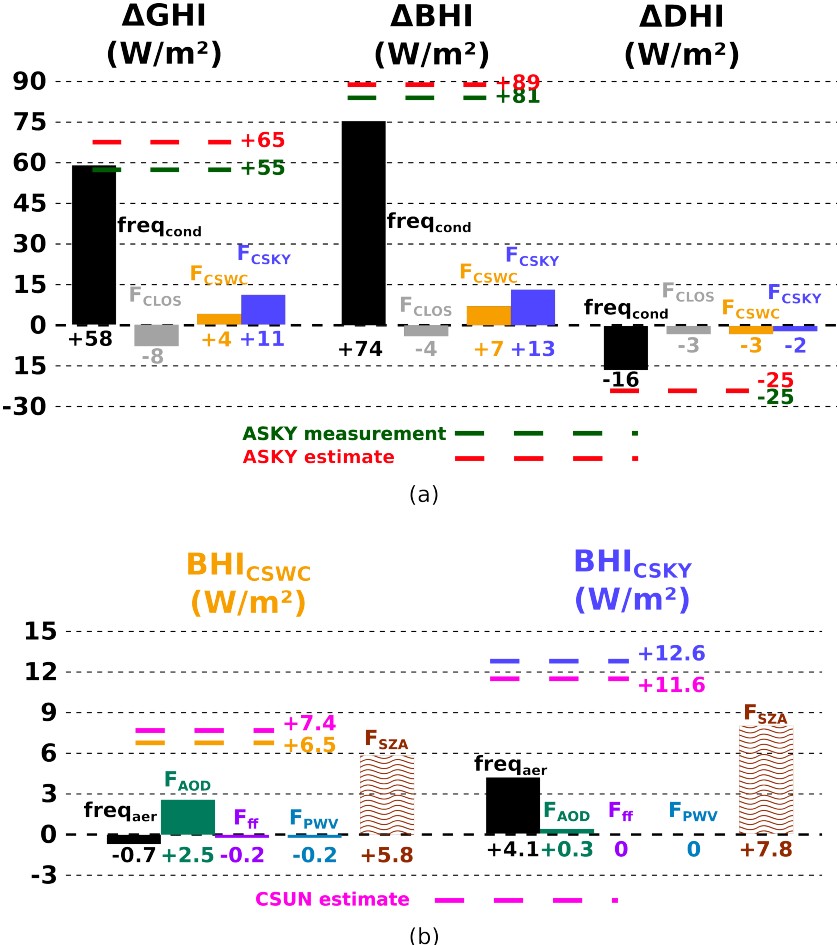

**Figure 7.** Illustration of the decomposition for spring 2020 : (a) of the all-sky irradiances (GHI, BHI, and DHI), and (b) of the BHI in clear-sun with clouds and clear-sky conditions. In Figure (a), the green dashed lines represent the observed deviation from the 2010-2022 spring average for spring 2020. The red dashed lines correspond to the deviation derived from the decomposition based on the sky conditions. The decomposition is represented by the black, gray, orange, and blue colored columns, which respectively correspond to the contributions of the variability in the frequency of occurrence of sky conditions ($freq_{cond}$) and the intrinsic variability of the irradiances in CLOS ($F_{CLOS}$), CSWC ($F_{CSWC}$), and CSKY ($F_{CSKY}$) conditions. In Figure (b), the orange and blue dashed lines represent the terms $F_{CSWC}$ and $F_{CSKY}$ from Figure (a). The pink dashed lines represent the values estimated from the decomposition of the BHI based on aerosol classes. The black, green, violet, blue, and brown columns correspond to the contributions of the variability in the frequency of occurrence of aerosol classes ($freq_{aer}$) and various considered parameters (AOD, ff, PWV, and SZA).





two times more frequent, representing 34% of situations in spring 2020 compared to the spring average of 15%. This change in sky conditions lead, as shown in Figure 7a, to an increase in BHI of 74 W/m$^2$ (+83% of the total increase in BHI) and a decrease in DHI of -16 W/m$^2$ (-64% of the total change in DHI). It did result in an increase in GHI of +58 W/m$^2$, which represents 89 % of the total increase of GHI. Our approach also estimates that the remaining contribution (of only +7 W/m$^2$, +11% of GHI's change) comes mostly from changes in BHI resulting from partially opposite in sign effects, with fewer cloudy-sun situations being less bright at the surface ($F_{CLOS} = -8$ W/m$^2$), more clear-sky situations being brighter ($F_{CSKY} = +11$ W/m$^2$), and clear-sun with clouds being slightly brighter ($F_{CSWC}$=+4 W/m$^2$).

The results of the decomposition of the changes in BHI under clear sun situations (CSUN, i.e. the sum of CSWC and CSKY) related to changes in aerosol classes and scene's parameters, illustrated in Figure 7b, provide more insight into the remaining +17 % increase of BHI beside the effects of the change in sky situation. They suggest that most of the increase due to change in scene's parameters is due to a decrease in SZA for CSUN situations (-1.5° and -0.9° in CSKY and CSWC situations respectively), that by itself causes an increase of around +14 W/m$^2$, which represents 71 % of the overall increase of $F_{CSUN}$. The remaining 29 % comes mostly from a change in aerosol properties. In CSWC situations, increase of BHI due to SZA is three times higher than the increase due to a change in aerosol properties (average AOD of 0.16 in 2020 compared to the 2010-2022 average of 0.22). In CSKY situation, the modification of aerosol class frequencies (significant decrease in the occurrence of the Continental Polluted class in favor of the Continental class, see Figure 5d) results in a contribution slightly more than half of that due to changes in SZA (+8 W/m$^2$).

Note that the variability of the mean SZA under clear-sun conditions could partly explain the decrease in both direct and diffuse irradiances estimated for cloudy-sun situations. Indeed, a decrease in SZA for CSUN situations implies an increase in SZA for opposing situations (i.e. CLOS conditions), potentially resulting in lower irradiance values for cloudy-sun conditions of spring 2020 compared to the average observed in spring over the period 2010-2022.

It is important to emphasize that all parameters considered here also play a role in cloudy-sun conditions. Since these situations have not been evaluated in the present study, the results presented here may underestimate (or overestimate) the overall impact of each parameter. Nonetheless our estimates are consistent with the results of Heerwaarden et al. (2021), which suggests that the observed irradiance record of spring 2020 in Cabauw is primarily due to the exceptionally low cloud fraction that year (contribution of approximately also 89%) with modest additional contributions of changes in AOD (6%) and water vapor content (5%).

Our methodology was also used to explain the variability of irradiances for other years in spring and summer. Our analysis highlighted again the primary influence of the variability of the sky conditions, which also explain, for example, 69% and 84% of the variability of the all-sky GHI observed during the summer of 2022 (maximum of 421 W/m$^2$) and spring 2013 (minimum of 285 W/m$^2$), respectively. It showed also again the primary importance of SZA, associated with the variations of the occurrences of the sky conditions, which plays a key role in the variability of the irradiances for other years, while the influence of the variability of both aerosol and water vapor contents are relatively limited. These results are consistent with the order of importance of the parameters determined by the sensitivity analysis in Section 3.2.2.





### 3.2.4 Multivariate analysis of spring and summer trends

As stated in Section 3.1.2 and Table 3c, all-sky GHI irradiances show significant positive trends in spring and summer with respective magnitudes of $4.0 \pm 1.9$ W/m$^2$/year and $4.2 \pm 1.9$ W/m$^2$/year, due to an important increase of the direct component, with magnitudes of $+4.4 \pm 2.3$ and $+4.7 \pm 1.9$ W/m$^2$/year in spring and summer, respectively. Figures 8a and 8b illustrate the

results of these trends' decomposition obtained from Equation 18.

The decomposition of the trends shows similar features in spring and summer. The observed upward trends in all-sky GHI and BHI are largely linked to the trends in the occurrence of sky conditions, with contributions in spring of approximately $+2.7$ (71%) and $+3.3$ (79%) W/m$^2$/year for the global and direct components, respectively, and $+2.6$ (62%) and $+3.4$ (74%) W/m$^2$/year in summer. These results are consistent with the negative trends in the frequencies of cloudy-sun situations observed

for both seasons, in almost equal favor of clear-sun with clouds and clear-sky situations (Table 3c). The remaining contributions in the GHI and BHI upward trends arise from almost only positive irradiance trends observed under the various sky conditions. The most important contribution among irradiance trends comes from CSWC conditions (approx. $+0.8$-$0.9$ W/m$^2$/year for both components), which align with the significant positive trends observed for these situations (Table 3c) and the relatively high seasonal frequencies associated (25% in spring, 22% in summer). The statistical significance of these trends and the similarities

between observed and derived all-sky GHI and BHI trends further validate these conclusions.

We further analyze the BHI variability in CSWC and CSKY conditions, for spring and summer, decomposing their trends following Equations 21 and 23. Note that apart from the CSKY BHI trend in summer (where no parameter trends are significant), the computed trends generally yield results similar to the observed trends, with magnitudes within the uncertainty of the observations ($\pm\sigma$). The significant trends observed for CSWC BHI in spring ($+3.7 \pm 1.7$ W/m$^2$/year) and summer ($+3.8 \pm 1.2$

W/m$^2$/year), that contribute to the all-sky BHI trends by 0.85 and 0.91 W/m$^2$/year, can be decomposed as in Figures 9a and 9b. It shows that the main contribution to these trends in spring are related to changes in aerosol loading, through variations in aerosol class' frequencies ($+0.21$ W/m$^2$) and increase of AOD ($+0.35$ W/m$^2$) that explain 87% of the CSWC BHI trend, while changes in SZA for those situations account for around 14% of the CSWC BHI trend. These results align with the significant trends observed for both AOD and aerosol class frequencies in spring (Table 3b). In summer, the contribution due to aerosol

is smaller, 49%, while the contribution from changes in SZA does reach 44%. Hence, although the trends in BHI in CSWC situations are very close in magnitude in spring and summer, the contribution of the different parameters are quite different for both seasons. It results from a decrease of AOD more than two times larger in spring compared to summer (see Table 3b), and a decrease in SZA three time smaller in spring ($-0.03 \pm 0.09$ degrees/year, confidence of 62%) than in summer ($-0.094 \pm$ 0.09 degrees/year, confidence of 80%). The contribution of the SZA variability is thus not to be neglected. Beside its effects

on CSWC BHI, it also explains the limited CSKY BHI trend in spring, as the effects of the AOD decrease is compensated by an increase in SZA ($+0.04 \pm 0.14$ degrees/year, confidence of 52%). Furthermore, changes in SZA ($-0.1 \pm 0.09$ degrees/year, confidence of 80%) represent almost the sole contribution to the CSKY BHI trend in summer (Figure 9).




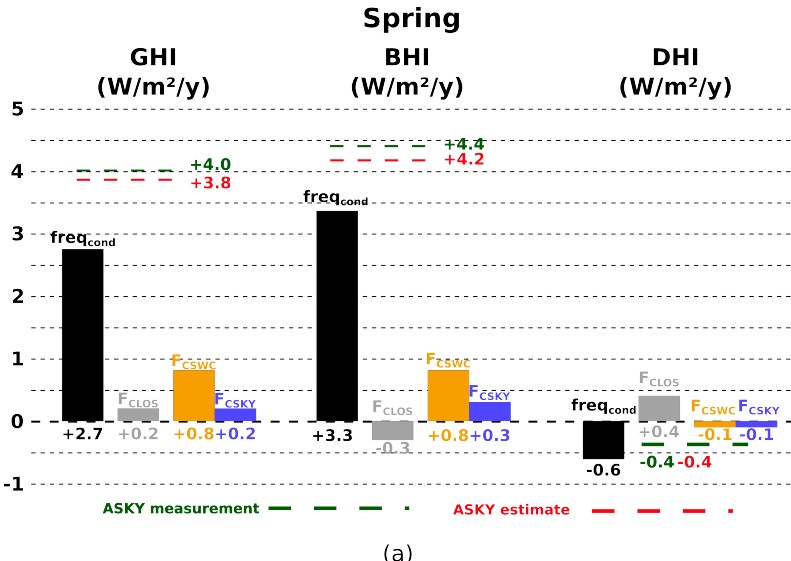

(a)

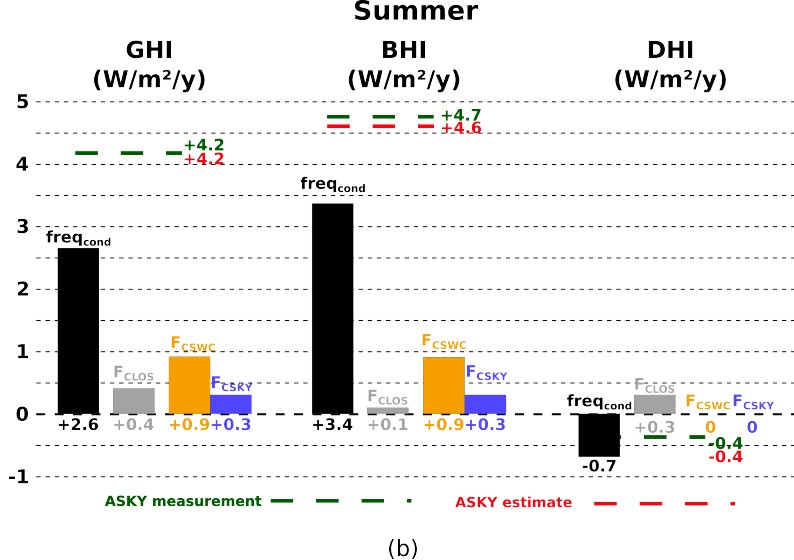

(b)

**Figure 8.** Decomposition of the observed trends in all-sky irradiances (a) in spring and (b) in summer. The meaning of the colored lines and columns is similar to that described in Figure 7a.



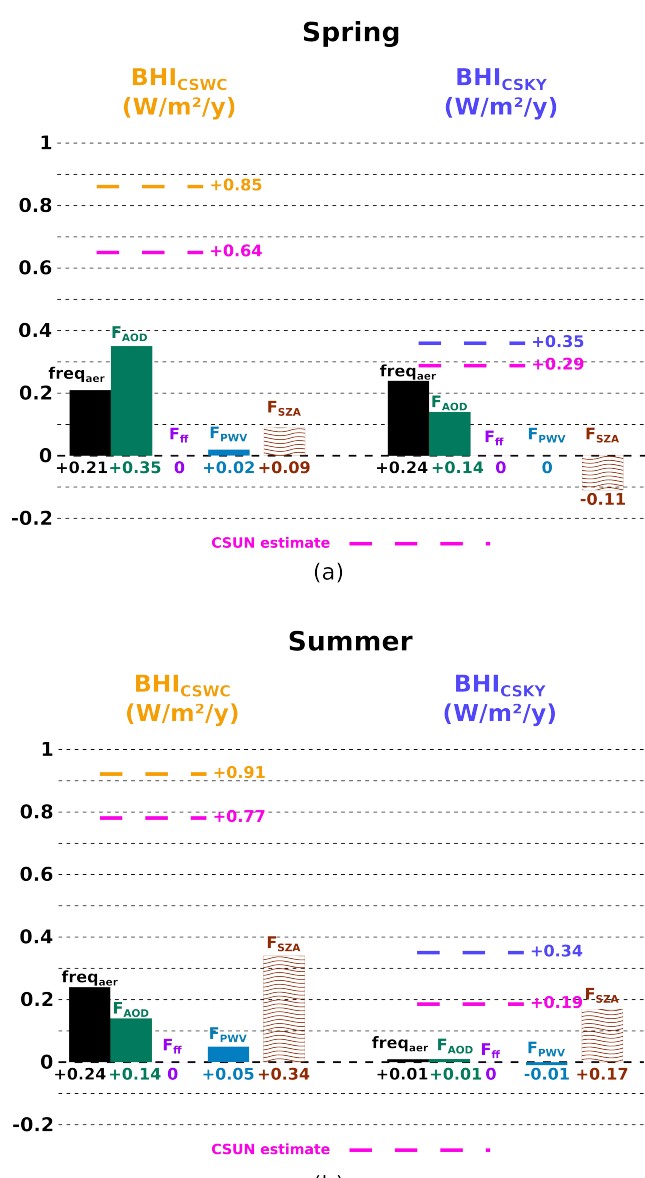

**Figure 9.** Illustration of the decomposition of observed trends in BHI under clear-sun with clouds and clear-sky conditions (a) in spring and (b) in summer. The meaning of the colored lines and columns is similar to that described in Figure 7b.



Finally, it is worth noting that the influence of the other atmospheric parameters remains relatively limited, even for the water vapor content whose contribution is maximum for both seasons in clear-sun with clouds conditions, but is limited to +3% and +6% in spring and summer, respectively.

## 4 Direct radiative effects (DRE) in Lille over 2010 - 2022

In this section, we study the radiative impact of atmospheric components in the solar domain over the period 2010-2022, focusing on clear sun situations. The objective here is to compute the direct radiative impact of aerosols in clear sky situations, and to analyse the overall effects of atmospheric components in clear-sun-with cloud situations.

Numerous studies in the literature have assessed the direct radiative impact of aerosols, both globally (Yu et al., 2006; Bellouin et al., 2013; Thorsen et al., 2020; Kinne, 2019; Zhou et al., 2005) and regionally (Papadimas et al., 2012; Nabat et al., 2014; Witthuhn et al., 2021; Bartók, 2017; Neher et al., 2019). Most of these studies focus on the net radiative impact of aerosols, which is the difference between net radiation incident at the surface (i.e. the difference between downwelling and upwelling solar flux) in the presence and absence of aerosols. Here, we chose instead to assess the direct (non-net) radiative effect of aerosols on the downwelling solar irradiances incident at the surface. It is consistent as our study is based on measurement of the downwelling surface surface irradiance and it allows a direct assessment of the aerosols' direct impact on the solar source exploited by surface processes (e.g. photosynthesis) and solar applications (thermal or photovoltaic solar systems). It is to be noticed that compared to most existing studies about the surface DRE of aerosols, we analyze here the effect of aerosols on both the direct and diffuse components of the solar irradiance.

Our analysis of the direct radiative effects is initiated in Section 4.1 with a focus on clear-sky conditions, through a statistical investigation of the aerosol mean direct effects over the period 2010-2022 (Section 4.1.1). It is complemented with an additional analysis of specific case studies (Section 4.1.2) that provide more analysis of invariant and distinctive aerosol effects. Section 4.2 provides an analysis of radiative effects in CSWC conditions where both aerosols and clouds acts on the diffuse component of SSI. These situations are particularly interesting as their occurrence is important (twice the occurrence of clear sky situations), and as on average, irradiance levels are strong with both a high direct irradiance and an exceptional level of diffuse irradiance. In addition to the masks allowing their identification, the radiative transfer simulations in aerosol-and-cloud free and in cloud-free situations allow a statistical analysis of the direct radiative effects of atmospheric particles and their cumulative effects.

### 4.1 Aerosol DRE in clear-sky conditions

The direct radiative effect (DRE) of aerosols is defined as:

$$\text{DRE}_{aer,\ CSKY} = F_{meas} - F_{pristine} \tag{26}$$

or in relative terms :

$$\text{DRE}_{aer,\ CSKY} = \frac{F_{meas} - F_{pristine}}{F_{pristine}} \tag{27}$$



where $F_{pristine}$ and $F_{meas}$ correspond respectively to the pristine simulations by SOLARTDECO and to the ATOLL measurements of the downwelling solar irradiance components (GHI, BHI or DHI) in clear-sky conditions. The Direct Radiative Effect Efficiency (DREE) of aerosol under clear-sky conditions, defined as :

$$\text{DREE}_{aer,\ CSKY} = \frac{\text{DRE}_{aer,\ CSKY}}{\text{AOD}_{440}} \tag{28}$$

is also investigated in this section, as it allows the comparison of the DRE for different aerosol loads (see for example Kok et al. (2017)). As the DRE is approximately linear with AOD (Satheesh and Ramanathan, 2000), DREE provides the sensitivity of DRE to AOD. An additional analysis of the direct radiative effect of aerosols is carried out in Section 4.1.2 for some case studies that are typically observed in Lille, to better appreciate the variations of the direct radiative effects of natural and anthropogenic particulate pollution.

### 4.1.1 Global statistical analysis

The mean direct radiative effect of aerosols in Lille is investigated over the whole period 2010 - 2022. Results are analyzed by season, by aerosol class, and also by distinguishing clean ($\text{AOD}_{440} \leq 0.1$) and polluted ($\text{AOD}_{440} > 0.1$) situations. Table 5 provides the mean absolute and relative DRE of aerosols along with the corresponding DREE for all coincident CSKY measurements, and for three components of the surface solar irradiance (SSI).

Our results show that, in Lille, over the period 2010-2022, the overall yearly-averaged radiative impact of aerosols constitute a loss of around -61 W/m$^2$ (-18%) of direct irradiance, and a gain of roughly +42 W/m$^2$ (+92%) of diffuse radiation, compared to pristine conditions. This leads to an important modification of the DHI contribution with a twofold (+105 %) increase of the average DHI/GHI ratio from 12 % in pristine conditions to an average of 24 % in CSKY conditions. The opposite-in-sign aerosol effects on BHI and DHI leads to a reduced effect of GHI, evidently always negative, which is on average reduced by about 20 W/m$^2$ (-5.5 %). During polluted CSKY situations, which happen 73 % of the time, the aerosol DRE exceeds -20 % and +100% for BHI and DHI, and -6 % for GHI. Inversely, for clean situations, that mostly append in fall and winter, the aerosol DRE is weaker and smaller in magnitude than -10 % and +50% for BHI and DHI, and -3 % for GHI, i.e. two time smaller that in polluted situations. The compensation mechanism between DRE on direct and diffuse irradiances shows, through the values of DREE, some aerosol-independent feature, with direct and diffuse components that are more and less three times and two times respectively more sensitive to the AOD than GHI.

The analysis of the DRE indicates that the maximum direct radiative impact happens both in absolute (-21/-68/+47 W/m$^2$) and relative (-6/-19/+102 %) terms in spring, with slightly higher values compared to summer (-19/-63/+44 W/m$^2$; -5/-16/+94 %), and also for continental polluted aerosols (-32/-99/+67 W/m$^2$, -9/-28/144 %). This result is consistent with our analysis of the seasonal variability of the solar environment in Lille, which shows higher values of AOD in spring than in summer (Figure 4b), and a high proportion of continental polluted aerosol in spring. The minimum direct radiative impact occurs in absolute values in winter (-16/-38/+22 W/m$^2$) and for maritime aerosols (-12/-36/+25 W/m$^2$). This result is also consistent with the observed lower values of AOD in winter and higher proportion of maritime aerosols during this season. It can also be related to lower SZA that lead to overall lower surface irradiances.





| DRE of aerosols in CSKY conditions in Lille (2010- 2022) | | |
|---|---|---|
| $(\text{W/m}^2)$ [%] $< \dfrac{\text{W}/m^2}{unit\ of\ \text{AOD}_{440}} >$ | | |
| **GHI** | **BHI** | **DHI** |
| **Total** (-19.7) [-5.5] <-125> | (-61.4) [-17.5] <-379> | (41.7) [91.7] <254> |
| **Winter** (-15.6) [-6.4] <-158> | (-37.9) [-18.3] <-386> | (22.3) [59.8] <227> |
| **Spring** (-21.4) [-5.9] <-116> | (-68.3) [-19.1] <-377> | (46.8) [101.7] <261> |
| **Summer** (-18.8) [-4.8] <-120> | (-62.7) [-16.4] <-376> | (43.9) [93.5] <256> |
| **Autumn** (-19.2) [-5.5] <-139> | (-53.7) [-15.9] <-385> | (34.5) [80.2] <245> |
| **Continental** (-15.0) [-4.0] <-129> | (-43.2) [-11.7] <-362> | (28.2) [61.5] <233> |
| **Continental polluted** (-31.6) [-8.7] <-105> | (-98.6) [-27.5] <-326> | (67.0) [144.4] <221> |
| **Mixed** (-22.3) [-6.2] <-120> | (-73.8) [-21.2] <-396> | (51.5) [112.6] <276> |
| **Maritime** (-11.5) [-4.2] <-152> | (-36.2) [-14.7] <-463> | (24.7) [61.3] <311> |
| **Desert dusts** (-25.4) [-6.2] <-126> | (-92.1) [-21.2] <-456> | (66.7) [149.3] <330> |
| **Clean** (-10.5) [-2.9] <-144> | (-30.1) [-9.2] <-403> | (19.6) [45.8] <259> |
| **Polluted** (-23.2) [-6.4] <-118> | (-73.5) [-20.7] <-369> | (50.2) [109.5] <252> |

**Table 5.** Mean (absolute) and [relative] aerosol Direct Radiative Effects (DRE) and associated absolute efficiencies (DREE) in clear-sky conditions in Lille over the period 2010 - 2022 for coincident AERONET measurements only. The instantaneous values were computed using ATOLL irradiance measurements and SOLARTDECO pristine simulations for all irradiance components. The "total" mean corresponds to the overall mean DRE and DREE for all coincident measurements in clear-sky conditions over the period 2010 - 2022. Similar averages were made for each season as well as for the different classes and for clean ($AOD_{440} \leq 0.1$) and polluted conditions.

The DRE efficiency (DREE) is higher in magnitude for Maritime and Desert Dust aerosols, with values above -450 W/m² and +310 W/m² for BHI and DHI, respectively. These results align with the conclusions of the sensitivity analysis (Section 3.2.2), which shows higher sensitivities of the direct and diffuse components to Maritime and Desert dust aerosols. As already hypothesized, this result may be linked to the differences in the size of the particles, as lower absolute value of Angström exponent for large particles imply higher extinction properties over the whole solar spectrum. Note that, on average, the radiative impact of the Maritime and Desert dust classes remains however relatively limited over the course of the year, as these two classes represent only 14% and 5% of observed situations, respectively. Moreover, although the Maritime class exhibits large DREE, the corresponding DRE values remain relatively small due to low corresponding aerosol optical depths. Therefore, the radiative impact of aerosols in clear-sky conditions is largely dominated by the Continental, Continental polluted, and Mixed classes, which represent 38%, 20%, and 22% of situations in Lille, respectively.



### 4.1.2 Focus on several case studies

In this section, the direct radiative effect of aerosol is computed for three clear-sky case studies characterized by different aerosol loads and properties. The objective is to analyse the intra-day variability of the DRE and DREE of aerosols for each
case study in order to complement the statistical analysis presented in the previous section 4.1.1. These days, almost entirely clear, were selected based on the availability of AERONET measurements, as well as through a visual analysis of the pictures of the sky imager. The mean atmospheric properties of each day are summarized in Table S1, with associated AOD and AE values represented in Figure S3. Table S1 also illustrates the performance and limitation of our clear-sky filter, which appears to work well under low aerosol loads, but less so for higher AOD values, in accordance with the results of Liu et al. (2021) that
assessed the performance of several clear-sky detection methods under various aerosol loads.

The first case study is representative of a typical spring pollution event, that has been already investigated as a heavy regional anthropogenic pollution event, recorded in the Paris region (Dupont et al., 2016) and over many French ground-based sites (Favez et al., 2021). This episode is characterized by an accumulation of fine particles over six consecutive days from the 9th to the 14th of March 2014, with a monotonous increase of the AOD from a minimum of 0.1 (Continental/Mixed class) on
the first day to a maximum of more than 0.6 (Continental polluted class) on the last day (Figure S3). Over Lille, our analysis shows that most days were associated to cloud free conditions except for the 03/11/2014, which was completely cloudy, and the 03/14/2014, which was only free of clouds between 10 and 15h UTC. The water vapor content and Angström exponents are relatively similar for the remaining five days with daily mean values ranging from 0.62 to 0.83 cm for the PWV and from 1.28 to 1.46 for the AE (Table S1). For concision, only the DRE and DREE values computed for the 03/09 (✕), 03/13 (●) and
03/14 (▶) are represented in the following figures and analyzed.

In Lille, dust intrusion episodes are rare (occurrence lower than 5%) and mostly one-off events. However, their radiative impact can be important as dust concentrations can reach high values. The second case study includes two dust intrusion events observed in Lille: a moderate event (09/17/2020 : ☐) with a value of $AOD_{440}$ of 0.23, and a very strong dust transport episode (04/01/2021 : ▼), characterized by a mean daily $AOD_{440}$ of 0.94. Note that the values presented for the 04/01/2021 correspond
to observations made after 15h UTC to avoid the presence of clouds.

The third case study represents two clean days characterized by similarly low values of AOD (∼0.06-0.07) and PWV (∼0.97 cm), one dominated by continental aerosols (02/25/2019 : ✙, AE∼1.35), and the other by maritime particles (05/28/2020 : ◇, AE∼0.74).

Figures 10a-f show the intra-day variability, for the various days introduced previously, of the relative DRE and DREE of
aerosols as a function of the solar zenith angle. Negative (resp. positive) values of SZA correspond to ATOLL observations performed before (resp. after) the daily solar noon. As expected, aerosols have an overall negative impact on the incoming total SSI as they reduce the direct component more than they increase the DHI, leading to lower GHI values (negative values of GHI DRE and GHI DREE, Figures 10a and d). Moreover, consistently with the results of Sections 4.1 and 3.2.2, the direct radiative effect of aerosols on the GHI is relatively moderate compared to that on the BHI and DHI, and overall, the DRE of
aerosols tends to increase (almost monotonously) with the AOD, as the variability of the GHI and BHI DREE is close for all




days, with a maximum of DRE (GHI/BHI/DHI) on the 04/01/2021 (important dust intrusion event) of about -43/-98/+610%. The results for these case studies show nonetheless that the DRE and DREE of aerosol also greatly depends on the solar zenith angle, as the GHI and BHI DRE (and DREE) appear to increase with the solar zenith angle, while the DHI DRE is maximum at lower solar zenith angles, in link with higher levels of incoming solar irradiance and a lower contribution of
the air mass to the scattered radiation. Furthermore, Figure 10f suggests that, for a given day, the diffuse efficiency is less dependent of the solar zenith angle although it is minimum at large SZA values. These case studies also confirm the results of Sections 4.1 and 3.2.2, which showed that the sensitivity of the BHI and DHI to the aerosol content is greater for low Angström exponents. This is particularly obvious for the diffuse component (Figure 10f), as days characterized by coarser aerosols (◇, □) show much higher DREE values throughout the day. Note however that this is true only for relatively low AOD values as
for the 04/01/2021, which is characterized by important aerosol loads, the associated DREE values are much lower than for the 05/28/2020 and 09/17/2020, and relate more to values found for finer aerosol contents. Although the day of the third case study with finer particles (02/25/2019, ✚) shows smaller BHI and DHI DREE than the associated day characterized by coarser particles (05/28/2020 : ◇), the GHI DREE is generally stronger by about -20 to -30%. This could be explained by the lower mean value of single scattering albedo (0.91 against 0.96, Table S1) which leads to a lower balance between the reduction of
the BHI and the increase in DHI for that day.

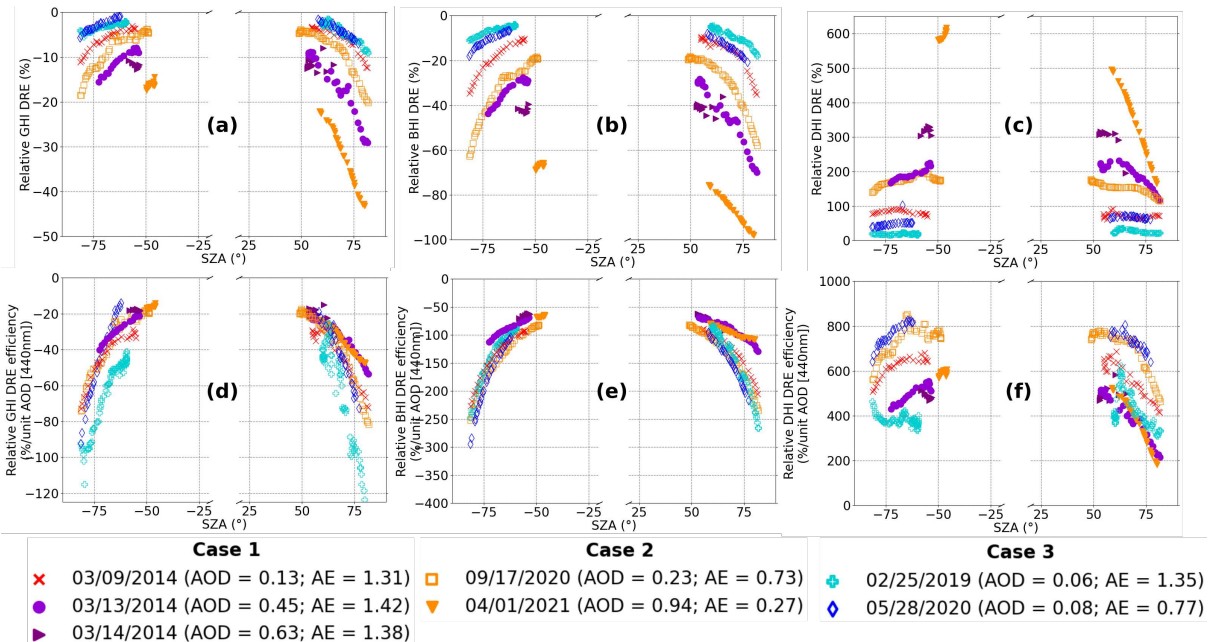

**Figure 10.** Evolution of the relative aerosol (a-c) DRE and (d-f) DREE as a function of solar zenith angle for the different case studies. The first column represents the aerosol effect over the GHI, the second for the BHI, and the third for the DHI. Negative (positive) values of SZA correspond to measurements taken before (after) the solar noon associated with each day.





## 4.2 DRE in clear-sun-with-cloud conditions

In clear-sun with cloud conditions, the overall particles DRE on the direct SSI component is only related to aerosols. On the contrary, due to the presence of clouds off the sun's direction, their effects on the diffuse component is more complex, with contributions of both aerosols and clouds.

Their respective effects on the SSI's diffuse component can generally not be separated. They are scene-dependent, and vary significantly with the cloud cover, notably the cloud fraction and the cloud opacities, even with the angular distance of the cloud cells to the sun direction or the vertical profile of aerosols relative to clouds. Undoubtedly, aerosol DRE on the diffuse SSI is minimal (resp. maximal) when the cloud fraction is high (resp. low). It is in any case out of the scope of the current study to ambition a complete distinction or disentangling of aerosols versus clouds DRE in CSWC situations.

The radiative effects of all atmospheric particles, hydrometeors and aerosols, lead to an overall DRE ($DRE_{all}$) that can be decomposed as:

$$DRE_{all} = F_{meas} - F_{pristine} = F_{meas} - F_{csky} + F_{csky} - F_{pristine} \tag{29}$$

$F_{meas} - F_{csky}$ is by definition the cloud's DRE, i.e. the difference between the downwelling solar irradiance actually measured and due to the overall effects of all atmospheric components (gas, aerosol and clouds), and the incoming radiation in the same

conditions without clouds (i.e. clear-sky). In the solar direction, $F_{csky} - F_{pristine}$ equals the aerosol DRE. However it is not the case concerning the diffuse SSI component. $F_{csky} - F_{pristine}$ is rather the radiation one would have compared to the pristine case if no cloud were present. Yet, in the rest of this section, $F_{csky} - F_{pristine}$ is named as in section 4.1 aerosol DRE, and is written as $F_{aer} - F_{pristine}$. Due the presence of clouds, $F_{aer}$ is obtained from SOLARTDECO clear-sky simulations.

The motivation to analyse $F_{aer} - F_{pristine}$ is to appreciate the difference in aerosol DRE between CSKY and CSWC situations, and the radiative 'compensation' between cloud and aerosol DRE in CSWC situations. Table 6 provides the mean absolute and relative values of $F_{aer} - F_{pristine}$ in CSWC situations over the period 2010-2022. Values of DRE are 15 to 20 % higher than in CSKY given in Table 5. This discrepancy is related to the slightly higher $AOD_{440}$ values observed in CSWC conditions (0.19) compared to CSKY conditions (0.17), as well as to slightly lower SZA values (60° on average in CSWC

against 62° in CSKY conditions), which lead to higher irradiance values, and thus greater absolute radiative effects. Note that DREEs are slightly greater than under clear-sky conditions, by about 30 W/m²/unit of AOD for all irradiance components. This result is consistent with the effects of lower solar zenith angles and with the observation that aerosol Ångström exponent are on average slightly lower (by 0.06) in CSWC situations compared with CSKY cases, leading to a higher DRE efficiency as illustrated on Figure 10f. Note that the study of CSWC situations allows us to analyze conditions characterized by the Strong

events class (see Table 2a), which were not identified as clear-sky either because of the presence of clouds or because of their high AOD values, as the clear-sky filter used in this study tend to erroneously misidentify CSKY conditions when the AOD is high (Liu et al., 2021). The aerosol DRE for the Strong events class appears very substantial for both direct (-260 W/m², i.e. -50%) and diffuse (+190 W/m², i.e. +381%) components, leading to a relatively important reduction of the GHI of about -70 W/m² (-13%) on average. The climatological impact of the Strong events class is however relatively limited as these situations



| DRE of aerosols in CSWC conditions in Lille (2010- 2022) | | |
|---|---|---|
| (W/m$^2$) [%] $< \dfrac{\text{W}/m^2}{unit\ of\ \text{AOD}_{440}} >$ | | |
| **GHI** | **BHI** | **DHI** |
| **Total** | | |
| (-21.2) [-6.0] <-126> | (-71.6) [-20.0] <-411> | (50.4) [108.1] <285> |
| **Winter** | | |
| (-17.2) [-6.5] <-158> | (-45.4) [-19.3] <-413> | (28.1) [71.7] <254> |
| **Spring** | | |
| (-24.2) [-6.2] <-121> | (-84.1) [-21.4] <-414> | (59.8) [125.4] <293> |
| **Summer** | | |
| (-20.6) [-5.1] <-110> | (-76.9) [-18.4] <-405> | (56.3) [115.0] <296> |
| **Autumn** | | |
| (-20.3) [-6.6] <-139> | (-62.2) [-21.2] <-417> | (41.9) [96.0] <277> |
| **Continental** | | |
| (-14.5) [-4.1] <-121> | (-45.4) [-12.9] <-372> | (30.9) [66.7] <252> |
| **Continental polluted** | | |
| (-31.3) [-8.0] <-91> | (-111.6) [-27.6] <-328> | (80.3) [165.8] <237> |
| **Mixed** | | |
| (-22.1) [-6.3] <-110> | (-78.4) [-21.9] <-395> | (56.3) [120.6] <284> |
| **Maritime** | | |
| (-16.5) [-5.5] <-198> | (-48.2) [-17.2] <-582> | (31.7) [74.5] <383> |
| **Desert dusts** | | |
| (-28.8) [-8.5] <-145> | (-103.1) [-30.6] <-517> | (74.3) [163.2] <372> |
| **Strong events** | | |
| (-70.2) [-12.8] <-76> | (-260.3) [-50.5] <-284> | (190.2) [381.1] <208> |
| **Clean** | | |
| (-12.3) [-4.0] <-163> | (-36.7) [-12.4] <-484> | (24.4) [56.2] <321> |
| **Polluted** | | |
| (-24.3) [-6.7] <-113> | (-83.8) [-22.7] <-386> | (59.5) [126.3] <273> |

**Table 6.** Mean (absolute) and [relative] aerosol 'DRE', and DREE, over the period 2010 - 2022 in Lille under clear-sun with clouds conditions. Due to the presence of clouds in the sky, the instantaneous values are not based on ATOLL irradiance measurements but rather computed using SOLARTDECO clear-sky simulations with as inputs aerosol properties issued from AERONET measurements.

only represent about 0.3% of the situations observed in Lille over the period 2010-2022.

As mentioned above, the overall direct radiative effect of atmospheric components in CSWC conditions is obtained from the difference between measurements and pristine-like simulations and results from combined and compensating effects of aerosol and clouds. Calculations of mean $F_{aer} - F_{pristine}$, $\text{DRE}_{cloud,\ CSWC}$, and $\text{DRE}_{all\ CSWC}$ over 2010-2022, per season

and aerosol situations are synthesized in Figure 11. The direct radiative effects of all atmospheric particles (aerosols and clouds) are represented in yellow in Figure 11, and those of clouds only are in grey. The DRE one would have in cloud-free situations is in blue and corresponds to the values in Table 6.

As expected, $\text{DRE}_{cloud,\ CSWC}$ for the direct SSI component is close to zero (with an accuracy of $\pm 1\,\text{W.m}^{-2}$), which is coherent with the absence of clouds in the sun's direction. This result further validates the performance of both the CSWC

filtering and our clear sky radiative transfer simulations (for most of the cases except for desert dust and strong event situations where small residuals exist).





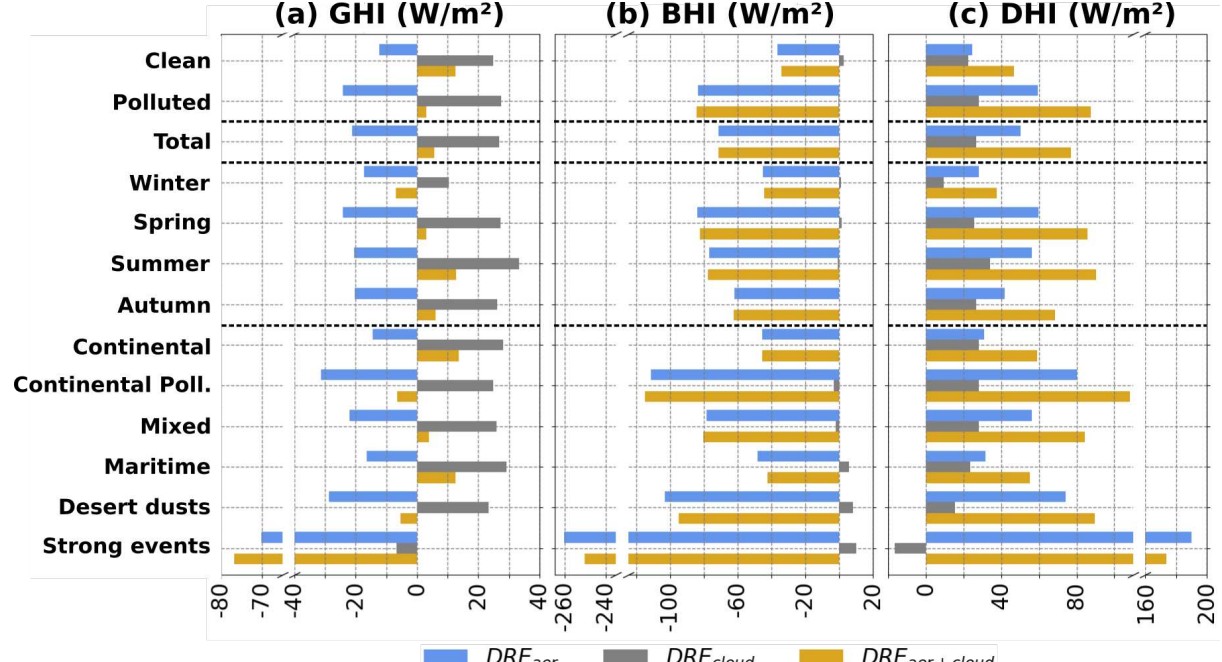

**Figure 11.** Mean Direct radiative effects, under CSWC conditions, of aerosols and clouds combined (yellow), clouds (gray), and cloud free DRE of aerosols (blue), on (a) the global irradiance, (b) the direct irradiance, and (c) the diffuse irradiance over the period 2010-2022. Points considered as the ones coincident AERONET measurements. Instantaneous values are calculated using ATOLL irradiance measurements and SOLARTDECO clear-sky simulations with and without aerosols. As in Table 5, the "total" averages correspond to the overall mean DRE over the period 2010-2022.

In the case of clean situations (AOD < 0.1), clouds contribute to an increase in DHI of around +22 W/m² (gray column, Figure 11c), comparable to the contribution of aerosols alone (+24 W/m², blue column). In these conditions, the decrease in

BHI due to aerosols (-37 W/m², Figure 11b) is overcompensated by the cumulative impact of both aerosols and clouds on the DHI, leading to GHI values greater by about +13 W/m² (yellow column, Figure 11a) than under quasi-pristine conditions (without aerosols and clouds). For polluted situations characterized by higher aerosol loads (AOD ≥ 0.1), which are more frequent, the cumulative effect of clouds and aerosols result in a GHI greater by +3 W/m² to the pristine situations. The cloud contribution on DHI, +28 W/m², is slightly higher than for clean situations while the aerosol 'cloud-free' contribution on BHI

and DHI is significantly higher. If one consider all CSWC situations, clouds add on average 25 W/m² compared to cloud-free atmospheres, i.e. half of the aerosol cloud-free DRE on the diffuse component. Cloud DRE is even higher than 50 W/m² twenty percents of the time. As a consequence, the mean observed GHI in CSWC situations is close to pristine conditions, or even slightly above by +4 W/m² (Figure 11a). The corresponding proportions of direct and diffuse irradiances are however in this case very different, with an average decrease of about -20% in BHI and an increase of +164% in DHI compared to pristine-like

conditions. The DHI proportion in the presence of clouds and aerosols under CSWC conditions is thus on average 27% higher





than under pristine conditions (cloud-free and aerosol-free).

The seasonal distinction reveals that the global DRE of particles is maximal in absolute values (resp. relative) in summer
(resp. autumn) with +12.7 W/m² (resp. +0.9%) above the pristine level. In spring, the global DRE is lower due to higher aerosol
DRE, but remains positive (+3 W/m²). In contrast, despite a weaker radiative impact of aerosols in winter, the corresponding
effect of clouds is so small (+10 W/m², i.e., three times less than the effect of aerosols, Figure 11c) that the reduction of
the BHI by aerosols is not fully compensated, with average GHI values lower than under pristine-like conditions by about -7
W/m² (Figure 11a). Similar results are observed for the Continental polluted, Desert dust, and Strong events classes, with a
decrease in GHI compared to pristine-like conditions of about -6, -5, and -77 W/m², respectively. In contrast, for maritime
and continental conditions, characterized by lower aerosol loads, the combined DRE of aerosols and clouds remains positive.
It should be noted that for conditions identified as CSWC at times where AERONET's algorithm fails to deliver information
about aerosols, certainly due to the presence of some thin clouds or varying-in-time conditions, the overall GHI is again on
the same order as for pristine conditions and the cloud DRE on GHI is of around 20 W/m², but it results from compensating
effects of clouds on the direct irradiance (-25 W/m², i.e. -7%), and on the diffuse irradiance (+45 W/m², +44%).
In summary, these results highlight the importance of clouds in CSWC situations that cause a significant enhancement effect on
the diffuse irradiance field. For these situations that occur 22% of the time, the mean GHI level is comparable to pristine-like
conditions (difference is +4 W/m²). The SSI is even higher than in pristine conditions during clean, summer, or continental-
aerosol moments with an averaged increase in solar radiation of up to 13 W/m². Clouds add on average between 20 and
30 W/m², depending on the situations, and overall on average around 25 W/m² compared with atmospheres that would be
cloud-free. Hence, they not only substantially increase the level of GHI but also modify the direct/diffuse partition of the SSI,
increasing by +4% (21 to 25%) the diffuse proportion compared to the cloud-free atmosphere, and by +15% (10 to 25%)
compared to the pristine atmosphere.

### 4.3   DRE in clear-sun conditions

Mean direct radiative effects can also be computed over clear-sun situations, i.e. the sum of CSKY and CSWC situations,
that represent on average 33 % of the situations per year over 2010-2022. These DREs write as the sum of the DRE per sky
conditions weighted by their respective occurrences, keeping in mind that the DRE of aerosols in CSWC conditions is equal to
$F_{aer} - F_{pristine}$, while it is rather, as already said, the DRE aerosols would have in the absence of clouds.
Overall, the yearly clear-sun aerosol DRE, for measurements coincident with AERONET retrievals, is -20 W/m² (-66.5 and
+46.6 on the direct and diffuse components, respectively), and the cloud DRE around +10 W/m². As clouds act only on the
diffuse component, the mean measured CSUN GHI is about 10 W/m² lower than under pristine conditions (no aerosols nor
clouds), but still 10 W/m² above the cloud-free level.
It is to be noticed that for all situations classified as clear-sun with cloud, including those where AERONET's retrieval fails,
the conclusion about the importance of cloud enhancement effects which set SSI levels close to the ones in pristine conditions
is still very valid: the level of GHI is close to the pristine one, with a difference of -0.8 W/m². At the scale of all situations



identified as clear-sun, the level of GHI is $8\,\mathrm{W/m^2}$ lower than under pristine conditions and $12\,\mathrm{W/m^2}$ above the cloud-free
level.

## 5    Conclusions

We perform in this study an analysis of the variabilities of global horizontal irradiance (GHI), its direct (BHI) and diffuse
(DHI) components measured routinely at 1 min resolution over the period 2010-2022, as well as coincident aerosol optical
properties measured by AERONET, with ground-based measurements from the ATOLL platform (Villeneuve d'Ascq, France),
in a region marked with extensive variabilities of sky conditions and aerosol loadings.

As the amount and property of the SSI is quite different depending on sky situations, as well as its sensitivity to atmospheric
parameters, we distinguish sky conditions by applying adapted filters based on irradiance's components that lead to three
categories: clear-sky (CSKY), clear-sun with cloud (CSWC) and cloudy-sun (CLOS) situations. Over 2010-2022, these sky
condition's occurrences represent 11, 22 and 67 % of the situations. The performances of this automatic filtering were validated
with visual analysis of around 12000 sky-imager's observation. The risk of misinterpretation is 5 and 12 % for clear sky and
clear sun (sum of CSKY and CSWC) situations, respectively, and higher for strong aerosol-load situations, which are rare in
our case. Climatologies of SSI and its components were obtained for these three situations, between sunrise plus 30 minutes
and sunset minus 30 minutes in order to minimize bias. CSWC situations, generally marked by irradiance enhancement effects
due to clouds (Gueymard, 2017; Mol et al., 2023), are particularly interesting with on monthly-average 13 % more GHI than
in CSKY situations (between +4 to +21 %) and diffuse proportions in CSWC situations on average 10 % above the mean pro-
portion in CSKY situations (31  against 21 %).

A second classification of sky's condition concerns the atmospheric column loading of aerosols. A six-class categorization
of aerosols, inspired by the work of Toledano et al. (2007), and based on photometric measurements of the aerosol optical
depth (AOD), serves a climatological purpose and allows the definition of new models of aerosol optical properties, based
on AERONET almucantar inversions. With this modelling, we perform radiative transfer simulations under cloud-free and
aerosol-and-cloud-free («pristine») assumptions that allow the computation of the direct radiative impact (DRE) of aerosols
and clouds in Lille over 2010-2022, as well as a sensitivity analysis of the solar irradiance to various input parameters, such as
the aerosol's optical depth and Ångström exponent, PWV (Precipitable Water Vapor content) or even SZA (Solar Zenith An-
gle). The computed sensitivities are used in conjunction with the above classifications to refine an analysis of the variabilities
of irradiance components observed in clear-sun conditions.

We focus on spring and summer seasons and analyse the coincident year-to-year variabilities over the period 2010-2022
of solar irradiances and sky conditions. The application of Mann-Kendall trend tests reveals for those seasons (significance's
threshold of 5%) statistically significant trends both for the occurrences of clear and cloudy conditions and AOD, and for the



global and direct components of the solar irradiance. Indeed, the average proportion of CLOS moments shows a decreasing trend of around -1 % per year for both seasons, in favor of a +0.5 % increased frequency of both CSWC and CSKY conditions. In addition, photometric measurements show a decrease in AOD over the period 2010-2022 for both seasons, particularly in spring, with an average trend of -0.008 per year. Finally, measurements of the incident solar radiation indicate significant upward trends in all-sky GHI for both seasons of around $+4 \pm 2$ W/m$^2$ per year, which are mainly driven by a rise in BHI of around $+4.4 \pm 2.3$ and $+4.7 \pm 1.9$ W/m$^2$/year, respectively in spring and summer. Additionally, in summer, significant positive trends are also observed for both GHI and BHI for clear-sky situations (+3.1 W/m$^2$/year), as well as under clear-sun-with-cloud situations (+3.7 and +3.8 W/m$^2$/year, respectively for GHI and BHI).

We conducted a multivariate analysis of the irradiance variabilities in order to quantify the different contributions to the variabilities. It is first employed to analyze the particular case of spring 2020, which exhibited record values for both clear-sky and clear-sun frequency (34 and 57 % compared to the spring average of 15 % and 37 %) and radiation (389 W/m$^2$ compared to the average of 334 W/m$^2$). The analysis indicates that nearly 89 and 83 % of the record increases in GHI and BHI, respectively, are attributable to exceptional sunlight conditions. This conclusion aligns with the results of Heerwaarden et al. (2021) that also identified the peak in spring 2020 irradiance in Cabauw (Netherlands) as primarily driven (also a 89 % contribution) by a lower cloud fraction. Our analysis reveals in addition that most of the increase, 71 %, in BHI observed in spring 2020 in clear-sun conditions is due to a decrease in the SZA (-1.5° and -0.9° in CSKY and CSWC situations respectively), rather than the yet observed change in aerosol properties (decrease by 0.06 of AOD and decrease of continental polluted class, in CSWC and CSKY situations respectively). These findings are consistent with the results of case studies and sensitivity analyses, which show that in the absence of clouds (clear-sky conditions), the SZA has the most impact on the global irradiance, followed by the AOD and PWV. The multivariate analysis of the temporal trends in global and direct irradiances reveals that the variability of sky conditions significantly influences the observed all-sky increases in spring and summer, with seasonal contributions for GHI of 71 and 62 % and 79 and 74 % for BHI, respectively. Our analysis also indicates a significant contribution in all-sky seasonal BHI (around 20 %) from the variability of the BHI in CSWC conditions for both seasons. The sensitivity analysis of BHI to atmospheric parameters indicates that while the increase in spring is mainly linked to the variability of the aerosol content (+88 %), and most notably to a decrease in AOD (-0.011 per year), it relates in summer to comparable contributions from the solar zenith angle (44 %) and aerosol content (49 %) variabilities.

With radiative transfer simulations of aerosol-and-cloud-free (pristine) conditions and of cloud-free conditions, the study of the direct radiative effects (DRE) of aerosol and clouds are possible in clear-sun situations. The quantification of the aerosol DRE in CSKY situations shows for our site a relatively modest average effect on GHI of -20 W/m$^2$ (-6 %), that results from an expected compensation between a negative effect on the direct (decrease three time larger: -61 W/m$^2$; -18 % ) component and a positive forcing on the diffuse (increase two time larger: +42 W/m$^2$;+92%) irradiance. The influence of aerosols on the solar environment is significant in CSKY situations as the proportion of diffuse irradiance is on average two times larger (24%) in the presence of aerosols than under pristine-like conditions (12%). The statistical results, as well as case studies, show that



the radiative impact of aerosols on irradiances is mainly driven by the AOD, so lower aerosol DRE are found in winter and during clean situations, and conversely higher aerosol DRE values are found in spring and for continental-polluted situations. It translates into DRE efficiencies (DREE) that are more case-independent than the DRE. In addition, results show that the DREE is larger for coarser particles, particularly on the diffuse irradiance.

The identification of CSWC situations that represent 22 % of the cases, offers the opportunity to quantify important compensation mechanisms between the effects of aerosols (unambiguously on BHI) that tend to reduce the SSI, and the effects of clouds (unambiguously on DHI) that tend to increase it. Our analysis reveals that clouds increase on average the diffuse, and thus the global irradiances, by about +27 W/m$^2$ (6%). This cloud enhancement effect compensates, or even overcompensates, especially in clean, summer or continental situations, the overall radiative impact of aerosols, resulting in GHI values comparable or even higher than in pristine conditions (+13 W/m$^2$ and +4 W/m$^2$ in clean and polluted situations), with a much higher diffuse proportion (25 % against 21 in cloud-free situations, 10 % with the pristine hypothesis).

We present in this work a methodology and its potential for analyzing the contribution of the variability of cloudy and clear sky conditions to changes in incident surface solar irradiances. A multivariate analysis reveals that the overall variability of the surface solar irradiance is influenced by a combination of factors whose impacts can sometimes accumulate (e.g. increase in clear-sun occurrence, reduction of aerosol loading and decrease of SZA) or compensate, and that simple geometrical effects are not to be neglected. Further investigations would concern clear-sun with clouds and cloudy-sun conditions, which are prevalent in Lille, in order to analyze the variability of the surface solar irradiance as a function of cloud cover parameters, including the fraction of the sky covered by clouds, the cloud optical depth and type.

*Code and data availability.* Python code to analyze data and generate figures is available from the first author upon request. The AERONET datasets were downloaded from the NASA AERONET website https://aeronet.gsfc.nasa.gov/. The ARTDECO original radiative transfer code was obtained from the ICARE Data Center in Lille, France (https://www.icare.univ-lille.fr/artdeco/). The dataset exploited in this study issued from ground measurements from the ATOLL (Atmospheric Observations in LiLLE, https://www.loa.univ-lille.fr/observations/plateformes. html?p=lille) platform in Villeneuve-d'Ascq, north of France, over the period 2010-2022, and containing both irradiance measurements, aerosol and gas column properties, obtained clear-sky and clear-sun flags, is freely available from an Easy System Data Repository in Chesnoiu et al. (2024).

*Author contributions.* GC: formal analysis, methodology, data handling, writing (original draft preparation). NF and IC: conceptualization, funding acquisition, supervision, methodology, writing (review and editing). FA and DC and IJ: help to operate and maintain the radiative sensors and the sky imager. MC and TE: collaborate on the design of SOLARTDECO, the solar version of the radiative transfer code ARTDECO. All authors have contributed to the final manuscript.



*Competing interests.* The authors declare no competing interests.

*Acknowledgements.* The authors would like to thank Gérard Brogniez who initiated in 2008 at LOA the observational dataset of global, direct and diffuse solar irradiances at 1-min resolution and Colette Brogniez who initiated the automatic acquisition of full hemispheric pictures of the sky with a sky-imager in 2009. The authors would like to thank also Philippe Goloub, PI of AERONET site of Lille and the personnel of the PHOTONS observation station for their effort in establishing and maintaining the AERONET Lille site.

This study is issued from the work of G. Chesnoiu during his PhD thesis financed by the ADEME and Région Hauts-de-France. Additional
1100 funds were granted under the LEFE (Les Enveloppes Fluides et l'Environnement)/IMAGO (Interactions Multiples dans l'Atmosphère, la Glace, et l'Océan) CNRS-INSU program over the period 2021-2023.



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
