# Peer review of "Influence of cloudy/clear-sky partitions, aerosols and geometry on the recent variability of surface solar irradiance's components in northern France"

_EGUsphere, 2024_

## Author Comment (AC1)

Dear Referee #1,

Thank you for your suggestions and remarks. Consideration of these comments have helped improve the manuscript. Below you will find the answers to each comment.

*In this study by Chesnoiu et al. the variability of surface solar irradiance and its components (direct and diffuse) is investigated for the period 2010-2022 over Lille, France, relying on ground-based measurements and radiative transfer simulations. Based on the classification of the sky conditions (for clouds and aerosols) they quantified the contribution of the different parameters on the variability and trends of the different solar irradiance components, and they obtained also climatologies for their site. The objectives of the study are quite straightforward and are addressed by a thorough analysis. The surface solar radiation climatology and trends that the authors provide here for Lille considering also the atmospheric parameters that can impact the calculated changes is of significance for assessing and understand changes in surface solar radiation. I consider the topic and results of this manuscript to fit the scope of ACP. However, I have some general and major comments (please see below 1-5) which should be addressed prior to publication.*

*1) My major comment concerning this study has to do with Section 4. Authors should rename the section, correct it and state clearer the objectives regarding this analysis. The analysis performed is a quantification of the direct (scattering and absorption) impact of aerosols in downwelling surface solar irradiance. Changes in downwelling surface solar irradiance due to aerosols calculated using eq. 29 are always negative (attenuation) due to their direct interactions of incoming solar radiation which is of relevance for surface related applications like solar energy as stated in the manuscript. The relative change (expressed in %) in downwelling surface solar irradiance due to aerosols presence was calculated with respect to an aerosol-free atmosphere using eq. 27. However, the radiative effect due to aerosol-radiation interactions REari according to IPCC report formerly known as direct radiative effect (DRE), is the change in radiative fluxes caused by combined scattering and absorption of radiation by anthropogenic and natural aerosols (Boucher et al., 2013). DREs are climate related quantities which are calculated at surface and top of the atmosphere (TOA) using net fluxes (downwelling minus upwelling) for shortwave and longwave radiation, for clear sky and all skies conditions, in order to assess the warming of cooling of the earth-atmosphere system. Authors should address which is the objective of this study, make the appropriate changes in section 4 including related references where applicable.*

*Boucher, O., D. Randall, P. Artaxo, C. Bretherton, G. Feingold, P. Forster, V.-M. Kerminen, Y. Kondo, H. Liao, U. Lohmann, P. Rasch, S.K. Satheesh, S. Sherwood, B. Stevens and X.Y. Zhang, 2013: Clouds and Aerosols. In: Climate Change 2013: The Physical Science Basis. Contribution of Working Group I to the Fifth Assessment Report of the Intergovernmental Panel on Climate Change [Stocker, T.F., D. Qin, G.-K. Plattner, M. Tignor, S.K. Allen, J. Boschung, A. Nauels, Y. Xia, V. Bex and P.M. Midgley (eds.)]. Cambridge University Press, Cambridge, United Kingdom and New York, NY, USA.*

To improve clarity, we followed the recommendations of the reviewer both to change the title of Section 4 and to favor the use of "radiative effect" instead of 'direct radiative effect' in the manuscript. Thus, in the revised manuscript the new title of Section 4 is "Radiative effects of atmospheric components on SSI". Section 4.1 has been renamed "Clear-sky conditions" and Section 4.2 "Clear-sun-with-clouds conditions".

In addition, the acronym "DRE" has been replaced by "$RE_d$" (*R*adiative *E*ffect on the *d*ownwelling surface solar irradiance). These changes aim to avoid any confusion with the climate related quantity that indeed consider the net radiative effect, generally both at the surface and the top of the atmosphere.

Paragraphs introducing Section 4 (lines 782 to 804) have been rewritten to clarify our objectives and approach (lines 736 to 748 of the revised manuscript):

"In this section we assess atmospheric component's, especially aerosols, radiative effects on the downwelling surface solar irradiances (called $RE_d$ for *R*adiative *E*ffect on the *d*ownwelling SSI), consistently with our ground-based dataset, and in relevance for surface related applications (such as photovoltaic solar systems) and surface processes (e.g. photosynthesis). Note that only a few studies have assessed the direct radiative impact of aerosols specifically on downward surface solar radiation, as for example done in Papadimas et al. (2012) over the Mediterranean basin. Furthermore, our approach encompasses the shortwave radiative effects of atmospheric particles on both global (GHI), direct (BHI) and diffuse (DHI) components of the downwelling SSI, as in Witthuhn et al. (2021). In Section 4.1 our analysis of the $RE_d$ focus on clear-sky conditions, through a statistical investigation of the aerosol radiative effects on all downwelling SSI components (GHI, BHI, DHI) over the whole period 2010-2022. In addition, Section 4.2 provides an analysis of aerosols and clouds radiative effects in CSWC conditions. This approach relies on two sets of pristine (i.e., aerosol-and- cloud free) and cloud-free simulations, which by comparison with ground-based measurements that include the effect of clouds on SSI, allow the quantification of both aerosols and clouds' $RE_d$ in CSWC conditions. Finally, the respective and cumulative $RE_d$ of clouds (on DHI) and aerosols (on DHI and BHI) can be quantified over all CSUN (CSKY and CSWC) situations, that represent on average 33% of sky conditions in Lille."

*2) It is stated at Lines 273-274 that AOD440, AOD550, AE440-870 and relative humidity are the "remaining inputs" to SOLARTDECO. I think that this gives the wrong impression that these are the inputs in RT, while these values are used to select and adjust the new optical properties for the specific aerosol mixture. Please clarify and adjust 2.3.1 and 2.3.2 accordingly. For example, in Lines 343-344 it should be clarified that it is not only the fact that Mie calculations are time consuming that were not performed for every RT simulation, but the fact that inversion data are not always available, which has already stated earlier in the manuscript. In addition, while it is described in detail how the new aerosol optical properties are derived for each simulation, it is not clear the vertical structure of the aerosol layer (only notes in Lines 654-655 and 672-673) and how the extinction is scaled to measured aerosol load in the total column. Please clarify which are the parameters to be set (are those in Table 4?) apart from defining the new aerosol optical properties based on the Mie calculations. Another confusing part is Lines 383-384. SSA, g and ff are "estimates of optical properties derived from SOLARTDECO"? How this information is consistent with what is stated in Sections 2.3.1 and 2.3.2 regarding the inputs in RT?*

In order to clarify all these aspects describing our use of the radiative simulations with SOLARTDECO, Sections 2.3.1 and 2.3.2 have been extensively modified.

The classification of aerosol properties, which was initially in Section 2.3.2, has been moved to Section 2.2 ("Classification of atmospheric conditions in Lille"). Section 2.2 now includes classifications of both the sky conditions (2.2.1) and aerosol conditions (2.2.2).

The description of SOLARTDECO (Section 2.3) has been improved: a flux diagram has been added for clarity (new Figure 2 in the revised manuscript), especially on the input parameters.

The motivation behind the pre-computed aerosol optical properties has been rewritten more clearly in Section 2.3 (lines 256-259).

The vertical structure of aerosols has been defined more carefully (exponential decay of the aerosol density with a 2 km scale height, lines 286-287), and the definition of the scaling of the extinction to the measured aerosol load has been added to the main text (Section 2.3, lines 279-284).

It has been made clear that the estimates of aerosol optical properties are not products of SOLARTDECO but rather outputs of a related routine, which mixes pre-computed aerosols optical properties of the fine and coarse modes to produce inputs (Cext, SSA, phase function) for DISORT (lines 254-256). Lines 383-384 appeared awkward and redundant and have been removed.

*3) Consider to move 3.2.1 and 3.2.2 to the methodology Section.*

This change in the organization of the paper has been done in the revised version. Sections 3.2.1 and 3.2.2 are now included as part of Section 2 "Data and Methods", in Section 2.4 titled "Multivariate analysis of the SSI variability". Section 2.4.1 describes the methodology, Section 2.4.2 the sensitivity study of clear-sky SSI.

**Specific comments**

**Line 10: Change "all sky" to "all skies" throughout the manuscript.**

This expression is widely used to describe all skies situations, encompassing both cloudy and clear conditions (Boers et al., 2017; Wild, 2009; Xie et al., 2016). In the present work, our approach is similar, although the distinction involves three categories of sky conditions (clear-sky, clear-sun with clouds and cloudy-sun) instead of two (cloudy or clear).

In the submitted version line 10 of the abstract, the "-" was missing in the "all-sky" expression. This has been corrected in the revised manuscript. But the use of this expression is adopted and rather conventional in earlier publications, as that of "Clear-sky", or "Clear-sun-with-cloud".

References:

Boers et al. (2017): Impact of aerosols and clouds on decadal trends in all-sky solar radiation over the Netherlands (1966–2015). (DOI: 10.5194/acp-17-8081-2017)

Wild (2009) : Global dimming and brightening: A review. (DOI: 10.1029/2008JD011470)

Xie et al. (2016): A Fast All-sky Radiation Model for Solar applications (FARMS): Algorithm and performance evaluation. (DOI: 10.1016/j.solener.2016.06.003)

**Line 47: I am not following those "increase" and "decrease" descriptions inside the parenthesis where are referred to?**

There was a mistake in the descriptions inside the parentheses. Aerosols and clouds lead to a decrease in direct radiation while increasing the diffuse component.

For clarity, the sentence was changed to:

"*However, depending on their optical properties, aerosols and clouds influence incident radiation by reducing the direct component while enhancing the diffuse component.* "

**Line 49: I suggest including also here the importance of direct normal irradiance to concentrating solar power systems providing also references (e.g. Sengupta et al., 2021, https://www.nrel.gov/docs/fy21osti/77635.pdf)**

Thanks for this suggestion, the provided reference has been added in the revised version of the manuscript.

**Lines 271: Please remove the 500 in "of respectively 407 and 209 500 ppmv".**

The concentration of oxygen has been changed from 209 500 ppmv to 209 000.

**Lines 273-274: These sentences are confusing, regarding the geometry. The only geometry to be determined is the sun position through solar zenith angle and regarding viewing angle the whole dome is considered, right? It is stated explicitly in Line 283 that the "horizontal irradiances" were calculated, so please clarify if the other geometries are important at this part since no computations for tilted surfaces were performed.**

Indeed, in this work only horizontal surface irradiances were computed by integration over the whole dome, with only the solar zenith angle as a geometrical input.

Nonetheless, SOLARTDECO is also able to compute radiances for specific viewing zenith and azimuth angles.

However, as such computations were not involved in the present study, we chose for clarity to remove the mention of geometrical inputs other than the SZA.

**Line 281: Please consider the change from "of the incoming and outgoing spectral irradiances" to "of the incoming and outgoing spectral solar irradiances"**

The adjective "solar" was added.

**Line 367: These are absolute differences? Please, clarify.**

The sentence was not quite clear and the indicated threshold was not correct.

It was changed as follows, lines 306-310 of the revised manuscript:

"Moreover, the performances of SOLARTDECO are well within the error margins expected for network-operational instruments (Meteorological Organization, 2008) **as mean absolute differences (MAD) are close to the resolution of network instruments (5 W/m²)** for all irradiance components and more than 95% of the comparisons have **mean differences** lower than ±20 W/m²."

**Line 439: Is 435 W/m2 the correct number for clear-sky? The grey line is below 400 in Fig. 4 (h).**

Indeed, 435 W/m² corresponds to the mean climatological value obtained by averaging all 1-minute measurements over the period 2010-2022. For consistency, we modified the text to fit Figure 4h, which reports the averages computed based on the monthly values.

The modified sentence is : "In comparison, the measured surface flux in clear-sky conditions is twice as high, with an average value around 375 W/m2." (lines 506-507 of the revised manuscript).

**Line 451: Is 69% correct in summer for all-sky conditions according to Fig.4 (e)?**

The diffuse proportions of irradiance were not consistent with those shown on Fig. 4e due to differences in averaged calculations, as explained in the previous comment. The text has been updated to match Fig. 4e: the averages are computed based on the monthly values. The new sentence is : "Moreover, the variability of the optical air mass has a great influence on the proportion of diffuse irradiance, which varies, under all-sky conditions, between 51% in summer to more than 65% in winter (lines 518-520 of the revised manuscript).

**Lines 653-654 and 658-662: It would be helpful to provide also those mean values used as refence.**

| SZA (°) | $AOD_{550}$ | $SSA_{550}$ | ff | $H_{aer}$ (km) | RH (%) | PWV (cm) | $O_3$ (DU) | $O_2$ (ppmv) | $CO_2$ (ppmv) | $\alpha$ |
|---|---|---|---|---|---|---|---|---|---|---|
| 61.71 | 0.13 | 0.93 | 0.72 | 2 | 55 | 1.46 | 341 | 209 000 | 407 | 0.15 |

As shown line 405 of the revised manuscript, the above table describing the mean values used for each parameter involved in the sensitivity study has been added to the supplements (new Table S1).

**Line 660: Remove "logically".**

"Logically" has been removed (line 413 of the revised manuscript).

**Figure 7: Last 3 lines of caption need to be clarified better.**

The caption was clarified as follows:

"The additional decomposition of $BHI_{CSWC}$ and $BHI_{CSKY}$ with respect to the scene's parameters is illustrated in panel (b). The orange and blue columns from panel (a), which represent the intrinsic variability of the BHI under CSWC and CSKY conditions, respectively, are represented as dashed lines of the same color in panel (b). The pink dashed lines represent the values estimated from the decomposition of the BHI according to the scene's parameters as in Equation 6. The latter decomposition is also illustrated by the colored columns, which represent the contributions of the variability in the frequency of occurrence of aerosol classes ($freq_{aer}$, black column), as well as the variability of the AOD (green column), ff (violet column), PWV (blue column), and SZA (brown column)."

**Line 724: FCSUN here is BHICSWC?**

$F_{CSUN}$ corresponds to the sum of $BHI_{CSWC}$ and $BHI_{CSKY}$ from Figure 7.

For clarity, "[…] overall increase of $F_{CSUN}$." has been replaced with "[…] overall increase in BHI under clear-sun conditions." as shown line 679 of the revised text.

**Technical corrections**
**Line 386: Replace 2020 with 2010.**

Done.

**Figure 4: In (e), (f) and (h) percentages that reflect the contribution of the DHI to the overall mean yearly GHI are missing. For (b) and clear-sky this is "blue line" or green? In addition in the 6$^{th}$ line AERONET is twice.**

Percentages have been added in Figures 4e, f, and h.

For panel (b), the color of clear-sky conditions is indeed green. The caption of Figure 4 has been modified accordingly in the revised manuscript.

The first « AERONET » of the sixth line has also been removed.

**Lines 507-508: Change the color of the lines insides parenthesis according to Fig. 5b and 6b.**

Colors mentioned inside parenthesis have been modified, as shown lines 577-578 of the revised text.

**Line 623: Replace "dFclear/dt" with dFi/dt**

The subscript was changed accordingly, as shown line 375 of the revised text.

**Line 626: In eq. 22 probably this "Fclear" is F?**

« Fclear » is indeed F, the subscript has been removed (eq. 8 of the revised manuscript, line 378).

**Line 791: Word "surface" is twice**

One iteration of "surface" has been replaced by "solar", although the text of this paragraph has been modified.

***Figures 7, 8, 9: Should be enlarge and brown columns better solid than shaded.***

The size of each figure was modified while keeping panels (a) and (b) on the same page.

Brown columns are now solid.

---

## Author Comment (AC2)

*Review of paper "Influence of cloudy/clear-sky partitions, aerosols and geometry on the recent variability of surface solar irradiance's components in northern France" by G. Chesnoiu et al.*

Dear Referee #2,

Thank you for your suggestions and remarks. Consideration of these comments have helped to improve the manuscript. Below you will find the answers to each comment.

*The paper is dedicated at assessing the role of aerosol and clouds in modulating diffuse and global solar irradiance components at Lille.*

*Identifying different cloudiness conditions, separating and quantifying effects have been and are challenging objectives.*

*The combination of selection methods which use measurements of direct normal and diffuse horizontal irradiances are applied to identify different cloud and aerosol conditions.*

*Based on the applied selection methods, the authors assess the role of clouds and aerosol on variability and modulation of global and direct horizontal irradiances. Radiation transfer calculations, and simplified schemes for aerosol characterization based on AERONET observations are used to assess the role played by different parameters in the solar radiation modulation. Inter-annual variability, especially in spring and summer, is also investigated together with main driving factors.*

*The paper is very long and incorporates many different methodological aspects, data, and results. The results are interesting, and the separation of the different conditions and factors is worth the attempt. However, the description of some of the applied procedure would need a clearer presentation and some assumptions need to be better justified and discussed. The impact of the implied uncertainties should also be assessed and discussed.*

*As said, the paper is very long and addresses different topics. The authors might consider focussing on a smaller number of topics, and/or separating the content into two different papers.*

We recognize that the submitted manuscript was relatively long, albeit not excessively. It is for example very comparable in size with Witthuhn et al (2021) accepted in ACP, and the manuscript comprises numerous figures. However, in the revised version, we made a significant effort to reduce the length of the paper, avoiding redundancies, and reorganize it as will be described below.

The paper addresses indeed several topics : climatology of SSI per sky condition, climatology of column aerosol loading, sensitivity study of SSI to aerosol properties, co-analysis of SSI and sky-content variabilities, and quantification of the radiative effects of both aerosol and clouds in clear sun conditions. We believe that these developments build overall a consistent analysis that should not be split in two parts for the purpose of this work. The objective is to describe a methodology and its results to analyze the variability of SSI per sky conditions, disentangling and ranking

different contributions that explain the observed SSI variabilities, and providing a climatology of radiative effects while analysing interesting compensation mechanisms. Most of the developments presented in the manuscript are necessary and complementary to reach the main objective of the paper. Furthermore, it necessitates the technical description of our methodology and tools. This results in an overall not short publication.

We made efforts to reduce the main text and to make clearer the sections and our results. We hope the resulting revised paper is more digestible. Overall, the length of the main text (up to "Code and data availability") has been reduced from around 1082 lines (46 pages) to 970 lines (42 pages).

The list of the changes made to reduce the length of the article are the following, in order of importance:

- Section 2.2.1 (Description of the cloud-screening methods) was moved to the appendices (Appendix A) as the methods described in this section are not original and are already described more thoroughly in the corresponding articles.

- The text of Section 2.2 was revised to avoid redundancies and limit superfluous information.

- Section 2.3 (Radiative transfer simulations) was also thoroughly modified for concision and clarity.

- Section 2.3.2 was moved to Section 2.2 (now called "Classification of atmospheric conditions") for clarity. The description of the aerosol classification was revised, notably, Table 2 now includes the definition of each class (previously in the text) and Table 2b was removed for concision.

- Figure 2 was also moved to the supplementary materials, and is Figure S1 in the revised manuscript.

- The description of the methodology used to define aerosol optical properties of fine and coarse modes (previously in Section 2.3.2) has been moved to the appendices (Appendix B).

- Section 3.1 has been revised to avoid redundancies.

- The methodology (Section 3.2.1) and associated sensitivity study (Section 3.2.2) of the multivariate analysis (Section 3.2) have been moved to Section 2 in response to a comment of the first reviewer.

- Section 4, especially the introduction, has been revised for concision and clarity.

- Section 4.1.2 (clear-sky case studies) has been removed completely as it was judged too long and of relatively low added value as it stands for the present study. Thus, Figure 10 of the submitted version has been removed in the revised manuscript.

- Section 4.3, regarding the mean radiative effect of aerosols and clouds for all clear-sun (i.e. CSKY and CSWC) situations, has been merged to Section 4.2 (radiative effects in CSWC conditions) as a conclusion and revised for concision and clarity.

- The term "surface solar irradiance" was mentioned more frequently as "SSI" to reduce the length of titles and sentences, also the use of the acronyms, once defined, CSKY, CSWC, CLOS was generalized.

- Figure 1 was modified to reduce superfluous information and size.

*Specific details are given below.*

*l. 101: what is intended with "relatively stable"? Please, add information on the calibration changes, and if any calibration correction or other adjustement was applied. This is important information when a long term data record is being examined, such as it is the case of this paper.*

We clarified lines 113-114 of the revised manuscript (changes are in bold text):

"Moreover, both instruments were calibrated in 2012, 2017, and 2022 and **new calibration factors were applied after each calibration, with differences in calibration factors lower than 3\%.**"

*Section 2.2.1. In my opinion, the presentation of the applies algorithms should be improved. Maybe, the addition of a flux diagram might help.*

*In particular, the method used to seperate clear sky and clear sun conditions should be better justified and discussed. This is at the basis of most of the following analyses and requires a stronger verification. In particular, the method by Batlles et al. (2000) is correctly classified by Guyemard et al. (2019) among those for "cloudless sky" (clear sky detection methods of the first kind), which are based on solar irradiance measurements. Thus, the use of the difference between the cases selected by Batlles et al. (2000) as "clear sun" (l. 216) requires in my opinion a specific discussion and justification. The cases falling in category CSWC also strongly depend on the accuracy of this algorithm. The need of a stronger justification is also supported by the relatively poor performance found in section 2.2.2 (e.g., 67% precision for the Batlles et al. method). Maybe, the adoption of a "clear sun" selection method based on DNI measurements (see, e.g., Guyemard et al., 2019) might be explored.*

Reviewer 2 is right, the method and its results, used to distinguish clear-sky, clear-sun and cloudy-sun conditions should be better discussed in the manuscript.

It should be underlined, as mentioned by Gueymard et al (2019), that "all methods have obvious strengths and weaknesses", hence the choice of a cloud screening method is always a compromise between accuracy and efficiency.

The choice of the Garcia et al (2014) algorithm has been already justified and discussed in a parallel publication (Elias et al 2024, AMT). Regarding the method of Batlles et al (2000), this method, as stated by Gueymard et al. (2019), was indeed initially intended for the detection of completely clear skies based on hourly data. However, as stated lines 178-179: "the study of Gueymard et al. (2019) showed that on a 1-minute basis it performs better for the identification of clear-sun moments with results among the best performing methods. » Table 6 of Gueymard et al (2019) shows in particular that according to their new index, the method of Batlles et al (2000) appears to be the second best performing method for clear-sun detection at a 1-minute resolution, despite its initial aim as a clear-sky detection method.

The best clear-sun detection method according to Table 6 of Gueymard et al (2019) is the one of Ineichen et al (2009), which is based on only one empirical equation. The choice of the Batlles method is thus motivated by its good performances, its high portability, as well as the use of both GHI and DHI measurements compared to the method of Ineichen et al. (2009).

Further inspection of Batlles False Positives from our analysis notably shows that most misidentifications are linked to the presence of cloud edges or clouds of low atmospheric optical depth in the sun direction, such as Cirrus scenes or small isolated clouds, which cannot always be identified by the filtering method due to their relatively limited impact on the Direct Normal Irradiance (DNI). This is supported by the fact that for most Batlles False Positive cases (1352 out of 1356), the DNI is greater than the WMO sunshine threshold of 120 W/m$^2$, and the inverted atmospheric optical depth from DNI ($\tau_{inverted} = \cos(SZA) \times \log(\frac{I_{TOA}}{DNI})$) is on average 0.37, with a maximum of 0.7, indicating that these scenes are sunny and can be considered as quasi clear-sun. These results also suggest that the adoption of a "clear-sun" selection method based on DNI measurements only should not really improve the identification of CSUN conditions.

Here are two examples, for the 05/26/2018, of scenes misidentified by both the AERONET cloud-screening and Batlles methods:

[Figure]

[Figure]

Moreover, it should be noted that, in the present study, the method of Batlles et al. (2000) is not only used to identify clear-sun moments but also importantly to distinguish clear-sun moments from cloudy-sun moments. While the method of Batlles et al. (2000) performs far from perfectly for the identification of pure clear-sun conditions (i.e. high FP score), its ability to identify cloudy-sun moments is particularly satisfactory with a FN score of merely 0.8%, yielding a confidence level ($\frac{TN}{TN+FN}$) of 98.8%.

The overall precision of the Batlles et al. (2000) in distinguishing both clear-sun and cloudy-sun conditions is illustrated by the "risk" ($\frac{FP+FN}{FP+FN+TP+TN}$), as it represents the percentage of "risk" of the algorithm misidentifying the scene. According to Table 1, the overall risk of the Batlles algorithm misidentifying the scene is of 12%. A parallel analysis of the performances of the AERONET cloud-screening method, based on level 1.0 ("raw") and 2.0 (cloud-screened and quality checked) in Lille, compared to our manual inspections of sky images over January and May 2018 shows similar performances in distinguishing clear-sun and cloudy-sun conditions, even if the AERONET cloud-screening method is better at identifying clear-sun conditions, as summarized in the table below. Note that in this case the overall sample is reduced due to the lower frequency of AERONET observations, which leads to a total of 1916 coincident AERONET measurements and sky images, against 12403 points for irradiance data.

| | TP | TN | FN | FP | Precision | Risk |
|---|---|---|---|---|---|---|
| | Filter → clear Obs → clear | Filter → cloudy Obs → cloudy | Filter → cloudy Obs → clear | Filter → clear Obs → cloudy | $\dfrac{TP}{TP+FP}$ | $\dfrac{FP+FN}{FP+FN+TP+TN}$ |
| **Clear-sun (AERONET)** | 1176 (61.2%) | 495 (25.8%) | 122 (6.4%) | 123 (6.4%) | 91% | 13% |

Clarifications and justifications on the choice of both methods have been added as follows lines 172-177 of the revised text:

"Both methods were evaluated in Gueymard et al. (2019) and were found to perform particularly well for these distinctions on a 1-minute basis. Moreover, the method of García et al. (2014) relies on the popular detection scheme developed by Long and Ackerman (2000), and does not require parallel clear-sky simulations. It also relies on collocated AOD information, which should improve the detection of clear skies under higher aerosol loads compared to the initial method of Long and Ackerman (2000). Both methods used in this study are described in more details in Appendix A."

Moreover, the description of both methods has been moved to the appendix section to reduce the length of the main text.

**Section 2.2.2. Also with respect to the previous comment, results of the performance analysis need more discussion.**

Lines 233-239 of the submitted manuscript (now lines 191-205 of the revised text) have been modified as follows (changes are in bold letters):

"The results are presented in Table 1 for both clear-sun and clear-sky conditions simultaneously. For the clear-sky detection, the revised method of García et al. (2014) gives satisfying results with a low FP score (2.7%) compared to the TP score (12.5%) which leads to an overall good precision of 82%. The latter score highlights that 82% of the time the sky is correctly identified as clear by the algorithm. **This method also presents a relatively low FN score (2.6%), suggesting that it is also able to accurately isolate cloudy moments (either CSWC or CLOS). This leads to overall satisfactory performances for the distinction of clear-sky and cloudy conditions, with a risk of only 5% to misidentify sky conditions.**

**For the identification of clear-sun conditions, the method of Batlles et al. (2000) produces satisfactory but less optimal results. The method shows a strong ability in identifying cloudy-sun moments (FN score of 0.8%) and shows an overall satisfactory risk of only 12% for the distinction between clear-sun and cloudy-sun conditions. However, its FP and TP scores are closer compared to the Garcia method and thus its precision in correctly identifying clear-sun conditions is lower (67%), which might lead to a slight overestimation of the proportion of clear-sun situations.**

**Nonetheless, further analysis indicates that the Batlles FP cases are often due to the presence of cloud edges or clouds of low atmospheric optical depth in the Sun's direction, such as Cirrus scenes or small isolated clouds, and that most of the time, 1352 out of 1356 FP cases, DNI values are greater than 120 W/m². These FP cases, while not rigorously clear-sun cases, are thus almost entirely sunny moments (sunshine threshold defined by the WMO) and can be considered as quasi clear-sun.**

**In addition, a similar analysis of the performances of the AERONET cloud-screening method over January and May 2018 further comforts the results of the Batlles method, as it highlights comparable performances in distinguishing clear-sun and cloudy-sun conditions, with a satisfactory risk of 13 %. "**

*The performance analysis was carried out considering only two months of 2018. Was there a reason why January and May were chosen?*

As the manual inspection of sky images, one by one, is rather tedious, it was limited to only two months. The months of January and May were chosen as they represent seasons with contrasted meteorological conditions and SZA ranges. The year 2018 was chosen as it closely follows the 2017 calibration of the pyranometer and pyrheliometer.

*Secondly, the selection obtained for the Batlles et al. (2000) method (clear sun) does not appear to produce very convincing results: there is about 13% of wrong determinations. How this is taken into account in the following analyses? How much it may impact the obtained results on cloud forcing in the different situations?*

Although the 12% risk of wrong determinations of the Batlles method may seem relatively high, as mentioned earlier, a parallel analysis of the performances of the AERONET cloud-screening method, based on level 1.0 ("raw") and 2.0 (cloud-screened and quality checked) in Lille, compared to our manual inspections of sky images over January and May 2018, shows similar performances in distinguishing clear-sun and cloudy-sun conditions, with a risk score of 13%. Hence, we believe that the results of the Batlles et al. (2000) cloud screening method are quite convincing in Lille.

Furthermore, it should be noted that the low precision of the Batlles method in identifying clear-sun conditions has a limited impact on our analysis.

Indeed, regarding aerosol and cloud radiative effect computations in CSWC conditions, the occurrence of clouds in the direction of the Sun should be very small as we only compute radiative effects of clouds and aerosols for moments coincident with AERONET observations, which are also cloud-cleared in the direction of the Sun, with a 91% accuracy (see the previous table).

The precision of the Batlles method thus could only eventually impact our climatological results (frequency of occurrence, mean irradiances, etc.). However, given the good precision of the Garcia et al (2014) method in identifying clear skies and the high precision of the Batlles method in distinguishing cloudy-sun conditions, we can assume that the impact of the Garcia and Batlles misclassification is relatively limited on our statistics. This impact is not easily quantifiable with our database, and thus the influence of wrong determinations is not taken into account within our study. Nonetheless, we added in Section 3.1.2 a few cautionary sentences regarding the observed trends in the occurrences of sky conditions (lines 555-561 of the revised text):

**"These trends are issued from only 13 years of data, and rely on the imperfect filtering of sky conditions. Their scope should thus be considered with caution as they are quite sensitive to year-to-year variability, and as uncertainty in the occurrences of sky conditions impact their validity. However, it is worth noticing that the strongest (significantly decreasing) trend in CLOS situations was found to be associated with a high confidence in the related occurrences (1.2% of misidentifications) and that this observation is consistent with results from CM SAF data (C3S,**

**2024) that show since 2010 repeated negative annual anomalies in cloud cover and positive ones in sunshine duration, over European land areas and relative to the reference period 1991-2020."**

*As a test case, I would suggest making calculations for January and May, 2018, based on classifications obtained from the visual analysis of sky imager data, and comparing results with those obtained by the automatic algorithms. This might provide an indication on the possible impact of the clear sun selection method.*

Although not clearly stated, we understood that this comment relates to radiative effect computations. Hence, in the following response, no comments are made on the potential of a climatological and multivariate analysis of January and May 2018, based on classifications obtained from the visual analysis of sky imager data.

As stated in response to the previous comment, the influence of the cloud-screening method on the aerosol and cloud radiative effect computations in CSWC conditions should be very small as we only compute radiative effects of clouds and aerosols for moments coincident with AERONET observations, which are also cloud-cleared in the direction of the Sun, with a 91% accuracy.

Hence, we believe it is not necessary to make additional calculations of aerosol and cloud radiative effects for the specific timestamps of the visually analyzed sky images of January and May 2018.

*Section 2.3.2. The presentation of the used aerosol models is not very clear, and a more detailed description is needed. Various aspects appear confuse to me, and need additional clarification. The definition of some variables is lacking.*
*I am listing below some questions that in my opinion need clarification; however, some of these may be due to the limited understanding of all the steps used in the method I could obtain after reading the section several times. The author should state clearly when AERONET measured values and/or simulated data are used in the different steps, and what parameters are used in the simulations.*

*Also in this case, maybe a flux diagram may help understanding the applied procedures.*

A flux diagram has been added to the main text for clarity (new Figure 2 of the revised version).

The description of SOLARTDECO in Section 2.3.1 has been largely revised. For clarity, the description of the mixing of pre-computed aerosol optical properties (Section 2.3.2 of the submitted manuscript) has been moved to the appendix section (Appendix B), and the description of the aerosol classification has also been moved prior to the description of SOLARTDECO (Section 2.2.2 of the revised manuscript).

*It is not clear to me how the different size distributions are chosen starting from the AERONET inversions of cases falling in the 60 different classes. Do the authors take an average distribution over those retrieved from AERONET observations in the corresponding class?*

Indeed, the size distributions used to pre-compute the aerosol optical properties defined in SOLARTDECO were averaged over the available number of AERONET inversions, which are divided into 60 bins (6 classes of aerosols times 10 classes of relative humidity).

Each bin includes a mean total refractive index, as well as two mean radii and standard deviations, which are used to describe the size distribution of each mode (fine or coarse).

This allows the pre-computation of 120 models of aerosol optical properties based on the 60 bins and associated fine and coarse modes.

This is now detailed as follows in the main text of Section 2.3.1 (lines 263-267 of the revised manuscript):

"For each subset of the look-up table, the size distribution is divided between two modes (fine or coarse). Then, datasets of $Cext_\lambda$, $SSA_\lambda$ and phase function components are computed for each mode through Mie calculations, based on the normalized mean number size distributions (i.e. number size distribution of each mode divided by the corresponding number concentration) and the total mean complex refractive index."

*I also do not understand the use of ff and cf in equations 6 and 7. If ff is the fine mode fraction applied to the AOD (eq. 6), in my opinion the same factor can not be applied in the same way to AE (eq. 7). By combining a given fine and coarse mode fraction of AODfine and AOD coarse produces an AEtot which is different from ff AEfine + cf AEcoarse.*

Equations 6 and 7 form a system of two equations with two unknowns. The weights ff and cf are thus computed using both measurements of $AOD_{440}$ and AE, and not only the AOD. Hence, ff and cf can be used to reproduce both $AOD_{440}$ and AE measurements based on the fine and coarse mode properties.

As mentioned in the response to the quick reports, we understand that the present ensemble of Equations 6 and 7 forms an unconventional and, in appearance, puzzling system of equations as the same weighting coefficients are used for different physical quantities, aerosol optical depth (AOD) and Angström exponent (AE), with the latter defined as a logarithm of AOD ratios. The chosen system of equations forms however a solvable system of two independent equations with two unknowns that provides the solution of the weights "ff" and "cf", that represent the decomposition of aerosol's observations on a fine and a coarse aerosol mode.

Note that a comparative study with a more classical system, involving AOD at 870 nm instead of AE in Equation 7, has also been performed to assess the impact of using AE.

Overall, as shown in the figure below, the results using $AOD_{870}$ measurements instead of AE are very similar to those presented in Figure 3.

[Figure]

Hence, both systems appear as valid choices for our study.

The current system presented in the study, involving the Angström exponent, was chosen as it relies on the same inputs as the classification of aerosol optical properties.

The figure above (issued from the use of $AOD_{440}$ and $AOD_{870}$) was added to the supplementary materials to relieve any doubts one might have regarding the good performances of our methodology. It is cited as follows in Section 2.3.1 (lines 277-279 of the revised manuscript):

"Furthermore, proxy simulations were also conducted with another parametrization, based on a more common system involving the measured $AOD_{440}$ and $AOD_{870}$. Comparisons with ground-based measurements, similar to those presented for the chosen system in the following section, show very comparable results (Figure S2)."

*Also, please specify how Cext and Csca are derived (I assume they are calculated for the specific fine and coarse size distribution and refractive index identified for each class) and calculated; is the aerosol number concentration included in the formula? How is it derived? NCfine and NCcoarse are derived from AERONET inversions (l. 339). If I understood well, they were derived using the size distribution retrieved by AERONET, which may be different from the average ditribution assumed for the specific class in the Mie calculation for the LUT. In this case, the use of AERONET values for NCfine and NCcoarse in the calculated optical properties may produce incorrect results.*

Indeed, using $NC_{fine}$ and $NC_{coarse}$ for both Mie calculations and mixing of the fine and coarse modes would produce incorrect results. However, our Mie calculations do not involve $NC_{fine}$ and $NC_{coarse}$, as we use normalized mean number size distributions. Instead, $NC_{fine}$ and $NC_{coarse}$ are only used as in Equations 10 and 11 to produce specific mixing ratios for each simulations.

Outputs of the Mie computations include $C_{sca}$, which is necessary to derive the single scattering albedo.

Regarding $NC_{fine}$ and $NC_{coarse}$, these quantities are derived from AERONET inversions along with the mean radius and standard deviation of the size distribution, and each of the 60 subset of our look-up table includes an average value of $NC_{fine}$ and $NC_{coarse}$.

It is now clearly stated in Section 2.3.1, lines 263-275 (clarifications associated to the above question are highlighted in bold text):

"For each subset of the look-up table, the size distribution is divided between two modes (fine or coarse). Then, datasets of $Cext_\lambda$, $SSA_\lambda$ and phase function components are computed for each mode through **Mie calculations, based on the normalized mean number size distributions (i.e. number size distribution of each mode divided by the corresponding number concentration) and the total mean complex refractive index**. Overall, the look-up table includes 120 datasets, divided between the two modes, six classes and ten relative humidity bins. These datasets are used to compute the total aerosol optical properties needed for each radiative transfer simulation at the resolution of AERONET direct-sun measurements by mixing the pre-computed properties of the fine and coarse modes selected based on $AOD_{440}$, $AE_{440-870}$ and surface relative humidity measurements. The methodology adopted in the present study to compute the total aerosol optical properties is described in details in Appendix B. It involves solving a system of two equations with two unknowns, for each simulation, based on the measured values of $AOD_{440}$ and $AE_{440-870}$ with the additional use $AOD_{550}$ data. **The two coefficients derived from this system are then used, jointly with the mean number concentrations of the fine and coarse modes derived from AERONET size distributions, to mix the pre-computed optical properties of the two modes specifically for each simulation.**"

**l. 383-385: I could not understand what aerosol optical properties were in the end used in the simulations. Is the method of section 2.3.2, or a different set of values (see also l. 275-278: please, explain how this is made)?**

As stated in the main text of the revised version (Section 2.3.1, lines 254-256): "Regarding the absorption and scattering of solar radiation by aerosols, SOLARTDECO uses a look-up table of extinction coefficients ($Cext_\lambda$), single scattering albedo ($SSA_\lambda$) and components of the phase function ($P_{11}$,$P_{21}$,$P_{34}$ and $P_{44}$), which are used as inputs for radiative transfer simulations."

The proper inputs of the radiative transfer simulations are $Cext_\lambda$, $SSA_\lambda$ and phase function components. However, SOLARTDECO does require additional inputs such as $AOD_{440}$, AE and $AOD_{550}$, which are used to mix the pre-computed optical properties of the aerosol fine and coarse modes for each simulation, as described in Section 2.3.2 (now Appendix B).

**Minor comments are below:**

**l. 46-47: The sentence "depending on their optical properties, aerosols and clouds influence incident radiation by altering both the direct (increase) and diffuse (decrease) components" is unclear (in particular "increase" and "decrease").**

It was indeed unclear.

For clarity, the sentence was changed to:

"*However, depending on their optical properties, aerosols and clouds influence incident radiation by reducing the direct component while enhancing the diffuse component.* "

**l. 47-49: solar concentration systems rely mainly on the direct component. This may be mentioned as an additional supporting motivation.**

Done.

**Sections 2.1.2-2.1.5: are the various measurements synchronized? What is the uncertainty on the time determination of the various instruments? Synchronicity is crucial when dealing with clouds, and the use of data/images acquired at different rates and possibly different times may impact the results.**

Acquisitions are synchronized as the clock of the different computers are synchronized every hour with a time server.

**l. 107: according to the manufacturer, the spectral range defined by the 50% points is 200-3600 nm for CMP22, and 200 to 4000 nm for CHP1.**

Modified.

**l. 109: irradiance measurements are available at 1-minute tome resolution. Are these individual measurements or an average over a defined time interval? If an average, is the standard deviation also acquired?**

Measurements provide instantaneous SSI (50 ms), sampled at 1 minute resolution. We clarified lines 110-111 of the revised manuscript.

**l. 125: "important air mass" is not clear. What is intended? Relatively large AOD? In the literature there is a definite threshold on AOD at 440 nm needed to obtain reliable retrievals of SSA**

It is true that a definite threshold on AOD at 440 nm is needed to obtain reliable retrievals of SSA. However, the expression "important air mass" line 125 relates to a more general condition on the optical air mass of the atmosphere, which is mentioned on the AERONET website (https://aeronet.gsfc.nasa.gov/new_web/system_descriptions_operation.html):

"More than eight almucantar sequences are made daily at an optical air mass of 4, 3, 2 and 1.7 both morning and afternoon."

This condition alone translates in a range of solar zenith angles between 54° (air mass of 1.7) and 76° (air mass of 4), which greatly reduces the availability of AERONET inversions heedless of the aerosol optical depth.

We clarified as follows (see lines 127-128 of the revised text, changes are highlighted in bold text):

"However, as it requires a relatively important **optical** air mass **(SZA > 50°)** [...]"

**l. 149-150: how wind speed and direction affect simulations of surface solar irradiance?**

Wind speed and direction were used to identify consistencies in air mass identification and thus aerosol class.

It was an awkward sentence. We corrected as follows (see lines 152-154 of the revised text):

"Meteorological observations of relative humidity (RH) have also been used to perform simulations of the surface solar irradiance in clear-sky conditions, as described in Section 2.3.1, while wind speed, and wind direction measurements were used to complete the climatological study of the aerosol content and irradiance measurements in Section 3.1."

**l. 225-226: how time time delays between irradiance measurement and image is taken into account?**

There is a very high synchonicity (same clock).

Nonetheless, to account for the potential motion of clouds in or out of the pyrheliometer's field of view due to a potential time delay between the irradiance measurements and the sky images, we considered the scene as cloudy-sun when clouds close to the Sun's direction were moving towards the Sun. This was done by looking at the movement of clouds over consecutive sky images.

**l. 321: using relative humidity as one of the factors for aerosol classifications implies that the aerosol properties are essentially determined by the mixed layer properties, which is plausible at a continental site like Lille, where however sea breeze might still have some influence, producing a vertical differentiation in the aerosol characteristics. Are there evident dependencies of aerosol optical properties on surface relative humidity? Are there previous studies supporting this classification scheme?**

Rewiever 2 is right to wonder about the influence of the relative humidity on the aerosol optical properties. In the present study, we chose to represent aerosols as an homogeneous layer with particles of the same nature (i.e. aerosol class). This is consistent with the use of AERONET inversions of size distribution and complex refractive index, which represent the properties of the overall aerosol layer and are not specifically defined depending on the altitude.

The idea of using RH as an input to the look-up table of aerosol optical properties was inspired by the work of Hess et al. (1998, DOI: 10.1175/1520-0477(1998)079<0831:OPOAAC>2.0.CO;2), who defined several models of cloud and aerosol optical properties (OPAC package), which have been widely used since then. In their work, the aerosol optical properties were computed for several values of relative humidity.

As surface RH is the only long-term measurement of relative humidity in Lille, it was used to classify AERONET inversions, instead of RH values at higher altitudes. This is also coherent with our assumption of an homogeneous layer of aerosol optical properties.

It is true nonetheless that the aerosol layer is often not homogeneous, especially when sea breeze is involved. However, investigating the influence of the inhomogeneity of the aerosol layer is out of the scope of our study.

Finally, regarding the dependency of aerosol optical properties to surface RH, it should be mentioned that Table 4b shows that the overall logarithmic sensitivity of solar irradiances to typical changes in surface RH in Lille, represented by the coefficient of variation, is lower than 0.5%.

This suggests that the aerosol optical properties used within the framework of our study are quite insensitive to the surface relative humidity. Note that the results of Table 4b correspond to simulations for the Continental class of aerosols, but quite similar values were found for other aerosol classes.

***l. 246-249: it is not clear to me the advantage of using +/-30 minutes with respect to sunrise and sunset with respect to selecting an appropriate solar zenith angle. This makes probably more difficult a comparison between data obtained with different daily averaging methods.***

As stated line 248: "This limitation (i.e. the +/- 30 minutes criterion) was chosen to eliminate as many measurements in winter as in summer, which would not have been the case for a criterion based on the solar zenith angle.".

Nonetheless, it should be mentioned that the +/- 30 minutes criterion is small enough so that it is equivalent to using a criterion on the solar zenith angle of around 86°. This means that comparisons of different daily averages is still feasible, although the results presented in our study correspond to averages of 1-minute measurements over longer periods (monthly, yearly, seasonally, etc.). In contrast, the use of a greater time-delta (1 hour or more) would have lead, as you mentioned, to difficulties in comparing daily averages for different seasons.

***Figure 4: the legend defines "AERONET reference" what is called "overall" AOD in the caption.  I suggest using the same notation.***

Done.

***l. 422-423: it is not possible to determine the ciclonic/anticiclonic condition from the wind flow direction alone. This should be better explained.***

Changed to : "Moreover, the predominantly northeast wind flow observed in spring in Lille (Figure S4b), **could be the result of** a higher frequency of anticyclonic conditions during this season.", see lines 491-492 of the revised text.

***l. 430-432: this is not convincing.  Also airmasses from other directions appear to be possibly influenced by marine aerosol.***

It is true that air masses from other directions are possibly influenced by marine aerosols. However, other directions are more influenced by anthropogenic activities, especially from the Benelux region, which can overshadow the contribution of marine aerosols.

In contrast, Winter is characterized by prevailing winds from the West, which should be less influenced by anthropogenic activities.

The sentence was modified to highlight the prevailing influence of western winds (not south-west specifically), as follows (see lines 499-500 of the revised text, changes are in bold text):

« This finding aligns with surface wind direction and speed measurements from the ATOLL platform, which highlight prevailing **westerly winds** (Figure S2a), **that are less influenced by anthropogenic activities**. »

***l. 437: "minimum" istead of "minimal"***

Done.

**l. 478: please, define the limits of the spring and summer periods used in the analysis (different seasonal separations are found in the literature)**

Done, precisions were added regarding the defined seasons:

SPRING : March-April-May (MAM)

SUMMER: June-July-August (JJA)

**l. 486-488: does the statistical significance test take into account the uncertainty in the determination of the different conditions? I think these trends should be seen with some caution due to the uncertainties associated with the determinations of the specific conditions (in particular occurrence of clear sky with clouds conditions). Also in the conclusions, possible effects of the uncertainties on trend determinations should be mentioned (l. 1026-1029).**

We are aware that the performances of our classification of the sky conditions (Table 1) and the relatively short duration of our study period (only 13 years) may influence the statistical robustness of the trends presented in our analysis.

Nevertheless, these trends are consistent with other results from the literature obtained over longer periods of time.

Especially, the results presented by the Copernicus Climate Change Service (C3S) based on CMSAF CLARA-A3 and SARAH-3 data show an overall decrease in cloud fraction as well as increases in sunshine duration and surface solar irradiance over Europe since 1991, which are consistent with our findings.

Reference: https://climate.copernicus.eu/esotc/2023/clouds-and-solar-radiation

Furthermore, it should be mentioned that the Mann-Kendall tests used in our study take into account, to some extent, the uncertainty associated with the different quantities studied.

In the present study, the uncertainty in the frequency of the different sky conditions was set as 1 % for the seasonal Mann-Kendall tests. Further analysis even shows that trends in cloudy-sun occurrences presented in the study are valid up to 5% uncertainty, while clear-sun with clouds trends remain significant below 3 % uncertainty.

Therefore, the results presented in this study remain valid for the period considered.

It is true nonetheless that our study would benefit from an extended study period, which would improve the statistical significance of the observed trends.

As stated in response to a previous comment, a few cautionary sentences regarding the observed trends in the occurrences of sky conditions have been added to the revised version of Section 3.1.2 (lines 555-561 of the revised text):

"**These trends are issued from only 13 years of data, and rely on the imperfect filtering of sky conditions. Their scope should thus be considered with caution as they are quite sensitive to year-to-year variability, and as uncertainty in the occurrences of sky conditions impact their validity. However, it is worth noticing that the strongest (significantly decreasing) trend in CLOS situations was found to be associated with a high confidence in the related occurrences (1.2% of misidentifications) and that this observation is consistent with results from CM SAF data (C3S,**

**2024) that show since 2010 repeated negative annual anomalies in cloud cover and positive ones in sunshine duration, over European land areas and relative to the reference period 1991-2020."**

A reference to the C3S study has also been added regarding trends in SSI (lines 603-604 of the revised text):

**"It is also coherent with the general increase in SSI and sunshine duration observed over Europe since 1991 based on CM SAF SARAH-3 data (C3S, 2024)."**

In addition, the following paragraph was added to the conclusions (lines 920-924 of the revised text):

**«Note that although the statistical robustness of our trends may be affected by the performances of our classification of sky conditions and the length of the study period (13 years), they appear consistent, especially under all-sky conditions, with longer-term results in Europe (C3S, 2024; Ningombam et al., 2019; Boers et al., 2017).»**

**l. 590 and 599: all the terms right of sigma should be included in parentheses**

Done.

**l. 618-619; see previous comment**

Done.

**l. 654: Do the author use a logarithmic aerosol vertical profile, or a single layer at a specific height? Please, specify what is intended with aerosol layer height.**

Precisions were added in Section 2.3.1 on the description of the aerosol vertical profile (lines 286-287 of the revised text):

**"In this study, the aerosol layer was defined as an exponential decay of the aerosol density with a 2 km scale height (i.e. $AOD_{\lambda,mix}(z) = AOD_{\lambda,mix} \times e^{-z/2}$, with z the altitude in km)."**

In this case the aerosol layer height corresponds to the 2 km scale height defined in the study.

**l. 808: using the same symbol for the absolute and the relative direct radiative effect is misleading.**

Based on a comment from the other reviewer, the acronym "DRE" has been changed to "$RE_d$" (***R**adiative **E**ffect on the **d**ownwelling surface solar irradiance)

The relative DRE (newly $RE_d$) was changed to "rDRE" ($rRE_d$).

**Section 4.1.1: it may be worth mentioning that, since DRE depends strongly also on SZA, changes in the time/seasonal distribution of the occurred changes may have affected the results.**

The influence of the SZA on the observed minimum of aerosol radiative effect in winter was already mentioned lines 840-841. However, the sentence wasn't correct nor complete.

It was changed as follows (see lines 782-783 of the revised manuscript, changes are in bold text):

"It can also be related to **higher solar zenith angles** that lead to overall lower surface irradiances **in winter, and thus lower aerosol $RE_d$ in absolute values**."